# Controls on the water vapor isotopic composition near the surface of tropical oceans and role of boundary layer mixing processes

Camille Risi[1], Joseph Galewsky[2], Gilles Reverdin[3], and Florent Brient[4]

[1]Laboratoire de Météorologie Dynamique, IPSL, CNRS, Sorbonne Université, Paris, France
[2]Department of Earth and Planetary Sciences, University of New Mexico, Albuquerque, USA
[3]Sorbonne Université, CNRD/IRD/MNHN, LOCEAN, IPSL, Paris, France
[4]CNRM, Université de Toulouse, Météo-France, CNRS, Toulouse, France

**Correspondence:** Camille Risi (Camille.Risi@lmd.jussieu.fr)

**Abstract.** Understanding what controls the water vapor isotopic composition of the sub-cloud layer (SCL) over tropical oceans ($\delta D_0$) is a first step towards understanding the water vapor isotopic composition everywhere in the troposphere. We propose an analytical model to predict $\delta D_0$ motivated by the hypothesis that the altitude from which the free tropospheric air originates ($z_{orig}$) is an important factor: when the air mixing into the SCL is lower in altitude, it is generally moister, and thus it depletes more efficiently the SCL. We extend previous simple box models of the SCL by prescribing the shape of $\delta D$ vertical profiles as a function of humidity profiles and by accounting for rain evaporation and horizontal advection effects. The model relies on the assumption that $\delta D$ profiles are steeper than mixing lines, and that the SCL is at steady state, restricting its applications to time scales longer than daily. In the model, $\delta D_0$ is expressed as a function of $z_{orig}$, humidity and temperature profiles, surface conditions, a parameter describing the steepness of the $\delta D$ vertical gradient and a few parameters describing rain evaporation and horizontal advection effects. We show that $\delta D_0$ does not depend on the intensity of entrainment, in contrast to several previous studies that had hoped that $\delta D_0$ measurements could help estimate this quantity.

Based on an isotope-enabled general circulation model simulation, we show that $\delta D_0$ variations are mainly controlled by mid-tropospheric depletion and rain evaporation in ascending regions, and by sea surface temperature and $z_{orig}$ in subsiding regions. In turn, could $\delta D_0$ measurements help estimate $z_{orig}$ and thus discriminate between different mixing processes? For such isotope-based estimates of $z_{orig}$ to be useful, we would need a precision of a few hundred meters in deep convective regions and smaller than 20 m in stratocumulus regions. To reach this target, we would need daily measurements of $\delta D$ in the mid-troposphere and accurate measurements of $\delta D_0$ (accuracy down to 0.1 ‰ in the case of stratocumulus clouds, which is currently difficult to obtain). We would also need information on the horizontal distribution of $\delta D$ to account for horizontal advection effects, and full $\delta D$ profiles to quantify the uncertainty associated with the assumed shape for $\delta D$ profiles. Finally, rain evaporation is an issue in all regimes, even in stratocumulus clouds. Innovative techniques would need to be developed to quantify this effect from observations.

# 1 Introduction

## 1.1 What controls the water vapor isotopic composition?

The water vapor isotopic composition (e.g. $\delta D = (R/R_{SMOW} - 1) \times 1000$ expressed in ‰, where $R$ is the D/H ratio and SMOW is the Standard Mean Ocean Water reference) has been shown to be sensitive to a wide range of atmospheric pro-
cesses (Galewsky et al., 2016), such as continental recycling (Salati et al., 1979; Risi et al., 2013), unsaturated downdrafts (Risi et al., 2008, 2010a), rain evaporation (Worden et al., 2007; Field et al., 2010), the degree of organization of convection (Lawrence et al., 2004; Tremoy et al., 2014), the convective depth (Lacour et al., 2017b), the proportion of precipitation that occurs as convective or large-scale precipitation (Lee et al., 2009; Kurita, 2013; Aggarwal et al., 2016), vertical mixing in the lower troposphere (Benetti et al., 2015; Galewsky, 2018a, b), mid-troposphere (Risi et al., 2012b) or upper-troposphere
(Galewsky and Samuels-Crow, 2014), convective detrainment (Moyer et al., 1996; Webster and Heymsfield, 2003), ice microphysics (Bolot et al., 2013). It is therefore very challenging to quantitatively understand what controls the isotopic composition of water vapor.

A first step towards this goal is to understand what controls the water vapor isotopic composition in the sub-cloud layer (SCL) of tropical (30°S-30°N) oceans. Indeed, this water vapor is an important source moistening air masses traveling to land regions
(Gimeno et al., 2010; Ent and Savenije, 2013) and towards higher latitudes (Ciais et al., 1995; Delaygue et al., 2000). It is also ultimately the only source of water vapor in the tropical free troposphere, since water vapor in the free troposphere ultimately originates from convective detrainment (Sherwood, 1996), and convection ultimately feeds from the SCL air (Bony et al., 2008). Therefore, the water vapor isotopic composition in the SCL of tropical oceans serves as initial conditions to understand the isotopic composition in land waters and in the tropospheric water vapor everywhere on Earth. We focus here on the SCL
because, by definition, there is no complication by cloud condensation processes.

The goal of this paper is thus to propose a simple analytical equation that allows us to understand and quantify the factors controlling the $\delta D$ in the water vapor in the SCL of tropical oceans. So far, the most famous analytical equation for this purpose has been the closure equation developed by Merlivat and Jouzel (1979) (MJ79). This closure equation can be derived by assuming that all the water vapor in the SCL air originates from surface evaporation. The water balance of the SCL can be
closed by assuming a mass export at the SCL top (e.g. by convective mass fluxes) and a totally dry entrainment into the SCL to compensate this mass export. The MJ79 equation has proved very useful to capture the sensitivity of $\delta D$ and second-order parameter d-excess to sea-surface conditions Merlivat and Jouzel (1979); Ciais et al. (1995); Risi et al. (2010d). However, the $\delta D$ calculated from this equation suffers from a high bias in tropical regions Jouzel and Koster (1996). This bias can be explained by the neglect of vertical mixing between the SCL and air entrained from the free troposphere (FT). The MJ79
equation can better reproduce surface water vapor observation when extended to take into account this mixing (Benetti et al. (2015), hereafter B15). This extension requires to know the specific humidity ($q$) and water vapor $\delta D$ of the entrained air. To get these values, they assume that the air entrained into the boundary layer comes from a constant altitude. However, this does not reflect the complexity of entrainment and mixing processes in marine boundary layers.

## 1.2  Entrainment and mixing mechanisms

Figure 1 summarizes our knowledge about these entrainment and mixing processes. In stratocumulus regions, clouds are thin and the inversion is just above the lifting condensation level (LCL). Air is entrained from the FT by cloud-top entrainment driven by radiative cooling or wind shear instabilities (Mellado, 2017), possibly amplified by evaporative cooling of droplets (Lozar and Mellado, 2015). Both Direct Numerical Simulations (Mellado, 2017) and observations of tracers (Faloona et al., 2005) and cloud holes (Gerber et al., 2005) show that air is entrained from a thin layer above the inversion, thinner than 80 m and as small as 5 m. The boundary layer itself is animated by updrafts, downdrafts and associated turbulent shells that bring air from the cloud layer downward (Brient et al., 2019; Davini et al., 2017).

In trade-wind cumulus regions, the cloudy layer is a bit deeper. Observational studies and large-eddy simulations have pointed out the important role of thin subsiding shells around cumulus clouds, driven by cloud-top radiative cooling, mixing and evaporative cooling of droplets (Jonas, 1990; Rodts et al., 2003; Heus and Jonker, 2008; Heus et al., 2009; Park et al., 2016). This brings air from the cloudy layer to the SCL. Subsiding shells may also cover overshooting plumes of the cumulus clouds, entraining FT air into the cloud layer (Heus and Jonker, 2008).

In deep convective regions, unsaturated downdrafts driven by rain evaporation (Zipser, 1977) are known to contribute significantly to the energy budget of the SCL (Emanuel et al., 1994). Large-eddy simulations show that subsiding shells, similar to those documented in shallow convection, also exist around deep convective clouds (Glenn and Krueger, 2014). In the clear-sky environment between clouds, turbulent entrainment into the SCL may also play a significant role (Thayer-Calder and Randall, 2015).

Therefore, whatever the cloud regime, air entering the SCL from above may originate either from the cloud layer or from the free troposphere, depending on the mixing mechanism. Therefore, in this paper in contrast with B15, we let the altitude from which the air originates, $z_{orig}$, be variable. We do not call it "entrained" air because entrainment sometimes refer to mixing processes through an interface (e.g. De Rooy et al., 2013; Davini et al., 2017), whereas air in the SCL may also enter through deep, coherent and penetrative structures such as unsaturated downdrafts. We do not call it FT air either, since it may originate from the cloudy layer.

## 1.3  Goal of the article

To acknowledge the diversity and complexity of mixing mechanisms, we extend the B15 framework in several ways. First, we assume that we know the shape of $\delta D$ profiles as a function of $q$. Second, we write the specific humidity of the air originating from above the SCL as a function of $z_{orig}$. Third, we account for rain evaporation and horizontal effects.

While B15 focused on observations during a field campaign, we also apply the extended equation to global outputs of an isotope-enabled general circulation model, with the aim to quantify the different factors controlling the $\delta D$ variability in the Tropics. The variable $z_{orig}$ will emerge as an important factor. Therefore, we discuss the possibility that $\delta D$ measurements at the near surface and through the lower FT could help estimate $z_{orig}$, and thus the mixing processes between the SCL and the air above.

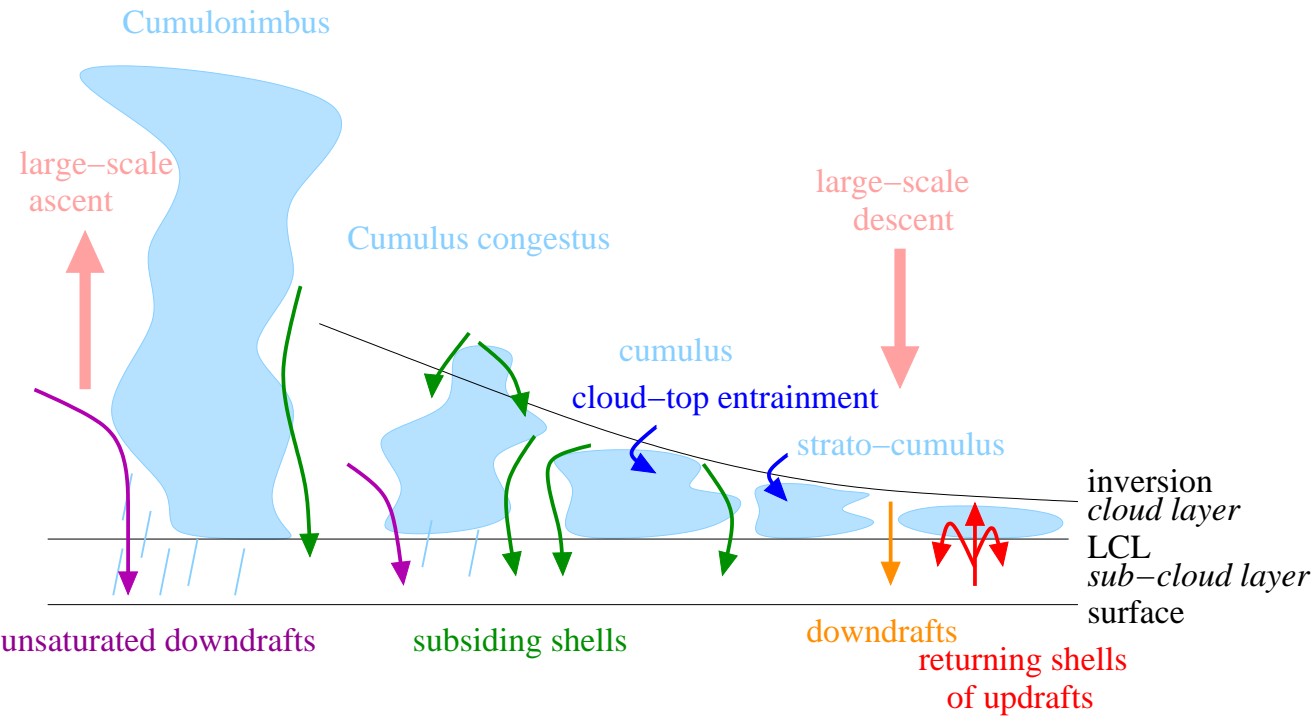

**Figure 1.** Schematics showing the different types of clouds and mixing processes as a function of the large-scale circulation.

Note that we focus on $\delta D$ only. Results for $\delta^{18}O$ are similar. We do not aim at capturing the second-order parameter d-excess, because our model requires some knowledge about free tropospheric vertical profiles of isotopic composition. While $\delta D$ is known to decrease with altitude (Ehhalt, 1974; Ehhalt et al., 2005; Sodemann et al., 2017), vertical profiles of d-excess are more diverse and less well understood (Sodemann et al., 2017)). In addition, the need for an extension of MJ79 is more needed for $\delta D$ than for d-excess, since the effect of convective mixing is larger on $\delta D$ than on d-excess (Risi et al., 2010d; Benetti et al., 2014).

## 2    Theoretical framework

### 2.1    Box model and budget equations

Building on Benetti et al. (2014) and B15, we consider a simple box representing the SCL (Fig. 2). We assume that the air comes from above ($M$) and from the incoming large-scale horizontal advection ($F_{adv}$), and is exported through the SCL top ($N$, e.g. turbulent mixing or convective mass flux) and by outgoing large-scale horizontal advection ($F_{adv,out}$). We assume that the SCL is at steady state. For example, its depth is constant. Since the SCL properties may exhibit a diurnal cycle

(Duynkerke et al., 2004), this hypothesis restricts the application of this model to time scales longer than daily. The air mass budget of the SCL thus writes:

$$M + F_{adv} = N + F_{adv,out} \tag{1}$$

These fluxes also transport water vapor and isotopes. In addition, surface evaporation $E$ and rain evaporation $F_{evap}$ import water vapor and isotopes (Fig. 2).

Hereafter, to simplify equations, we use the isotopic ratio $R$ instead of $\delta D$.

The SCL is usually well-mixed (Betts and Ridgway, 1989; Stevens, 2006; De Roode et al., 2016). We thus assume that the humidity and isotopic properties are constant vertically and horizontally in the SCL. They are noted $(q_0, R_0)$. The humidity and isotopic properties of the mass flux export $N$ are thus also $(q_0, R_0)$. The properties of the flux $M$ are noted $(q_{orig}, R_{orig})$. The properties of the incoming air by horizontal advection are noted $(q_{adv}, R_{adv})$. For simplicity we neglect here the effect of horizontal gradients in humidity (i.e. $q_{adv} = q_0$), assuming that the main effect of horizontal advection on $\delta D_0$ arises from horizontal gradients in $\delta D$. Appendix C explains how $R_{adv}$ can be calculated. At steady state, the water budget of the SCL writes:

$$M \cdot q_{orig} + E + F_{evap} + F_{adv} \cdot q_0 = (N + F_{adv,out}) \cdot q_0 \tag{2}$$

This model is consistent with SCL water budgets that have already been derived in previous studies (Bretherton et al., 1995), except that we consider steady state. This equation can be solved for $q_0$:

$$q_0 = q_{orig} + \frac{E + F_{evap}}{M} \tag{3}$$

The SCL humidity $q_0$ is thus sensitive to $M$, justifying that it can be used to estimate the mixing intensity or the "entrainment velocity" $w_e = M/\rho_0$ ($\rho$ being the air volumic mass) (Bretherton et al., 1995).

At steady state, the water isotope budget of the SCL writes:

$$M \cdot q_{orig} \cdot R_{orig} + E \cdot R_E + F_{evap} \cdot R_{evap} + F_{adv} \cdot q_0 \cdot R_{adv} = (N + F_{adv,out}) \cdot q_0 \cdot R_0 \tag{4}$$

where $R_E$ is the isotopic composition of the surface evaporation. It is assumed to follow the Craig and Gordon (1965) equation:

$$R_E = \frac{R_{oce}/\alpha_{eq} - h_0 \cdot R_0}{\alpha_K \cdot (1 - h_0)} \tag{5}$$

where $R_{oce}$ is the isotopic ratio in the surface ocean water, $\alpha_{eq}$ is the equilibrium fractionation calculated at the ocean surface temperature (SST) (Majoube, 1971), $\alpha_K$ is the kinetic fractionation coefficient (MJ79) and $h_0$ is the relative humidity normalized at the SST ($h_0 = q_0/q_s(SST, P_0)$ where $q_s$ is the saturation specific humidity at SST and $P_0$ is the surface pressure).

We write the isotopic composition of the rain evaporation, $R_{evap}$, as:

$$R_{evap} = \alpha_{evap} \cdot R_0$$

where $\alpha_{evap}$ is an effective fractionation coefficient. For example, if droplets are formed near the cloud base, some of them precipitate and evaporate totally into the SCL (e.g. in non-precipitating shallow cumulus clouds), then $\alpha_{evap} = \alpha(T_{cloudbase})$. In contrast, if droplets are formed in deep convective updrafts after total condensation of the SCL vapor, and then a very small

fraction of the rain is evaporated into a very dry SCL, then $\alpha_{evap} = 1/\alpha(T_{SCL})/\alpha_K$ (Stewart, 1975).

We note $\eta = F_{evap}/E$ the ratio of water vapor coming from rain evaporation to that of surface evaporation, and $\phi = F_{adv} \cdot q_{adv}/E$ the ratio of water vapor coming from horizontal advection to that coming from surface evaporation. We note $\beta = R_{adv}/R_0$ the ratio of isotopic ratios of horizontal advection to that of the SCL.

Note that in all our equations, we assume that temperature and humidity profiles and all basic surface meteorological vari-

ables are known. We attempt to express neither $h_0$ as a function of $q_0$ as in B15, nor the $q$ profile as a function of $q_0$. Our ultimate goal is to assess the added value of $\delta D$ assuming that meteorological measurements are already routinely done.

By combining all these equations, we get:

$$R_0 = \frac{(1 - r_{orig}) \cdot R_{oce}/\alpha_{eq} + \alpha_K \cdot (1 - h_0) \cdot r_{orig} \cdot (1 + \eta) \cdot R_{orig}}{(1 - r_{orig}) \cdot h_0 + \alpha_K \cdot (1 - h_0) \cdot (1 + \eta + (1 - r_{orig}) \cdot (\phi \cdot (1 - \beta) - \eta \cdot \alpha_{evap})}$$    (6)

where $r_{orig} = q_{orig}/q_0$ is the proportion of the water vapor in the SCL that originates from above.

An intriguing aspect of this equation is that the sensitivity to $M$ disappears. In contrast to $q_0$, $R_0$ is not sensitive to $M$. Therefore, it appears illusory to promise that water vapor isotopic measurements could help constrain the entrainment velocity that many studies have striven to estimate (Nicholls and Turton, 1986; Khalsa, 1993; Wang and Albrecht, 1994; Bretherton et al., 1995; Faloona et al., 2005; Gerber et al., 2005, 2013). The lack of sensitivity of $R_0$ to $M$ is explained physically by the fact that for a given $q_0$ and $q_{orig}$, if $M$ increases, then $E + F_{evap}$ increases in the same proportion to maintain the water balance.

Therefore, the relative proportion of the water vapor originating from surface and rain evaporation to that coming from above, to which $R_0$ is sensitive, remains constant. Rather, since $q$ and $R$ vary with altitude, $R_0$ is sensitive to the altitude from which the air originates.

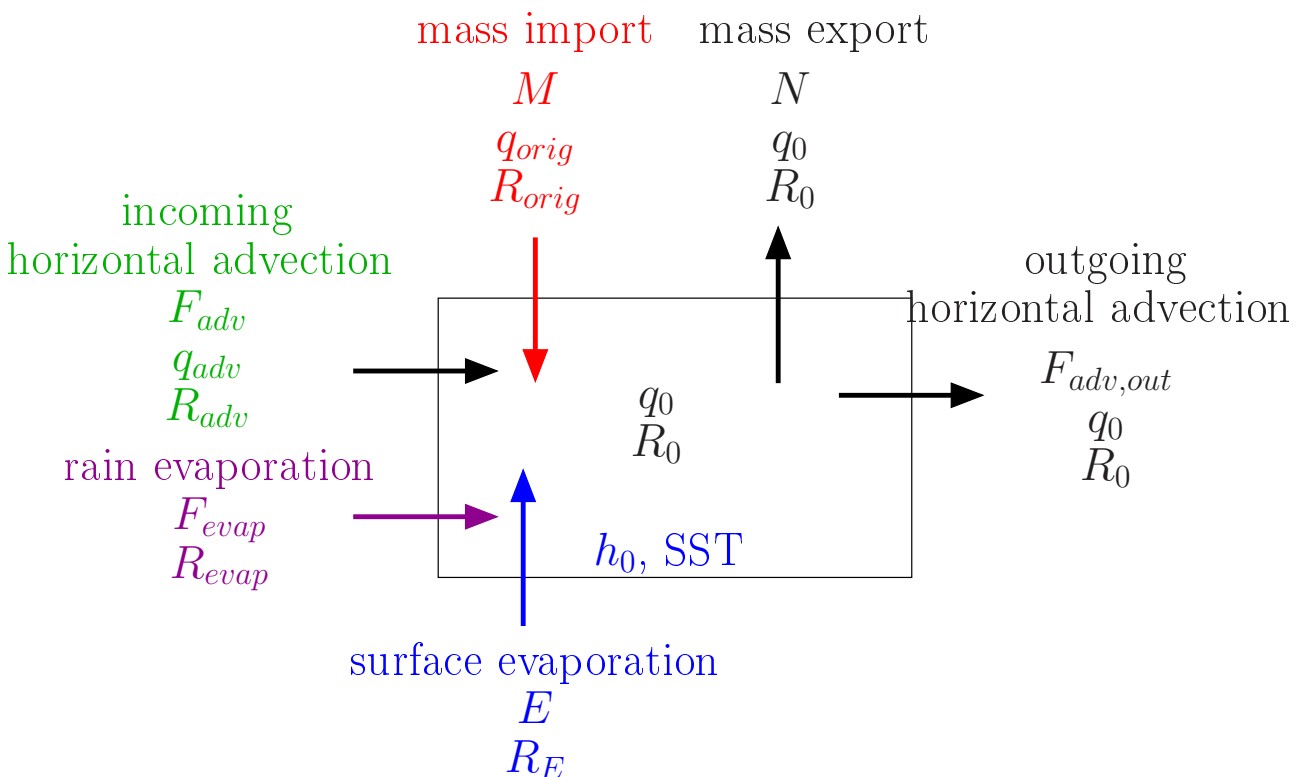

**Figure 2.** Schematics showing the simple box model on which the theoretical framework is based, and illustrating the main notations.

## 2.2 Closure if $\delta D$ profile follows a Rayleigh distillation line

Eq. (6) requires to know $q_{orig}$ and $R_{orig}$. B15 take these values from GCM outputs at 700 hPa. In contrast, here we acknowledge the diversity and complexity of mixing mechanisms by keeping the possibility to take $q_{orig}$ and $R_{orig}$ at a variable altitude $z_{orig}$.

5    If the goal is to predict $R_0$ from $z_{orig}$ , we can apply Eq. (6) if we know the $q$ and $\delta D$ vertical profiles. Conversely, if the goal is to predict $z_{orig}$ from $R_0$, we can numerically solve Eq. (6) if we know the $q$ and $\delta D$ vertical profiles. No analytical solution exists in the general case, but a numerical solution can be searched for $z_{orig}$ based on Eq. (6). However, the existence and unicity of the solution is not warranted for all kinds of profiles (e.g. Appendix A).

In practice, full isotopic profiles are costly to measure. In addition, our goal is to develop an analytical model. Therefore, in
10    the following we simplify the problem by assuming that the vertical profile of $R$ follows a known relationship as a function of $q$. Measured vertical profiles of $\delta D$ are usually bounded by two curves when plotted in a $(q, \delta D)$ diagram (Sodemann et al., 2017): Rayleigh distillation curve and mixing line.

First, we explore the case of a Rayleigh distillation curve (Dansgaard, 1964), as in Galewsky and Rabanus (2016):

$$R_{orig} = R_0 \cdot r_{orig}^{\alpha_{eff}-1} \tag{7}$$

where $\alpha_{eff}$ is an effective fractionation coefficient. Typically, $q$ decreases with altitude, so $R$ also decreases with altitude. However, in observations and models, vertical profiles of $R$ can be very diverse (Bony et al., 2008; Sodemann et al., 2017). The water vapor may be more (Worden et al., 2007) or less (Sodemann et al., 2017) depleted than predicted by Rayleigh curve using a realistic fractionation factor that depends on local temperature. Therefore, here we let $\alpha_{eff}$ be a free parameter larger than 1. Rather than assuming a true Rayleigh curve, we simply assume that $R$ and $q$ are logarithmically related. Effects of horizontal advection and rain evaporation on tropospheric profiles are encapsulated into $\alpha_{eff}$.

Injecting Eq. (7) into Eq. (6), we get:

$$R_0 = \frac{R_{oce}}{\alpha_{eq}} \cdot \frac{1}{h_0 + \alpha_K \cdot (1-h_0) \cdot \left( (1+\eta) \cdot \frac{1-r_{orig}^{\alpha_{eff}}}{1-r_{orig}} - \eta \cdot \alpha_{evap} + \phi \cdot (1-\beta) \right)} \tag{8}$$

A simpler form can be found if neglecting horizontal advection and rain evaporation effects ($\phi = \eta = 0$):

$$R_0 = \frac{R_{oce}}{\alpha_{eq}} \cdot \frac{1}{h_0 + \alpha_K \cdot (1-h_0) \cdot \frac{1-r_{orig}^{\alpha_{eff}}}{1-r_{orig}}} \tag{9}$$

As a consistency check, in the limit case where the air coming from above is totally dry ($r_{orig} = 0$), Eq. (9) becomes the MJ79 equation:

$$R_0 = \frac{R_{oce}}{\alpha_{eq}} \cdot \frac{1}{h_0 + \alpha_K \cdot (1-h_0)} \tag{10}$$

Equation (8) tells us that whenever $\alpha_{eff} > 1$, $R_0$ decreases as $r_{orig}$ increases (Fig. 3 red), i.e. as $q_{orig}$ is moister. Therefore, $R_0$ decreases as $z_{orig}$ is lower in altitude. This result may be counter-intuitive, but can be physically interpreted as follows. If $z_{orig}$ is high, mixing brings air with very depleted water vapor, but since the air is dry, the depleting effect is small. In contrast, if $z_{orig}$ is low, mixing brings air with water vapor that is not very depleted, but since the air is moist, the depleting effect is large (Fig. 4a).

Figure 3 (red) shows that the range of possible $\delta D$ values is restricted to -70 ‰ to -85 ‰. This explains why in quiescent conditions near the sea level in tropical ocean locations, the water vapor $\delta D$ varies little (Benetti et al. (2014), F. Vimeux pers. comm.). In the limit case where $r_{orig} \to 1$ (i.e. the air comes from the SCL top), $R_0 \to \frac{R_{oce}}{\alpha_{eq}} \cdot \frac{1}{h_0 + \alpha_K \cdot (1-h_0) \cdot \alpha_{eff}}$ (L'Hopital's rule was used to calculate this limit). This lower bound is not so depleted compared to the more depleted water vapor observed in regions of deep convection (e.g. Lawrence et al., 2002; Lawrence et al., 2004; Kurita, 2013). This is because when $r_{orig} \to 1$,

the water vapor coming from above has a composition very close to that of the SCL, so the depleting effect is limited. In addition, surface evaporation strongly damps the depleting effect of mixing. Only rain evaporation or liquid-vapor exchanges (Lawrence et al., 2004; Worden et al., 2007) can further decrease $R_0$ (Appendix B).

Figure 3 (green) shows that the sensitivity to $\alpha_{eff}$ is relatively small but cannot be neglected. Therefore, predicting $\delta D_0$ requires to have some knowledge about the steepness of the isotopic profiles in the FT. Rain evaporation and horizontal advection can have either an enriching or depleting effect, but do not qualitatively change the results (Fig. 3 purple and blue).

Now we consider the case of a mixing line. Detailed calculation in Appendix A show that the sensitivity to $r_{orig}$ is lost. An infinity of FT end members can lead to the same $\delta D_0$ when mixed with the surface evaporation, as illustrated in Fig. 4b and analytically demonstrated in Appendix A. Our main results (more depleted $\delta D_0$ as $r_{orig}$ increases, restricted range of $\delta D_0$ variations, relationship with $z_{orig}$) hold only for $\delta D$ profiles that are steeper than a mixing line. This is the case for profiles that are intermediate between a Rayleigh and a mixing line, as is usually the case in nature (Sodemann et al., 2017) or in a general circulation model (Appendix D1).

## 3   Model simulations, observations and methods

### 3.1   LMDZ simulations

We use an isotope-enabled general circulation model (GCM) as a laboratory to test our hypotheses and investigate what controls the isotopic composition. We use the LMDZ5A version of LMDZ (Laboratoire de Météorologie Dynamique Zoom), which is the atmospheric component of the IPSL-CM5A coupled model (Dufresne et al., 2012) that took part in CMIP5 (Coupled Model Intercomparison Project, Taylor et al. (2012)). This version is very close to LMDZ4 (Hourdin et al., 2006). Water isotopes are implemented the same way as in its predecessor LMDZ4 (Risi et al., 2010c). We use 4 years (2009-2012) of an AMIP (Atmospheric Model Intercomparison Project)-type simulation (Gates, 1992) that was initialized in 1977. The winds are nudged towards ERA-40 reanalyses (Uppala et al., 2005) to ensure a more realistic simulation. Such a simulation has already been described and extensively validated for isotopic variables in both precipitation and water vapor (Risi et al., 2010c, 2012a). The ocean surface water $\delta D_{oce}$ is assumed constant and set to 4 ‰. The resolution is 2.5° in latitude × 3.75° in longitude, with 39 vertical levels. Over the ocean, the first layer extends up to 64 m, and a typical SCL extending up to 600 m is resolved by 6 layers. Around 2500 m, a typical altitude for the inversion for trade-wind cumulus clouds, the resolution is about 500 m.

For our calculations, we only use tropical grid boxes (30°S-30°N) over tropical oceans (>80% ocean fraction in the grid box). In addition, to avoid numerical problems when estimating effect of horizontal advection and rain evaporation, only grid boxes and days where $E > 0.5$ mm/d are considered. This represents 99.7% of all tropical oceanic grid boxes.

Specific diagnostics for horizontal advection and rain evaporation are detailed in Appendix B and C.

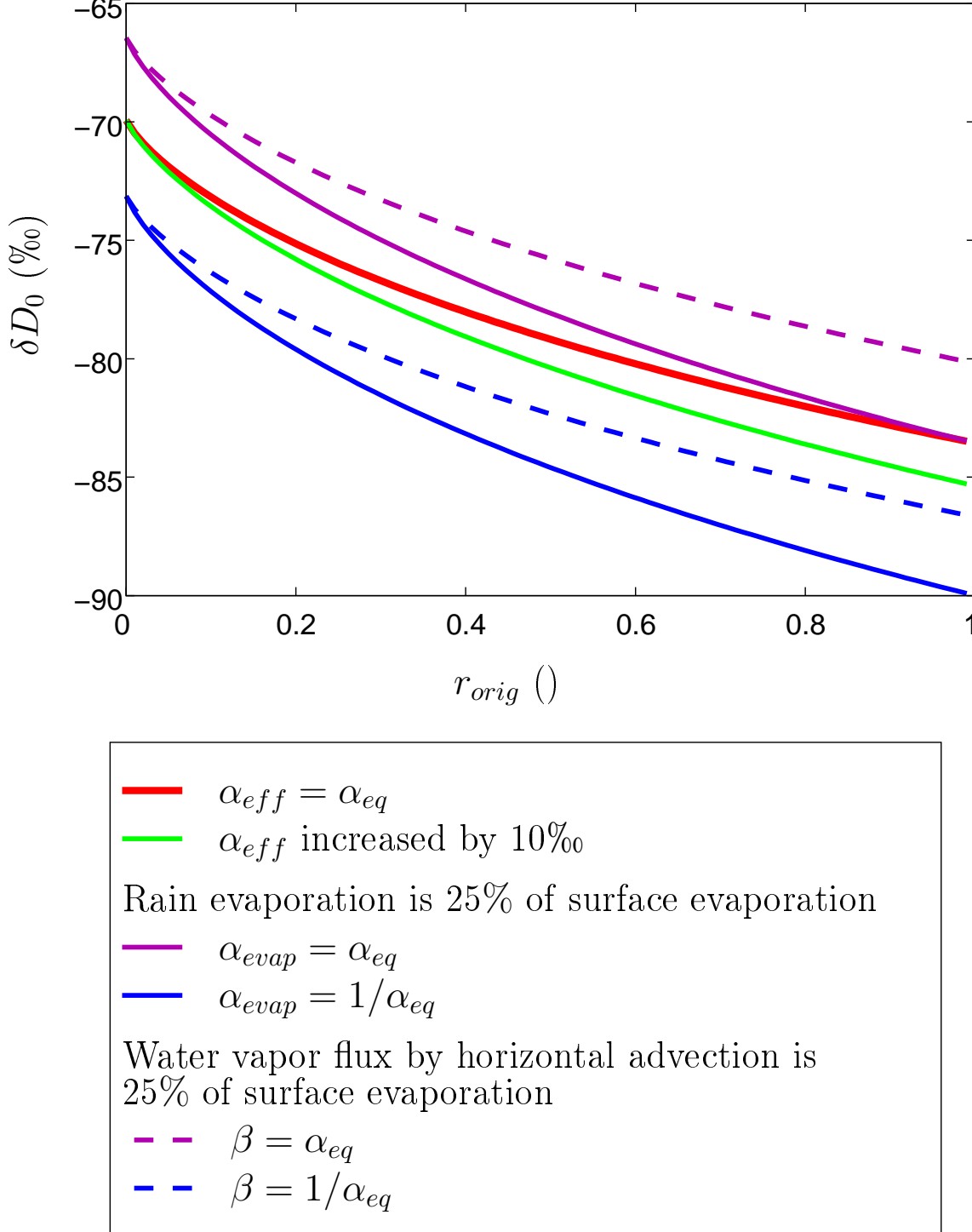

**Figure 3.** $\delta D_0$ as a function of $r_{orig}$ according to Eq. (9), with $\alpha_{eff} = \alpha_{eq}$ as an example (red). For this illustrative purpose, we assume SST=30°C, $h_0 = 0.8$, $\delta D_{oce} = 0$‰ and $\phi = \eta = 0$. The sensitivity to the effective fractionation factor $\alpha_{eff}$ (green) is shown. If rain evaporation is 25% of surface evaporation ($\eta = 0.25$), the solid pink and blue curves show the sensitivity to the effective fractionation factor $\alpha_{evap}$. If the incoming water vapor by horizontal advection is 25% of surface evaporation ($\phi = 0.25$), the dashed pink and blue curves show the sensitivity to the isotopic gradient quantified by $\beta$.

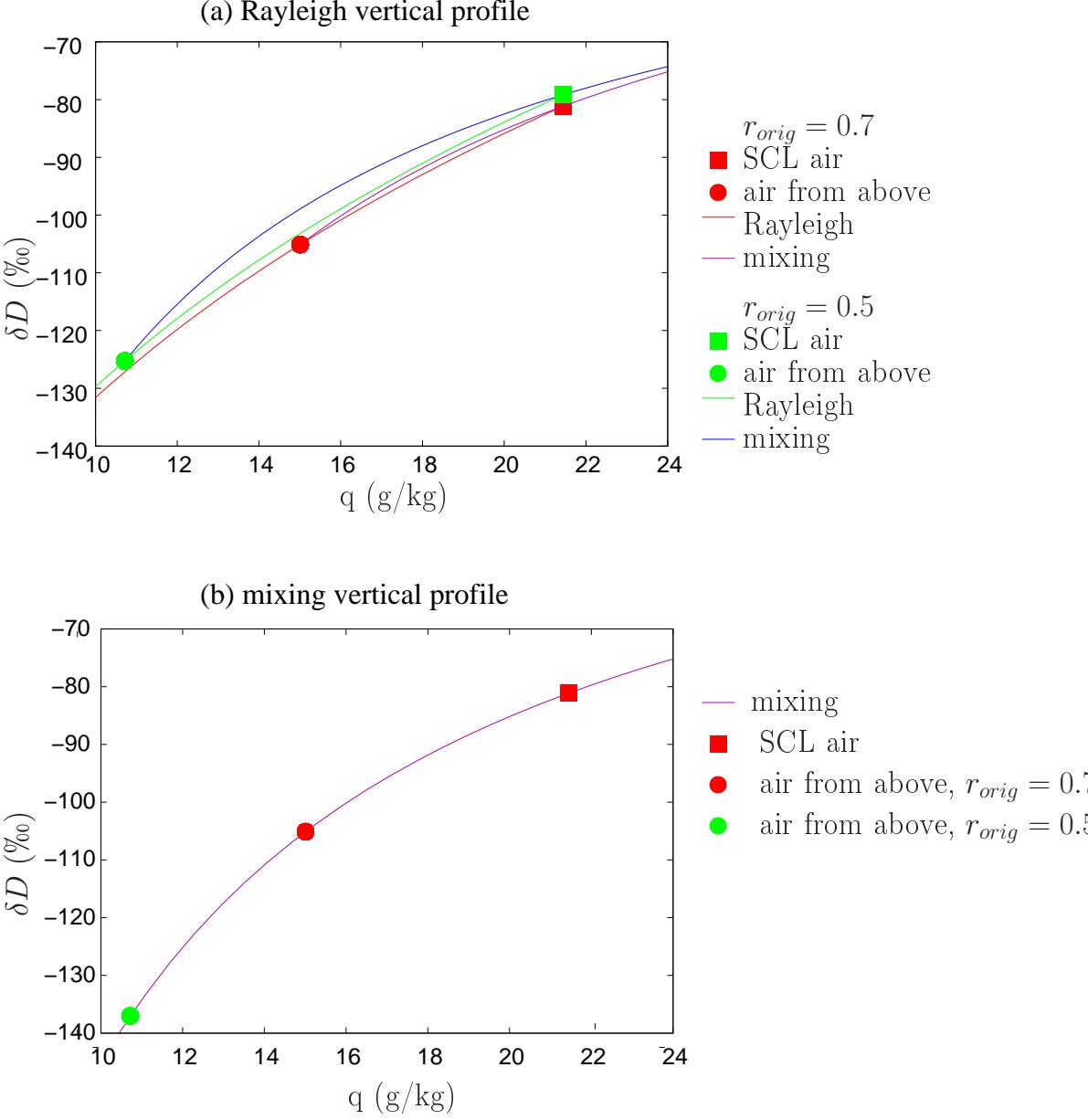

**Figure 4.** Idealized $q - \delta D$ diagrams showing how the SCL water vapor $\delta D$ is set, for the case where the tropospheric $\delta D$ profile follow Rayleigh distillation (a) or a mixing line (b). For this illustrative purpose, we assume SST=30 °C, $h_0 = 0.8$, $\delta D_{oce} = 0\,‰$ and and $\phi = \eta = 0$. In (a), the red curve shows the Rayleigh profile starting from the SCL and the purple curve shows the mixing line connecting the air coming from above to the surface evaporation, in the case $r_{orig} = q_{orig}/q_0 = 0.7$. The green curve shows the the Rayleigh profile starting from the SCL and the blue curve shows the mixing line connecting the air coming from above to the surface evaporation, in the case $r_{orig} = q_{orig}/q_0 = 0.5$. One can visually see that when $r_{orig}$ is lower, the mixing line is more curved, leading to more enriched values. In (b), the purple curve joins the SCL air and the air at all altitudes above the SCL. One can see that different values for $r_{orig}$ can lead to the same value of $\delta D$ in the SCL.

## 3.2 STRASSE observations

We also apply our theoretical framework to observations during the STRASSE (subtropical Atlantic surface salinity experiment) cruise that took place in the Northern subtropical ocean in August and September 2012 (Benetti et al., 2014). This campaign accumulates several advantages that are important for our analysis: (1) continuous $\delta D_0$ measurements in the surface water vapor (17m) at a high temporal frequency during one month (Benetti et al., 2014, 2015, 2017b), (2) associated surface meteorological measurements, including SST and $h_0$, (3) 22 radio-soundings relatively well distributed over the campaign period and providing vertical profiles of altitude, temperature, relative humidity and pressure, (4) ocean surface water $\delta D_{oce}$ measurements (Benetti et al., 2017a), (5) a variety of conditions ranging from quiescent weather to convective conditions, (6) on many vertical profiles, a well defined temperature inversion allows to calculate the inversion altitude.

We use $\delta D_0$ measurements on a 15-minute time step. The measurements in ocean water were interpolated on the same time steps using a Gaussian filter with a width of 3 days. The radio-soundings are used together with all water vapor isotopic measurements that are within 30 minutes of the radio-sounding launch. Only profiles during the ascending phase of the balloon are considered, because the descent phase is often located far away from the initial launch point (McGrath et al., 2006; Seidel et al., 2011).

## 3.3 Estimating the altitude from which the air originates

Here we explain how $z_{orig}$ is estimated based on LMDZ outputs. First, we assume that the $q$ and $\delta D$ at 500 hPa ($q_f$, $\delta D_f$) belong to a Rayleigh distillation line starting from the surface with effective fractionation $\alpha_{eff}$:

$$\alpha_{eff} = 1 + \frac{ln(R_f/R_0)}{ln(q_f/q_0)}$$

In a real field campaign, this assumption means that we do not need to measure the full vertical profile of $\delta D$, but only $\delta D_f$ at a given free tropospheric altitude (e.g. 500 hPa).

We checked that results are similar when defining the end member at 400 hPa rather than 500 hPa. However, the end member should be defined above 500 hPa to ensure that it is well above boundary layer processes. If the end member is defined below 500 hPa (e.g. 600 hPa), there are a few cases where $q$ increases with altitude ($q_f > q_0$) due to horizontal advection or convective detrainment from nearby moister regions; meanwhile, $\delta D$ decreases monotonically, leading to unrealistic values for $\alpha_{eff}$.

Second, $r_{orig}$ is estimated based on Eq. (9), using $\alpha_{eff}$, $\alpha_{eq}$, $\alpha_K$ , $\delta D_{oce}$, $h_0$ and $\delta D_0$ simulated by LMDZ.

Third, the altitude $z_{orig}$ is estimated from $r_{orig}$. Using the $q$ vertical profile, we find $z_{orig}$ so that $q(z_{orig}) = r_{orig} \cdot q_0$ (Fig. 5, red).

When estimating $z_{orig}$ from observations, we follow the same methodology except that in absence of measurements for $q_f$ and $\delta D_f$ we assume a constant $\alpha_{eff} = 1.07$ based on LMDZ simulation, and that $\alpha_{eq}$, $\alpha_K$ , $\delta D_{oce}$, $h_0$ and $\delta D_0$ come from surface observations.

Note that $r_{orig}$ and $z_{orig}$ are not direct diagnostics from the simulation, but rather a-posteriori estimates to match the simulated $\delta D_0$. Therefore, if assumptions underlying Eq. (9) are violated, then the estimate of $r_{orig}$, and subsequently $z_{orig}$,

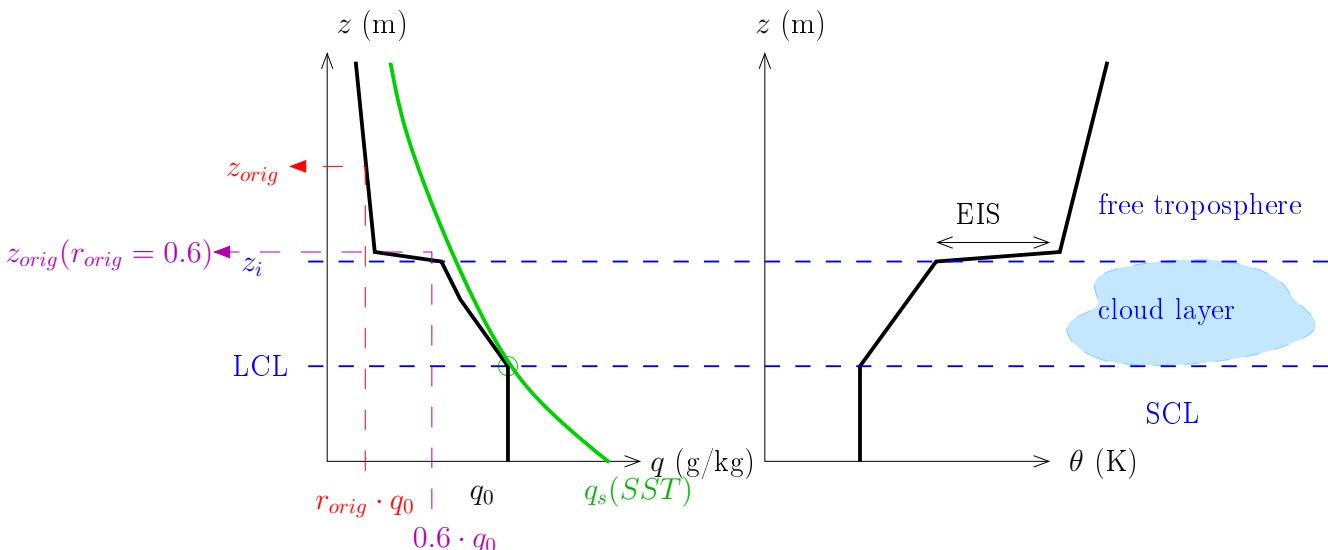

**Figure 5.** Schematics illustrating the typical structure of tropical marine boundary layers. The sub-cloud layer (SCL) extends from the surface to the lifting condensation level (LCL) and the cloud layer extends from the LCL to the inversion ($z_i$). EIS stands for the estimated inversion strength. Left: shape of the vertical profile in $q$ (black) and $q_s$ (green). Right: shape of the vertical profile in potential temperature $\theta$, inspired by Wood and Bretherton (2006). The LCL, $z_{orig}$ , $z_{orig,r_{orig}=0.6}$ and $z_i$ altitudes defined in section 3.4 are indicated.

will be biased. The estimate of $r_{orig}$ encapsulates the effect of mixing processes, but also all other processes that have been neglected in our theoretical framework, such as temporal variations in SCL depth, $q_0$ or $\delta D_0$ or vertical variations of $q_0$ or $\delta D_0$ within the SCL.

### 3.4 Boundary layer structure diagnostics

Figure 5 illustrates the structure of a typical tropical marine boundary layer covered by stratocumulus or cumulus clouds (Betts and Ridgway, 1989; Wood, 2012; Wood and Bretherton, 2004; Neggers et al., 2006; Stevens, 2006). The cloud base corresponds to the lifting condensation level (LCL). Below is the well-mixed SCL. Above is the cloud layer, topped by a temperature inversion. Above the inversion is the FT.

The LCL is calculated as the altitude at which the specific humidity near the surface equals the specific humidity at saturation
of a parcel that is lifted following a dry adiabat (Fig. 5).

The temperature inversion is an abrupt increase in temperature that caps the boundary layer. Therefore, a method to automatically estimate its altitude $z_i$ is to detect a maximum in the vertical gradient of potential temperature (Stull, 1988; Oke, 1988; Sorbjan, 1989; Garratt, 1994; Siebert et al., 2000). This method is sensitive to the resolution of vertical profiles (Siebert et al., 2000; Seidel et al., 2010). Therefore, we adapted this method in order to yield $z_i$ values that best agree with what we would es-
timate from visual inspection of individual temperature profiles. In LMDZ, we calculate $z_i$ as the first level at which the vertical

potential temperature gradient exceeds 3 times the moist-adiabatic lapse rate. In observations, we calculate $z_i$ as the first level at which the vertical potential temperature gradient exceeds 5 times the moist-adiabatic lapse rate, because radio-soundings are noisier than simulated profiles.

Finally, we calculate $z_{orig}(r_{orig} = 0.6)$, which is the $z_{orig}$ altitude if $r_{orig}$ is set to 0.6. This usually coincides with the altitude of strong humidity decrease near the inversion (Fig. 5).

## 3.5 Averages and composites

All calculations are done on daily values for LMDZ, and on 15-minute values for observations.

For LMDZ, when analyzing spatial and seasonal variability, seasonal averages are calculated at each grid box over tropical oceans by averaging all days of all years that belong to each season. Seasons are defined as boreal winter (December-January-February), spring (March-April-May), summer (June-July-August) and fall (September-October-November). For illustration purpose, all maps are plotted for boreal winter. Standard deviations are also calculated among all days of all years for each season.

The type of clouds and mixing processes depends strongly on the large-scale velocity at 500 hPa ($\omega_{500}$, map shown in Fig. 6a), with shallow clouds in subsiding regions and deeper clouds in ascending regions (Fig. 1). Therefore, it is convenient to plot variables as composites as a function of $\omega_{500}$ (Bony et al., 2004). To make such plots, we divide the $\omega_{500}$ range from -30 to 50 hPa/d into intervals of 5 hPa/d. In each given interval, we average all seasonal-mean values at all locations over tropical oceans for which seasonal-mean $\omega_{500}$ belongs to this interval (e.g. Fig. 8a will be an example). Note that such composites are done on seasonal-mean $\omega_{500}$ because cloud processes and their associated diabatic heating are tied to the large-scale circulation through energetic constrains (Yanai et al., 1973; Emanuel et al., 1994) that are best valid at longer time scales, otherwise, the energy storage term may become significant (e.g. Masunaga and Sumi, 2017). This is why $\omega_{500}$ is generally averaged over a month or longer (e.g. Bony et al., 1997; Williams et al., 2003; Bony et al., 2004; Wyant et al., 2006; Bony et al., 2013). In addition, we primarily focus on understanding the seasonal and spatial distribution of $\delta D_0$.

The cloud cover strongly correlates with the inversion strength, which can be quantified by the Estimated Inversion Strength (EIS, Wood and Bretherton, 2006) ( map shown in Fig. 6b) as a measure of inversion strength. We thus also plot variables as composites as a function of EIS. To make such plots, we divide the EIS range from -1 K to 9 K into intervals of 0.5K. In each given interval, we average all seasonal-mean values at all locations over tropical oceans for which seasonal-mean EIS belongs to this interval (e.g. Fig. 8b will be an example). Using seasonal-mean values is consistent with Wood and Bretherton (2006) and with the better link at longer time scales between cloud processes and the large-scale dynamical regime.

## 3.6 Decomposition method for $\delta D_0$

To understand what controls the $\delta D_0$ spatio-temporal variations, $\delta D_0$ is decomposed into 4 contributions based on Eq. (8). First, we define $r_{orig,bas} = 0.3$, $\alpha_{eff,bas} = 1.09$ , $SST_{bas}$=25 °C, $h_{0,bas} = 0.7$, $\phi_{bas} = 0$, $\eta_{bas} = 0$, $\beta_{bas} = 1$ and $\alpha_{evap,bas} = 1$ as a basic state. We call $\delta D_{0,func}(r_{orig}, \alpha_{eff}, SST, h_0, \phi, \beta, \eta, \alpha_{evap})$ the function giving $\delta D_0$ as a function of $r_{orig}$, $\alpha_{eff}$, SST, $h_0$, $\phi$, $\beta$, $\eta$ and $\alpha_{evap}$ following Eq. (8), and $\delta D_{0,bas} = \delta D_{0,func}(r_{orig,bas}, \alpha_{eff,bas}, SST_{bas}, h_{0,bas}, \phi_{bas}, \beta_{bas}, \eta_{bas}, \alpha_{evap,bas})$.

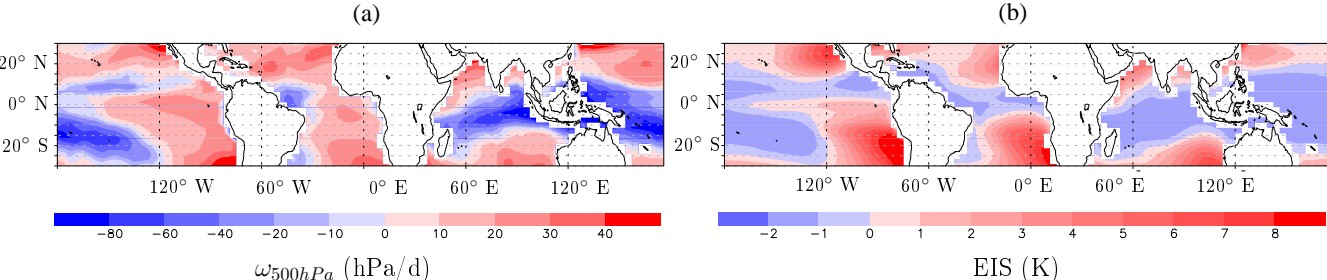

**Figure 6.** Maps of winter-mean $\omega_{500}$ (a) and EIS (b) simulated by LMDZ.

The relative contribution of $r_{orig}$ to $\delta D_0$ is estimated as $\delta D_{0,func}(r_{orig}, \alpha_{eff,bas}, SST_{bas}, h_{0,bas}, \phi_{bas}, \beta_{bas}, \eta_{bas}, \alpha_{evap,bas}) - \delta D_{0,bas}$. Similarly, the contributions of $\alpha_{eff}$, SST, $h_0$, $\phi$ and $\eta$ are to $\delta D_0$ are estimated as detailed in Table 1. All the contributions have the same units as $\delta D_0$ (‰). The sum of these components yields a quantity that is very close to the simulated $\delta D_0$, which confirms the validity of this linear decomposition. These components and their sum can be plotted as maps: Fig. 7 provides an example.

The relative contributions of each of these components to the $\delta D$ variability are quantified by performing a linear regression of each of the components as a function of $\delta D_0$. If the correlation coefficient is significant for a given factor, then the slope quantifies the contribution of this factor to the variability of $\delta D_0$. The sum of all contributions may not always be 1 due to non-linearity. Such a method has already been applied in previous studies (e.g. Risi et al., 2010b; Oueslati et al., 2016). The contributions to the seasonal-spatial variability of $\delta D_0$ can be quantified by performing the regression among all locations and seasons. The contributions to the daily variability of $\delta D_0$ can be quantified by performing the regression among all days of a given season at a given location.

### 3.7 Decomposition method for $r_{orig}$

To understand what controls $r_{orig}$, a similar method as for the decomposition of $\delta D_0$ can be applied. We can write $r_{orig}$ as:

$$r_{orig} = \frac{h(z_{orig}) \cdot q_s(\bar{T}(z_{orig}) + \delta T(z_{orig}), P(z_{orig}))}{q_0} \tag{11}$$

where $\bar{T}(z_{orig}) + \delta T(z_{orig}) = T(z_{orig})$ is the temperature at altitude $z_{orig}$, $\bar{T}$ is the tropical-ocean-mean temperature profiles, $h(z_{orig})$ and $P(z_{orig})$ are the relative humidity and pressure at $z_{orig}$, and $\delta T(z_{orig})$ is the temperature perturbation compared to $\bar{T}$. Therefore, the variability of $r_{orig}$ is decomposed into the effect of 4 factors: $q_0$, $z_{orig}$, $h(z_{orig})$ and $\delta T(z_{orig})$. In practice, $r_{orig}$ and $z_{orig}$ are calculated following section 3.3, then Eq. (11) is applied.

| Contribution | Calculation | Physical meaning |
|---|---|---|
| $r_{orig}$ | $\delta D_{0,func}(r_{orig}, \alpha_{eff,bas}, SST_{bas}, h_{0,bas}, \phi_{bas}, \beta_{bas}, \eta_{bas}, \alpha_{evap,bas}) - \delta D_{0,bas}$ | Altitude from which the air originates |
| $\alpha_{eff}$ | $\delta D_{0,func}(r_{orig,bas}, \alpha_{eff}, SST_{bas}, h_{0,bas}, \phi_{bas}, \beta_{bas}, \eta_{bas}, \alpha_{evap,bas}) - \delta D_{0,bas}$ | Steepness of the $\delta D$ vertical gradient in the FT |
| SST | $\delta D_{0,func}(r_{orig,bas}, \alpha_{eff,bas}, SST, h_{0,bas}, \phi_{bas}, \beta_{bas}, \eta_{bas}, \alpha_{evap,basic}) - \delta D_{0,bas}$ | SST |
| $h_0$ | $\delta D_{0,func}(r_{orig,bas}, \alpha_{eff,bas}, SST_{bas}, h_0, \phi_{bas}, \beta_{bas}, \eta_{bas}, \alpha_{evap,bas}) - \delta D_{0,bas}$ | $h_0$ |
| $\phi$ | $\delta D_{0,func}(r_{orig,bas}, \alpha_{eff,bas}, SST_{bas}, h_{0,bas}, \phi, \beta, \eta_{bas}, \alpha_{evap,bas}) - \delta D_{0,bas}$ | Horizontal advection through horizontal $\delta D$ gradients |
| $\eta$ | $\delta D_{0,func}(r_{orig,bas}, \alpha_{eff,bas}, SST_{bas}, h_{0,bas}, \phi_{bas}, \beta_{bas}, \eta, \alpha_{evap}) - \delta D_{0,bas}$ | Rain evaporation in the SCL |

**Table 1.** Equations to calculate the relative contributions of $r_{orig}$, $\alpha_{eff}$, SST, $h_0$, $\phi$ and $\eta$ to $\delta D_0$, and the physical meaning of these contributions.

## 4   Results from LMDZ

### 4.1   Decomposition of $\delta D_0$ variability

The spatial variations of $\delta D_0$ simulated by LMDZ (Fig. 7a) are characterized by depleted values near mid-latitudes and in dry subsiding regions (e.g. off the coast of Peru and over other regions of oceanic upwelling) and regions of atmospheric deep
convection (e.g. Maritime Continent). Consistently, $\delta D_0$ values exhibit a maximum for weakly ascending or subsiding regions: $\delta D_0$ decreases with increasing vertical velocity of both signs (Fig. 8a black); $\delta D_0$ decreases as EIS increases reflecting more stable, subsiding conditions (Fig. 8b black). This pattern is consistent with previous studies (e.g. Good et al., 2015). For the first time, we propose a theoretical framework to interpret this pattern, decomposing it into 6 contributions: $r_{orig}$, $\alpha_{eff}$, SST, $h_0$, rain evaporation and horizontal advection effects (section 3.6). We check that the reconstructed $\delta D_0$ from the sum of its 4
contributions is very similar to the simulated $\delta D_0$ (Fig. 7b, 8 dashed black).

In ascending regions, the main contribution explaining the more depleted $\delta D_0$ in deep convective regions is that of $\alpha_{eff}$ (Fig. 7d, 8a red). $\alpha_{eff}$ is higher in more ascending regions (Fig. D1d). This means that the main factor depleting $\delta D_0$ in deep convective regions is the fact that the mid-troposphere is more depleted. This leads to a steeper gradient (higher $\alpha_{eff}$), and thus a more efficient depletion by vertical mixing. This is consistent with deep convection depleting the water vapor most efficiently
in the mid-troposphere (Bony et al., 2008). The second main contribution is that associated with $r_{orig}$ (Fig. 7c, 8a green). $r_{orig}$ is larger in deep convective regions (as explained in section 4.2).

In subsidence regions, the main factor explaining the more depleted $\delta D_0$ as subsidence is stronger, or as EIS increases, is the cold SST (Fig. 7e, 8a pink), leading to larger $\alpha_{eq}$, and to a lesser extent the dry $h_0$ (Fig. 7f, 8a purple). The contribution of

$r_{orig}$ is also a significant contribution to the depletion of $\delta D_0$ in the cold upwelling regions, for example off Peru or Namibia (Fig. 7c). The shallower boundary layer there are associated with higher $r_{orig}$.

The contribution of rain evaporation on $\delta D_0$ is minor compared to other contributions, except in the deepest convective regions (Fig. 7g). Rain evaporation has a slightly depleting effect in regions of strong deep convection and a slightly enriching effect in regions of moderate deep convection. When the fraction of raindrops that evaporate is small, isotopic fractionation favors evaporation of the lighter isotopologues. Therefore in convective, moist regions, rain evaporation has a depleting effect on the SCL (Worden et al., 2007). In contrast, in drier regions, rain evaporates almost totally. The evaporation flux thus has almost the same composition as the initial rain, which is more enriched than the water vapor.

The contribution of horizontal advection to $\delta D_0$ is significant only where isotopic gradients are the largest (Fig. C1h). Horizontal advection has slightly enriching in deep convective regions and depleting in coastal regions (e.g. off the coasts of California, Peru, Mauritania, Namibia, India and Australia). For example, the Saharan layer in front of the North-Western African Coast leads to a strong effect of horizontal advection (Lacour et al., 2017a).

From a quantitative point of view, we can decompose the $\delta D_0$ seasonal-spatial variations into these different effects (section 3.6). In regions of large-scale ascent, $\alpha_{eff}$ is the main factor explaining the $\delta D_0$ seasonal-spatial variations (33 %), followed rain evaporation (20 %) and $r_{orig}$ (19 %) and (Table 2). In regions of large-scale descent, SST is the main factor explaining the seasonal-spatial variations (54 %), followed by $r_{orig}$ (29 %) $h_0$ (13 %), and $\alpha_{eff}$ (10 %) (Table 2). Note that the contribution of $r_{orig}$ would be similar if we neglect rain evaporation and horizontal advection effects (Table 2).

The decomposition method can also be applied to decompose the $\delta D_0$ variability at the daily time scale at each location and for each season (Table 3). On average, in ascending regions, $r_{orig}$ is the main factor (52 %), followed by rain evaporation (48 %) and $\alpha_{eff}$ (35 %). In subsiding regions, the effect of SST is muted due to its slow variability, and $r_{orig}$ (82 %) becomes the main factor.

Overall, the results highlight the importance of $r_{orig}$ as one of the main factors controlling the spatio-temporal variability of $\delta D_0$.

## 4.2   Decomposition of $r_{orig}$ variability

Given the importance of $r_{orig}$ in controlling the $\delta D_0$ variations, we now decompose $r_{orig}$ into its 4 contributions: $q_0$, $z_{orig}$, $h_{orig}$ and $\delta T_{orig}$ (section 3.6). Spatially, $r_{orig}$ is maximum in regions of strong large-scale ascent (Fig. 10a) such as the Maritime continent (Fig. 9a), and in very stable regions (Fig. 10b) such as upwelling regions (Fig. 10a). We check that the reconstructed $r_{orig}$ from the sum of its 4 contributions is very similar to the simulated $r_{orig}$ (Fig. 9b, 10 dashed black).

In regions of strong large-scale ascent, $r_{orig}$ is larger mainly because $h_{orig}$ is larger (Fig. 9e, 10a pink). This is because the moister the FT, the higher the contribution of vapor coming from above to the vapor of the SCL, and thus the higher $r_{orig}$ and the more depleted $\delta D_0$. This mechanism through which a moister FT leads to a more depleted $\delta D_0$ is consistent with that argued in B15. $z_{orig}$ damps this effect: when convection is stronger and the FT moister, convection is also deeper, so the air originates from higher altitudes where the air is drier.

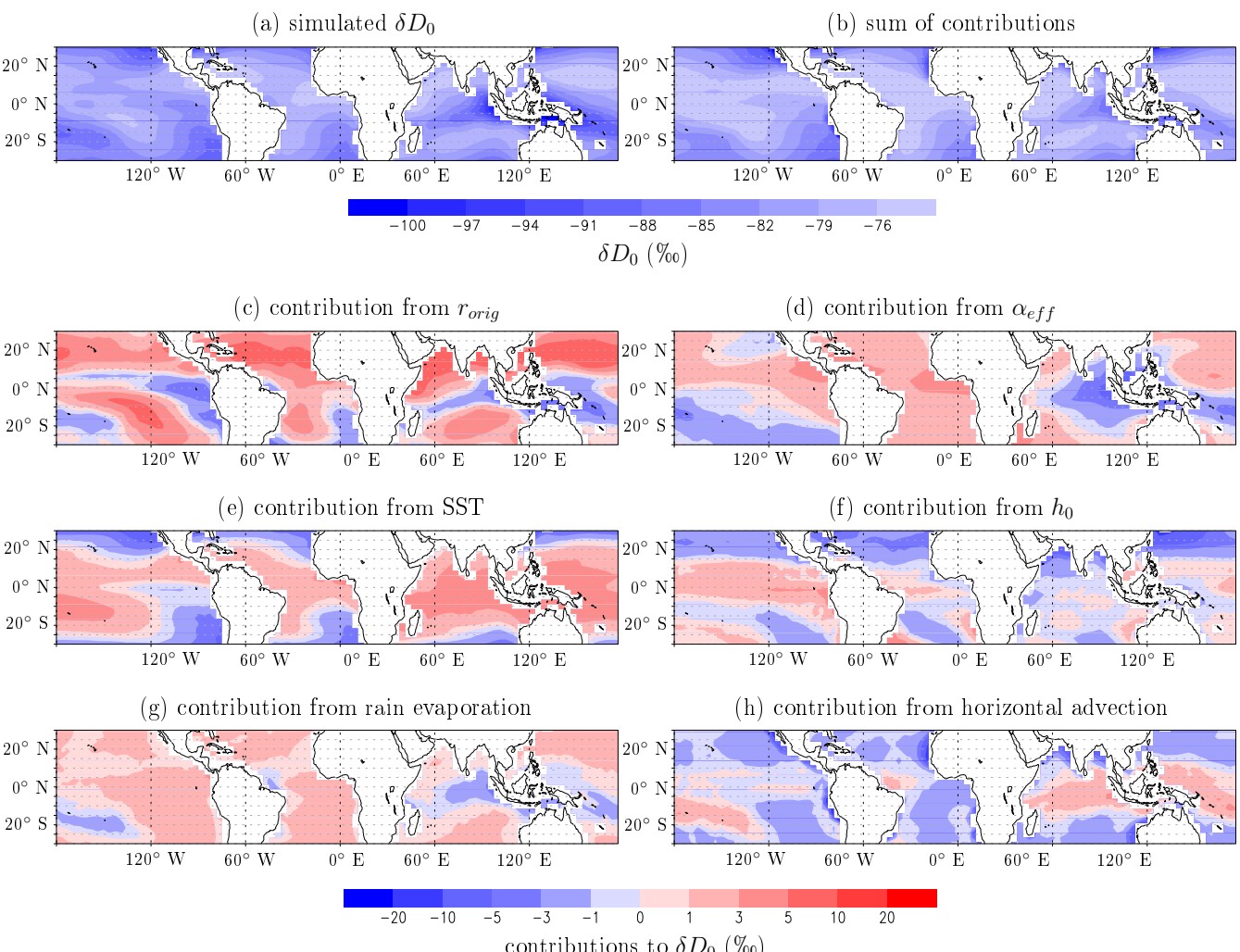

**Figure 7.** (a) Map of winter-mean $\delta D_0$ simulated by LMDZ. (b) Map of winter-mean $\delta D_0$ reconstructed as the sum of the 4 contributions. Tropical-mean $\delta D_0$ was added to compare with a on the same color scale. (c) Map of the contribution of $r_{orig}$ on winter-mean $\delta D_0$ calculated from Eq. (9) (see section 3.6). (d) Same as (b) but for $\alpha_{eff}$ varies. (e) Same as (b) but for SST. (f) Same as (b) but for $h_0$.

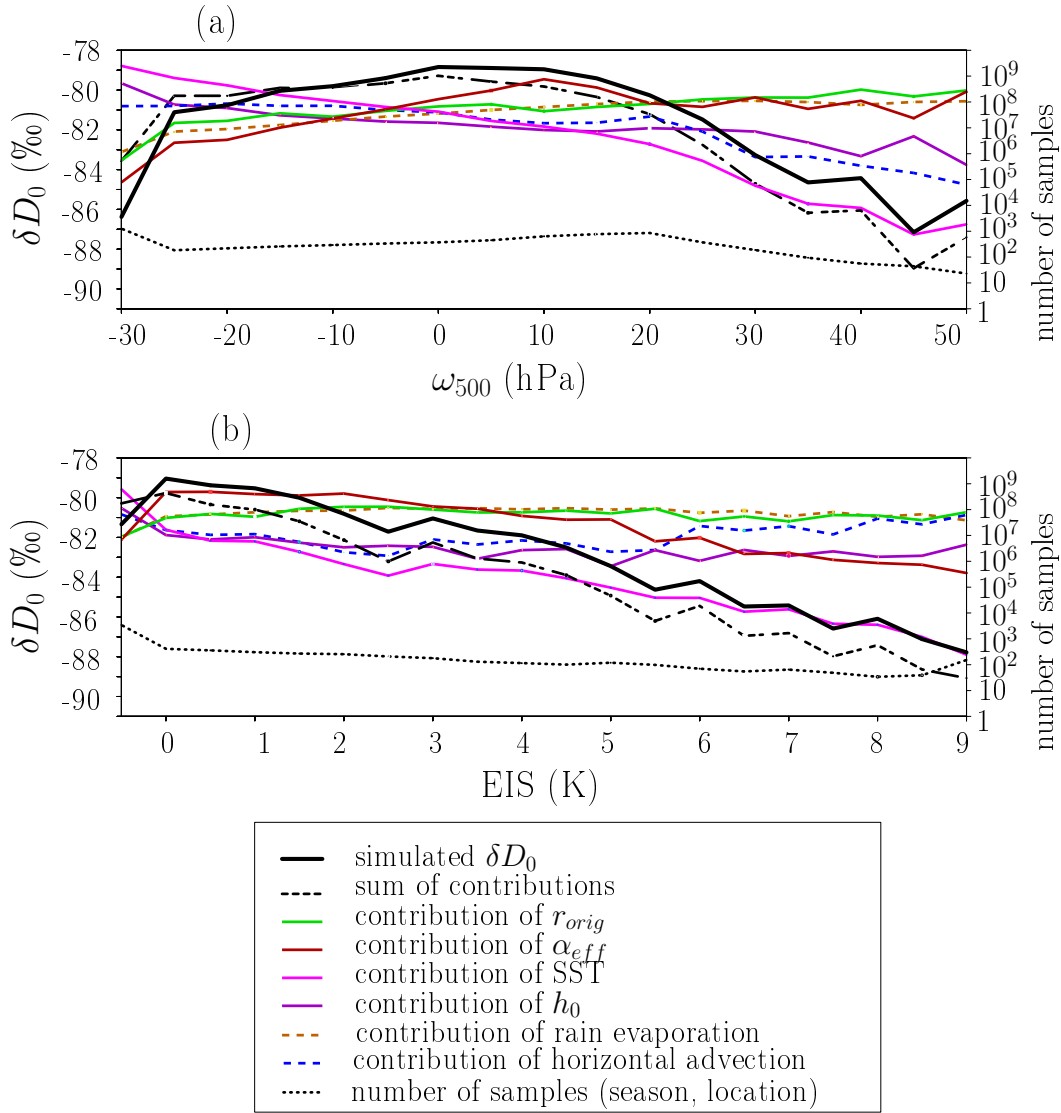

**Figure 8.** Composites as a function of $\omega_{500}$ (a) and of EIS (b) of the seasonal averages of $\delta D_0$ simulated by LMDZ over all tropical ocean locations (black). Same for the sum of the contributions (black dashed and for each individual contribution to $\delta D_0$: $r_{orig}$ varies (green), $\alpha_{eff}$ (dark red), SST (pink), $h_0$ (purple), rain evaporation (dashed brown) and horizontal advection (dashed blue). the tropical mean $\delta D_0$ was added to each contribution to plot on the same scale as simulated $\delta D_0$. The number of samples in each bin is indicated on a logarithmic scale on the right-hand-side (dotted black line).

| Regime | Ascending | | Subsiding | |
|---|---|---|---|---|
| | Correlation coefficient | Slope | Correlation coefficient | Slope |
| $r_{orig}$ | 0.59 | 0.19 | 0.52 | 0.29 |
| $\alpha_{eff}$ | 0.73 | 0.33 | 0.26 | 0.10 |
| SST | -0.23 | -0.06 | 0.89 | 0.54 |
| $h_0$ | (0.06) | (0.01) | 0.28 | 0.13 |
| Rain evaporation | 0.67 | 0.20 | -0.36 | -0.05 |
| Horizontal advection | -0.26 | -0.12 | (0.10) | (0.04) |
| $r_{orig}$ if rain evaporation and horizontal advection are neglected | 0.69 | 0.30 | 0.58 | 0.34 |

**Table 2.** Decomposition of the spatial and seasonal variation in $\delta D_0$ into its 6 contributions: effect of $r_{orig}$, $\alpha_{eff}$, SST, $h_0$, rain evaporation and horizontal advection. For each contribution, we show the correlation coefficient of the linear regression of the contribution as a function of $\delta D_0$. The analysis is done separately for ascending and subsiding regimes. All seasons and locations over tropical oceans ($30°N - 30°S$, ocean fraction>80 %, surface evaporation>0.5 mm/d) are considered. The threshold for the correlation coefficient to be statistically significant at 99 % is 0.15 or lower in all cases. We write correlation coefficient and slope values between brackets when they are not significant at 99 %.

| Regime | Ascending | | Subsiding | |
|---|---|---|---|---|
| | Correlation coefficient | Slope | Correlation coefficient | Slope |
| $r_{orig}$ | 0.46 | 0.52 | 0.42 | 0.82 |
| $\alpha_{eff}$ | 0.34 | 0.35 | 0.14 | 0.40 |
| SST | -0.12 | -0.01 | 0.25 | 0.22 |
| $h_0$ | 0.06 | 0.26 | 0.15 | 0.39 |
| Rain evaporation | 0.49 | 0.48 | 0.19 | 0.20 |
| Horizontal advection | -0.26 | -0.24 | -0.15 | -0.31 |

**Table 3.** As in Table 4 but at the daily scale. The correlation coefficients and slopes are averaged over all seasons and locations over tropical oceans ($30°N - 30°S$, ocean fraction>80 %), separately for ascending and subsiding regimes.

| Regime | Ascending | | Subsiding | |
|---|---|---|---|---|
| | Correlation coefficient | Slope | Correlation coefficient | Slope |
| $q_0$ | -0.82 | -0.33 | 0.21 | 0.12 |
| $z_{orig}$ | -0.91 | -0.67 | 0.77 | 0.41 |
| $h_{orig}$ | 0.98 | 1.82 | 0.53 | 0.96 |
| $\delta T_{orig}$ | 0.55 | 0.06 | -0.37 | -0.12 |

**Table 4.** Decomposition of the spatial-seasonal variation in $r_{orig}$ into its 4 contributions: effect of $q_0$, $z_{orig}$, $h_{orig}$ and $\delta T_{orig}$ variations. For each contribution, we show the correlation coefficient of the linear regression of the contribution as a function of $r_{orig}$. The analysis is done separately for ascending and subsiding regimes. All seasons and locations over tropical oceans ($30°N - 30°S$, ocean fraction>80 %) are considered. The threshold for the correlation coefficient to be statistically significant is 0.15 or lower in all cases. We write correlation coefficient and slope values between brackets when they are not significant at 99 %.

| Regime | Ascending | | Subsiding | |
|---|---|---|---|---|
| | Correlation coefficient | Slope | Correlation coefficient | Slope |
| $q_0$ | -0.37 | -0.04 | -0.23 | -0.07 |
| $z_{orig}$ | 0.91 | 0.39 | 0.80 | 0.39 |
| $h_{orig}$ | 0.58 | 0.78 | 0.58 | 1.18 |
| $\delta T_{orig}$ | -0.14 | 0.01 | -0.23 | -0.12 |

**Table 5.** As in Table 4 but at the daily scale. The correlation coefficients and slopes are averaged over all seasons and locations over tropical oceans ($30°N - 30°S$, ocean fraction>80 %), separately for ascending and subsiding regimes.

In very stable regions, $r_{orig}$ is larger because $q_0$ is larger (Fig. 9c, 10b green), consistent with the drier conditions in these regions of large-scale descent. Note that this effect can be seen only in most stable regions, but when considering all subsiding regions, the contribution is small (Table 2). $r_{orig}$ is larger also because $z_{orig}$ is lower in altitude (Fig. 9d, 10b red). As EIS increases, the boundary layers are shallower, the air comes from lower in altitude, $r_{orig}$ is higher and thus $\delta D_0$ is more depleted.

5 This mechanism was not considered in B15 but our decomposition shows that it is a key mechanism driving $r_{orig}$ and thus $\delta D_0$ variations in stable regions.

Quantitatively, in ascending regions, the main factor controlling the seasonal-spatial variations in $r_{orig}$ is $h_{orig}$ (182 %), dampened by $z_{orig}$ (-67 %) (Table 4). In descending regions, the main factor is also $h_{orig}$ (96 %), followed by $z_{orig}$ (41 %) (Table 4). At the daily scale, the same two factors dominate the variability of $r_{orig}$: $h_{orig}$ and $z_{orig}$ contribute to 78 % and

10 39 % of $r_{orig}$ variations in average over ascending regions, and to 118 % and 39 % in average over descending regions (Table 5).

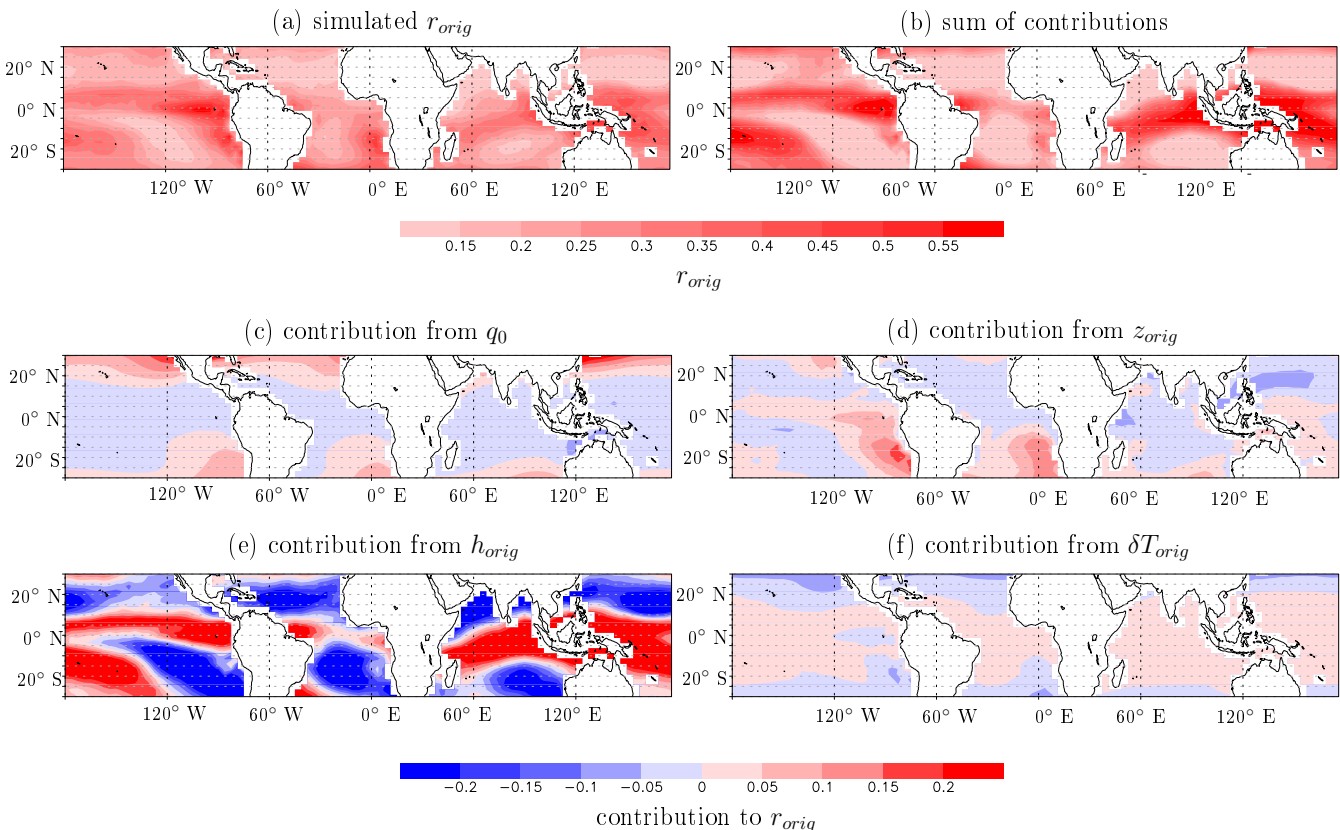

**Figure 9.** (a) Map of winter-mean $r_{orig}$ simulated by LMDZ. (b) Map of winter-mean $r_{orig}$ reconstructed as the sum of the 4 contributions. Tropical-mean $r_{orig}$ was added to compare with a with the same color scale. (c) Map of winter-mean $r_{orig}$ calculated from Eq. (11) if only $q_0$ varies (see section 3.6). (d) Same as (b) but of only $z_{orig}$ varies. (e) Same as (b) but if only $h(z_{orig})$ varies. (f) Same as (b) but if only $\delta T_{orig}$ varies.

### 4.3 Estimating altitude $z_{orig}$

Estimated altitude $z_{orig}$ is minimum in dry subsiding regions, especially in upwelling regions (Fig. 11a, Fig. 12), corresponding to regions with the strongest inversion (Fig. 11). This contributes to the depleted $\delta D_0$ in these regions.

As explained in 3.3, our estimate of $z_{orig}$ may be artificially biased due to the neglect of some processes in our theoretical framework. Ideally, to check whether $z_{orig}$ really physically represents the altitude from which the air originates, additional model experiments where water vapor from different levels are tagged (Risi et al., 2010b) would be needed. While we leave this for future work, we check whether $z_{orig}$ estimates are consistent with what we expect based on what we know about mixing processes in the marine boundary layers. We expect that in stratocumulus regions, air originates from a very shallow (a few tens of meters) layer above the inversion, whereas the mixing processes may be more diverse, and possibly deeper in the FT, as the boundary layer deepens (Fig. 1).

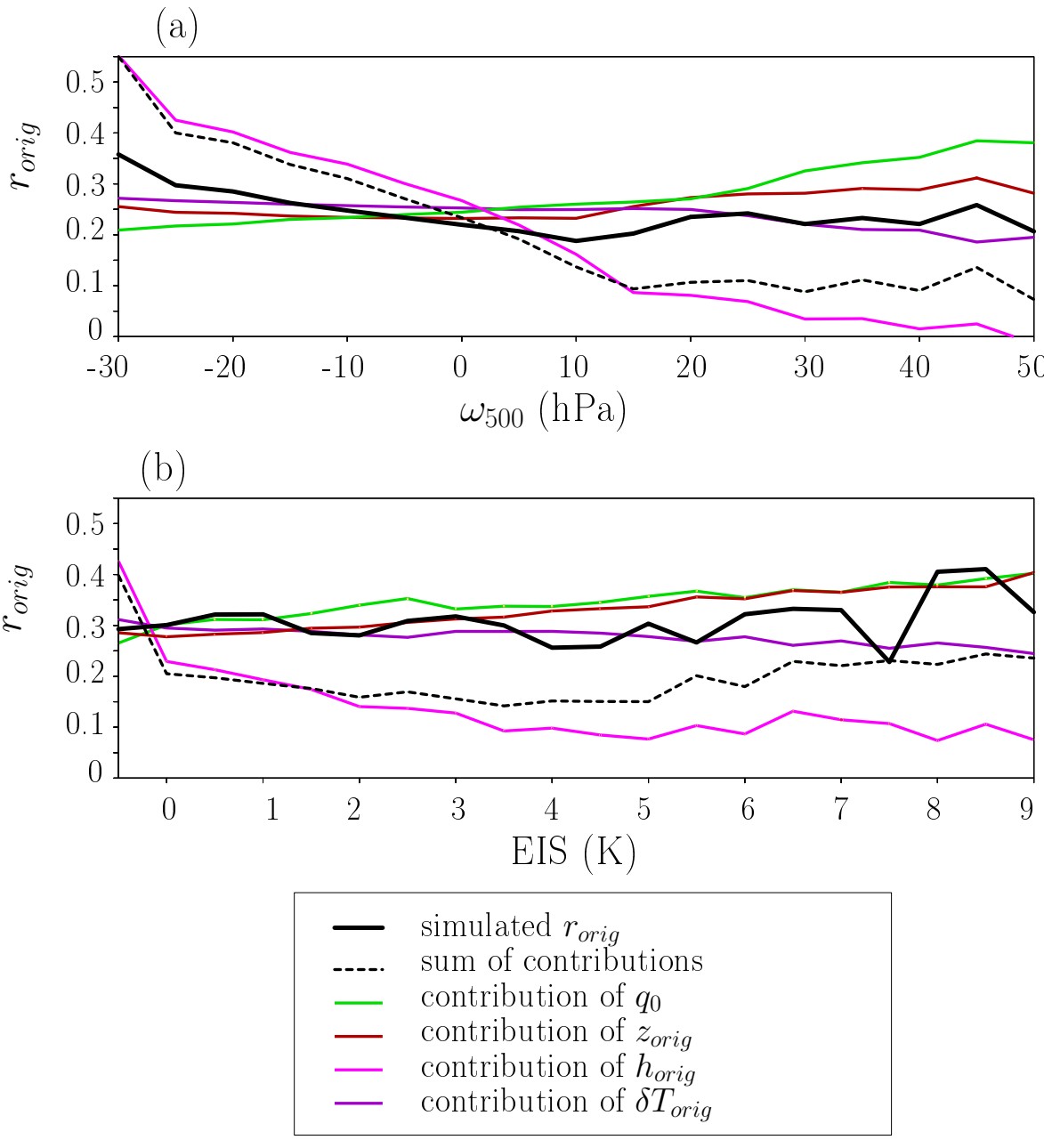

**Figure 10.** Composites as a function of $\omega_{500}$ (a) and of EIS (b) of the seasonal averages of $r_{orig}$ simulated by LMDZ over all tropical ocean locations (black). Same for the sum of the contributions (black dashed) and for each individual contribution to $r_{orig}$ : $q_0$ varies (green), $z_{orig}$ (dark red), $h(z_{orig})$ (pink) and $\delta T_{orig}$ (purple). The number of samples in each bin is indicated on a logarithmic scale on the right-hand-side as bars.

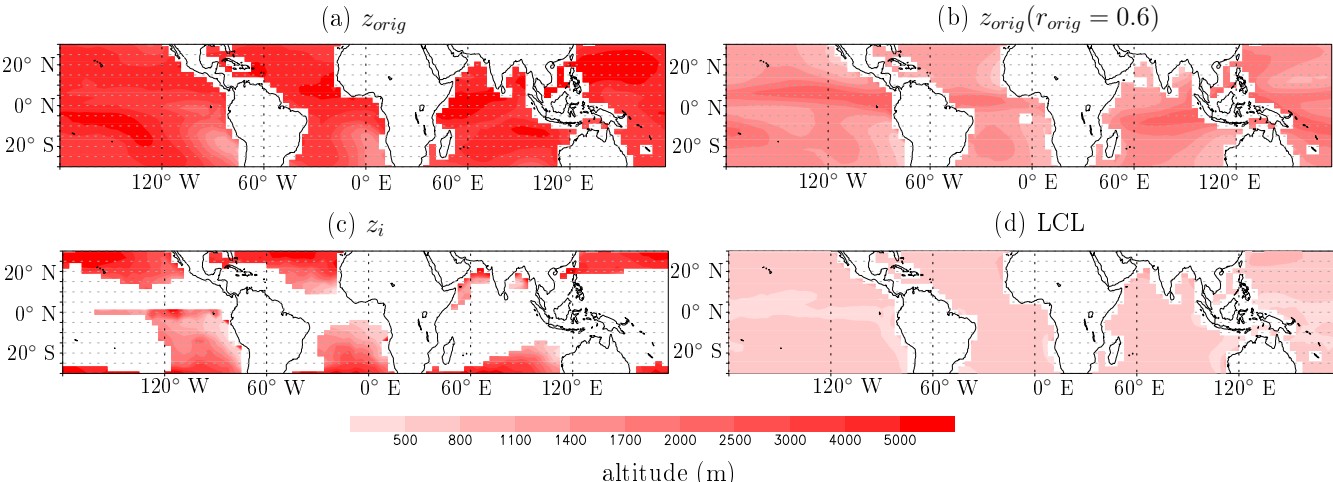

**Figure 11.** (a) Map of winter-mean $z_{orig}$ estimated from $\delta D_0$ simulated by LMDZ. (b) Same as (a) but $z_{orig}$ that we would estimate if $r_{orig}$ was constant set to 0.6, $z_{orig}(r_{orig} = 0.6)$. (c). Same as (a) but for $z_i$ simulated from LMDZ. Only days when EIS>2 K are considered, otherwise $z_i$ is difficult to estimate. (d) Same as (a) but for LCL simulated by LMDZ.

To check whether estimated $z_{orig}$ is consistent with this picture, we compare $z_{orig}$ to $z_{orig}(r_{orig} = 0.6)$ ($z_{orig}$ that we would estimate is $r_{orig}$ was set constant to 0.6) and $z_i$ (section 3.4), which are measures of the altitude of the humidity drop and temperature inversion respectively. As expected from Fig. 1, they are minimum in dry upwelling regions, intermediate in trade-wind regions, and maximum values in convective regions (Fig. 11c-d, 12 green, blue). Therefore, the low $z_{orig}$ in

upwelling regions reflects the low $z_i$. Consistently, in subsiding regions, $z_{orig}$ correlates well with $z_{orig,r_{orig}=0.6}$ (correlation coefficient of 0.52, statistically significant beyond 99 %). If we focus on very stable regions only (EIS>7 K), $z_{orig}$ correlates well with both $z_{orig}(r_{orig} = 0.6)$ and $z_i$ (correlation coefficient of 0.58 and 0.52 respectively, statistically significant beyond 99 %). The altitude $z_{orig}$ is a few meters above the inversion in stratocumulus regions, and up to 1km above the inversion in cumulus and deep convective regions (Fig. 12), consistent with our expectations from Fig. 1. This lends support to the fact that

at least in subsiding regions, our isotope-based $z_{orig}$ estimate effectively reflects the origin of air coming from above.

In ascending regions, in contrast, $z_{orig}$ does not correlate significantly with $z_{orig}(r_{orig} = 0.6)$ or $z_i$. This may indicate either that our $z_{orig}$ estimate is biased by neglected processes such as rain evaporation, or that in deep convective regions, the origin of FT air into the SCL is very diverse due to the variety of mixing processes (1).

## 5   Results from observations

To check whether our results obtained with LMDZ are realistic, we apply our methods to the measurements gathered during the STRASSE campaign. For simplicity and in absence of all necessary measurements, here we neglect the effects of rain evaporation and horizontal advection.

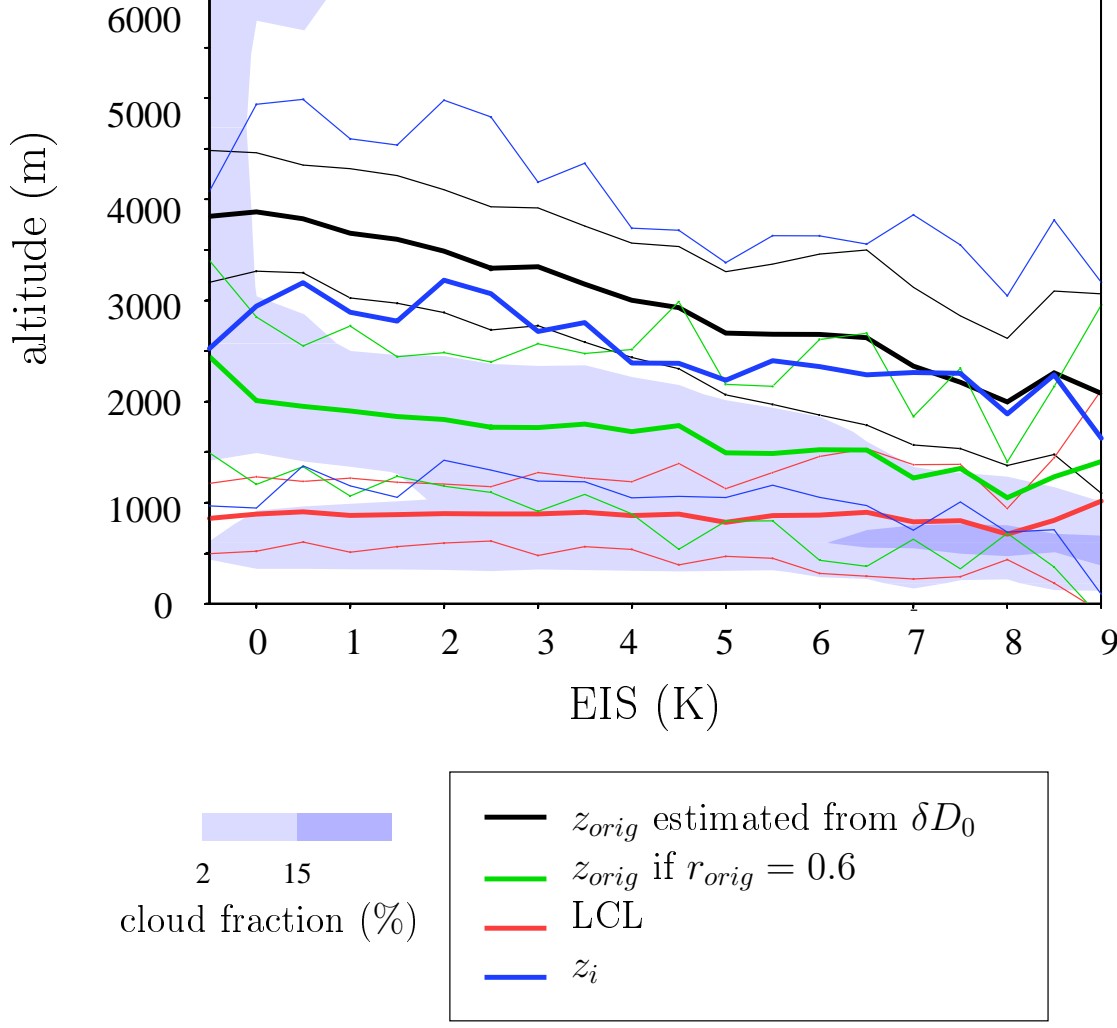

**Figure 12.** Composites as a function of EIS of seasonal-mean of $z_{orig}$ (black), $z_{orig}(r_{orig} = 0.6)$ (green), $z_i$ (blue) and LCL (red). The composite profiles of cloud cover are also shown, showing deep clouds when EIS is close to 0 and shallowest clouds when EIS is largest.

| Contributions to $\delta D_0$ | Correlation coefficient | Slope |
|---|---|---|
| $r_{orig}$ | 0.77 | 0.58 |
| SST | 0.57 | 0.16 |
| $h_0$ | 0.40 | 0.48 |

**Table 6.** Same as Table 2 but for the STRASSE observations. Linear regressions are calculated among 1977 data points.

Throughout the cruise, $\delta D_0$ shows a large variability, ranging from around -75 ‰ in quiescent conditions to -120 ‰ during the two convective conditions (Benetti et al., 2014) (Fig. 13a red). Variability in $r_{orig}$ is the major factor contributing to this variability (58 %) (Fig. 13a green, Table 6). This crucial importance of mixing processes is consistent with B15.

During the two convective events, the estimated $r_{orig}$ saturates at 1 (Fig. 13b). This proves that $r_{orig}$ estimated in these
conditions is biased high because it encapsulates the effect of neglected processes, i.e. depletion by rain evaporation. Equation (9) is not valid in this case. In addition, at the scale of a few hours, the steady-state assumptions may be violated. Rain evaporation may strongly deplete the SCL before surface evaporation has the time to play its dampening role, hence the possibility to reach very low $\delta D_0$ that cannot be predicted even when considering rain evaporation (Appendix B).

During the rest of the cruise, the main factors controlling the $r_{orig}$ variability are $z_{orig}$ (90 %) and $h_{orig}$ (70 %). The
importance of FT humidity in controlling $r_{orig}$ was already highlighted in B15. However, in their paper, the variability in $z_{orig}$ was neglected, whereas it appears here as the main factor.

Through September, the cruise goes from a shallow boundary layer in early September to deeper boundary layers with higher inversions, before reaching the convective conditions (Fig. 13c). Consistently with this deepening boundary layer, the air originates from increasingly higher altitudes. Remarkably, there are 6 days when $z_{orig}$ coincides with $z_i$ with a root means
square error of 31 ‰ and correlation coefficient of 0.996 (Fig. 13c). This indicates that the air exactly comes from the inversion layer. When recalling that $z_{orig}$ and $z_i$ are estimated from completely independent observations, the coincidence is remarkable and lends support to the fact that on these days, our $z_{orig}$ estimate is physical. However, there remains 9 days when $z_{orig}$ is much higher than $z_i$. This may reflect more penetrative downdrafts as we approach deeper convective regimes. But it may also be an artifact of our neglect of horizontal advection. For example, on these days which are characterized by lower $h_0$,
neglecting the advection of enriched water vapor from nearby regions with higher $h_0$ could be mis-interpreted as lower $r_{orig}$ and thus higher $z_{orig}$.

## 6 Discussion: what can we learn from water isotopes on mixing processes?

We have shown in the previous section that one of the main factors controlling $\delta D_0$ at the seasonal-spatial and daily scale are the proportion of the water vapor in the SCL that originates from above ($r_{orig}$), and that one of the main factor controlling
$r_{orig}$ is the altitude from which the air originates ($z_{orig}$). In turn, could we use water vapor isotopic measurements to constrain $z_{orig}$? This would open the door to discriminating between different mixing processes at play (Fig. 1). Since mixing processes

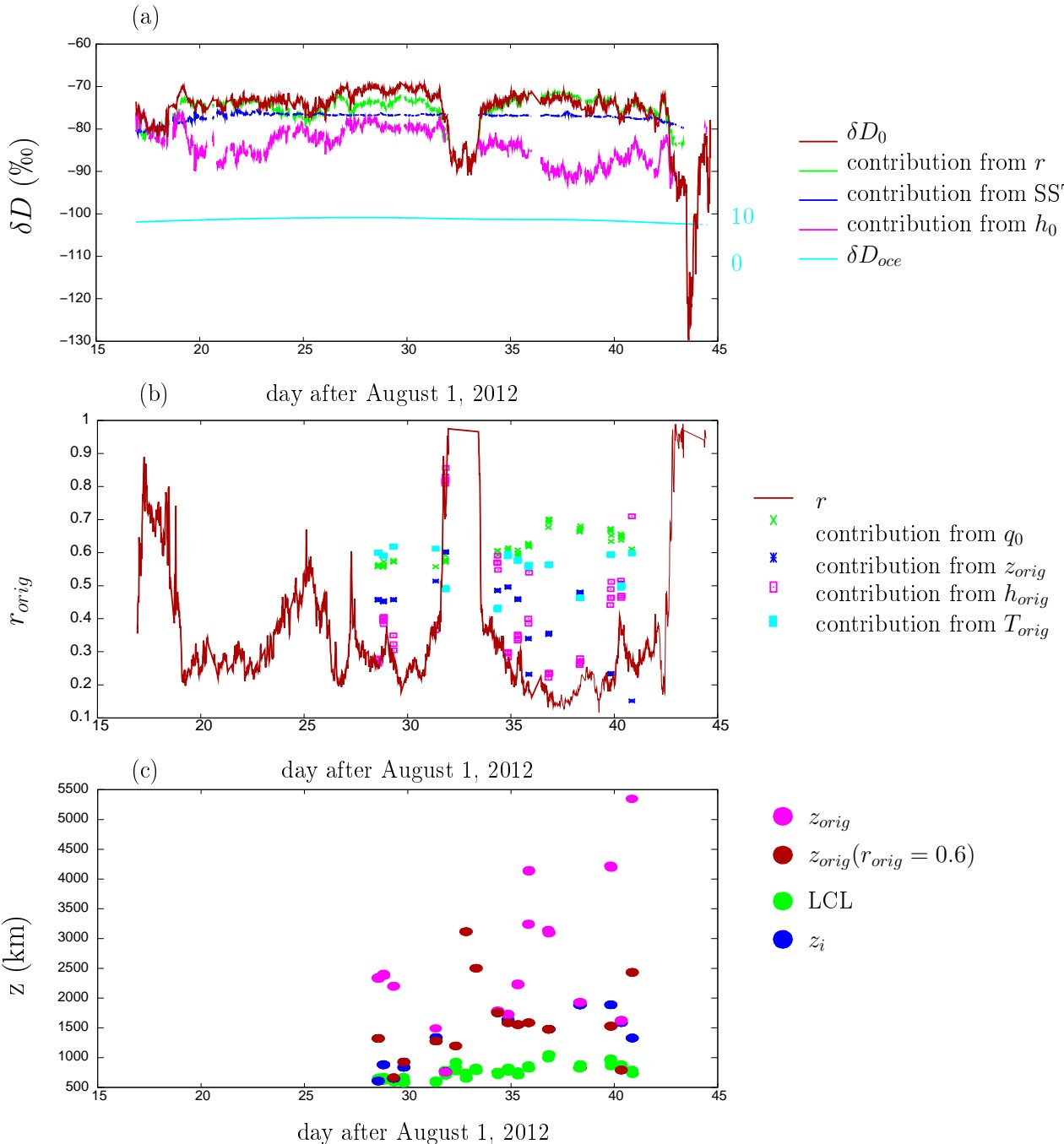

**Figure 13.** (a) Time series of $\delta D_0$ observed during the STRASSE cruise, together with its 4 contributions. The $\delta D$ of the surface ocean water is also plotted with the scale on the right. (b) Time series of $r_{orig}$ estimated from observations during the STRASSE campaign, together with its 4 contributions. (c) Times series of $z_{orig}$, $z_{orig}(r_{orig} = 0.6)$ , LCL and $z_i$ estimated from the STRASSE observations.

| Contributions to $r_{orig}$ | Correlation coefficient | Slope |
|---|---|---|
| $q_0$ | -0.46 | -1.49 |
| $z_{orig}$ | 0.66 | 0.90 |
| $h_{orig}$ | 0.81 | 0.70 |
| $\delta T_{orig}$ | -0.36 | -0.91 |

**Table 7.** Same as Table 4 but for the STRASSE observations. Linear regressions are calculated among 55 data points, so that correlation coefficients above 0.35 are statistically significant at 99 %.

are crucial to determine the sensitivity of cloud fraction to SST (Sherwood et al., 2014; Bretherton, 2015; Vial et al., 2016), such a prospect would allow us to improve our knowledge of cloud feedbacks, and hence of climate sensitivity.

With this in mind, we assess the errors associated with $z_{orig}$ estimates from $\delta D_0$ measurements, and discuss whether they are small enough for $z_{orig}$ estimates to be useful. In stratocumulus clouds where the air is believed to originate from the first few tens of meters above cloud top (Faloona et al., 2005; Mellado, 2017), $z_{orig}$ estimates are not useful if the errors are larger than a few tens of meters, e.g. 20 m. In cumulus clouds where mixing processes are more diverse and possibly deeper (Fig. 1), $z_{orig}$ estimates may be useful if errors are of the order of 80 m.

Let's assume that we have a field campaign where we measure $\delta D_0$, surface meteorological variables, temperature and humidity profiles (e.g. radio-soundings), and a few $\delta D$ profiles (e.g. by aircraft). This is what we can expect for example from the future EUREC4A (Elucidating the role of clouds-circulation coupling in climate) campaign to study trade-wind cumulus clouds (Bony et al., 2017). Below we quantify the effects of five sources of uncertainty on $z_{orig}$ estimates.

### 6.1 Measurement errors

The first source of uncertainty are measurement errors. We re-calculate $z_{orig}$ assuming an error of 0.4 ‰ on $\delta D_0$ (typical of what we can measure with in-situ laser instruments: Aemisegger et al., 2012; Benetti et al., 2014) and 1 ‰ on $\delta D_f$ (larger errors due to lower humidity and the increased complexity of measurements in altitude). The averaged errors on $z_{orig}$ and their standard deviations are plotted as a function of EIS in Fig. 14a. Whereas errors on $\delta D_f$ lead to errors on $z_{orig}$ of the order of 20 m (Fig. 14a, green), errors on $\delta D_0$ lead to errors on $z_{orig}$ of the order of 80 m (Fig. 14a, red). Yet in stratocumulus, none expects the air to originate from a higher altitude than 80 m above the inversion. Therefore, $\delta D_0$ measurements would need to be more accurate than usual to be useful in strato-cumulus regions, i.e. 0.1 ‰ to yield a 20 m precision on $z_{orig}$. In trade-wind cumulus regions, the precision of 0.4 ‰ is enough for $z_{orig}$ to be useful.

### 6.2 Neglecting rain evaporation

The second source of uncertainty is associated with neglecting rain evaporation. This effect can be quantified in a model, but it is very difficult to quantify in nature because it is complicated and uncertain to measure $\eta$ (Rosenfeld and Mintz, 1988), and it is even more complicated to measure or predict $\alpha_{evap}$. Rain evaporation can have a depleting or enriching effect depending

on microphysical details that are too complex to be addressed here (Graf et al., 2019). Neglecting rain evaporation leads to an error of the order of 500 m in regions of low EIS and 250 m in regions of strong EIS (Fig. 14b, brown). In regions of strato-cumulus regions, rain evaporation is a significant source of error in spite of the relatively small amount of precipitation available to evaporate. This is because total evaporation of the rain efficiently enriches the SCL, and easily modifies $\delta D_0$ by

more than the 0.1 ‰ targeted precision explained above. However, it is possible that LMDZ overestimates this source of error in trade-wind cumulus and strato-cumulus regions. LMDZ is one of the GCMs producing the strongest rain in srato-cumulus regions (Zhang et al., 2013), and GCMs are known to trigger convection to often in trade-wind cumulus regions (Nuijens et al., 2015a, b).

### 6.3    Neglecting horizontal advection

The third source of uncertainty is associated with horizontal advection. In nature, $\phi$ can be estimated from meteorological analyzes and $\beta$ can be estimated from near-surface isotopic measurements at several locations (e.g. sounding arrays during typical field campaigns). In absence of these additional measurements, neglecting this effect leads to an error of the order of 800 m (Fig. 14b, purple). This limits the usefulness of $z_{orig}$ estimates for all cloud regimes.

### 6.4    Daily variability in the steepness of $\delta D$ profiles

The fourth source of uncertainty arises from the daily variability in $\alpha_{eff}$ (Appendix D2). Estimating $\alpha_{eff}$ requires to measure $\delta D_f$ at 500 hPa. Satellite measurements are available but are affected by random errors that are too large for our application (Worden et al., 2011, 2012; Lacour et al., 2015). Precise in-situ measurements of water vapor $\delta D$ in altitude are costly and difficult (Sodemann et al., 2017).

    Let's assume that we have only one $\delta D_f$ value that represents the seasonal-average at a given location. To estimate the

resulting error on $z_{orig}$, we re-estimate $z_{orig}$ every day and at each location using $\alpha_{eff}^- + \sigma_{\alpha_{eff}}$ and $\alpha_{eff}^- - \sigma_{\alpha_{eff}}$. The error on $z_{orig}$ is calculated as $\left(z_{orig}(\alpha_{eff}^- - \sigma_{\alpha_{eff}}) - z_{orig}(\alpha_{eff}^- + \sigma_{\alpha_{eff}})\right)/2$. The averaged error and its standard deviation is plotted as a function of EIS in Fig. 14c (black). It is of the order of 400m, and rarely below 200m. If we attempt to estimate $\alpha_{eff}$ as the fractionation coefficient as a function of local temperature, errors would be even more dissuasive (Fig. 14c, blue).

    Therefore, estimating $z_{orig}$ from daily $\delta D_0$ measurements cannot be useful unless we measure $\delta D_f$ on a daily basis as well.

Practically, we could imagine measuring FT properties ($\delta D_f$) at the top of a mountain while we measure $\delta D_0$ at the sea level (e.g. on Islands such as Hawaii or La Réunion: Galewsky et al., 2007; Bailey et al., 2013; Guilpart et al., 2017) .

### 6.5    Rayleigh assumption for the shape of $\delta D$ profiles

Finally, as a fifth source of uncertainty comes the assumption that the $\delta D$ profile follows a Rayleigh distillation line (section 2.2). However, both in LMDZ (Appendix D1) and nature (Sodemann et al., 2017), $\delta D$ profiles are usually intermediate between

Rayleigh and mixing lines. The precision of our $z_{orig}$ estimate is maximum in the Rayleigh distillation case.

When trying to find a numerical solution for $z_{orig}$ directly from Eq. (6), a solution can be found only in 0.1 % of cases. This is because simulated $\delta D$ profiles are often close to a mixing line in the lower troposphere (Appendix D1). Whatever $z_{orig}$ in the lower troposphere, the $\delta D_0$ calculated from Eq. (6) is nearly constant because the $\delta D$ profile is close to a mixing line (Appendix A, Fig. 4b). Whatever $z_{orig}$ in the middle troposphere, the $\delta D_0$ calculated from Eq. (6) is also nearly constant because $r_{orig}$ there is very small. So whatever $z_{orig}$, the $\delta D$ calculated from Eq. (6) is nearly constant, and the numerical solution fails.

However, it is possible that $\delta D$ profiles simulated by LMDZ are closer to mixing lines than real profiles, since GCMs are known to overestimate vertical mixing through the troposphere (Risi et al., 2012b) and to mix the lower free troposphere too frequently by deep convection in trade-wind regions (Nuijens et al., 2015a, b). Therefore, the shape of $\delta D$ profiles simulated by LMDZ is not a sufficient reason to reject the Rayleigh assumption. The uncertainty associated with this assumption is very difficult to quantify in LMDZ. More measurements of full $\delta D$ profiles are very welcome to help quantify it.

To summarize, $\delta D_0$ measurements could potentially be useful to estimate $z_{orig}$ with a useful precision, but if we measure daily $\delta D_f$ in the mid-troposphere, if the shape of $\delta D$ profiles can be better documented, if we measure $\delta D_0$ at different places to quantify the effect of horizontal advection, and if we can invent innovative techniques to better quantify the effect of rain evaporation. In addition in strato-cumulus clouds, we need to measure $\delta D_0$ with an accuracy of 0.1 ‰.

## 7 Conclusion

We propose an analytical model to predict the water vapor isotopic composition $\delta D_0$ of the sub-cloud layer (SCL) over tropical oceans. This model relies on the hypothesis that the altitude from which the air originates, $z_{orig}$, is an important factor. We build on B15 who extended the Merlivat and Jouzel (1979) closure equation to make the explicit link between $\delta D_0$ and mixing processes. We further extend their equation: we assume a shape for the $\delta D$ vertical profiles as a function of $q$, and we account for horizontal advection and rain evaporation effects.

The resulting equation highlights the fact that $\delta D_0$ is not sensitive to the intensity of mixing processes. Therefore, it is unlikely that water vapor isotopic measurements could help estimate the entrainment velocity that many studies have striven to estimate (Bretherton et al., 1995). In contrast, $\delta D_0$ is sensitive to the altitude from which the air originates. Based on a simulation with LMDZ and observations during the STRASSE cruise, we show that $z_{orig}$ is an important factor explaining the seasonal-spatial and daily variations in $\delta D_0$, especially in subsidence regions. In turn, could $\delta D_0$ measurements, combined with vertical profiles of humidity, temperature and $\delta D$, help estimate $z_{orig}$ and thus discriminate between different mixing processes? For such isotope-based estimates of $z_{orig}$ to be useful, we would need a precision of a few hundreds meters in deep convective regions and smaller than 20 m in strato-cumulus regions. To reach this target, we would need daily measurements of $\delta D$ in the mid-troposphere and very accurate measurements of $\delta D_0$, which are currently difficult to obtain. We would also need information on the horizontal distribution of $\delta D$ to account for horizontal advection effects, and full $\delta D$ profiles to quantify

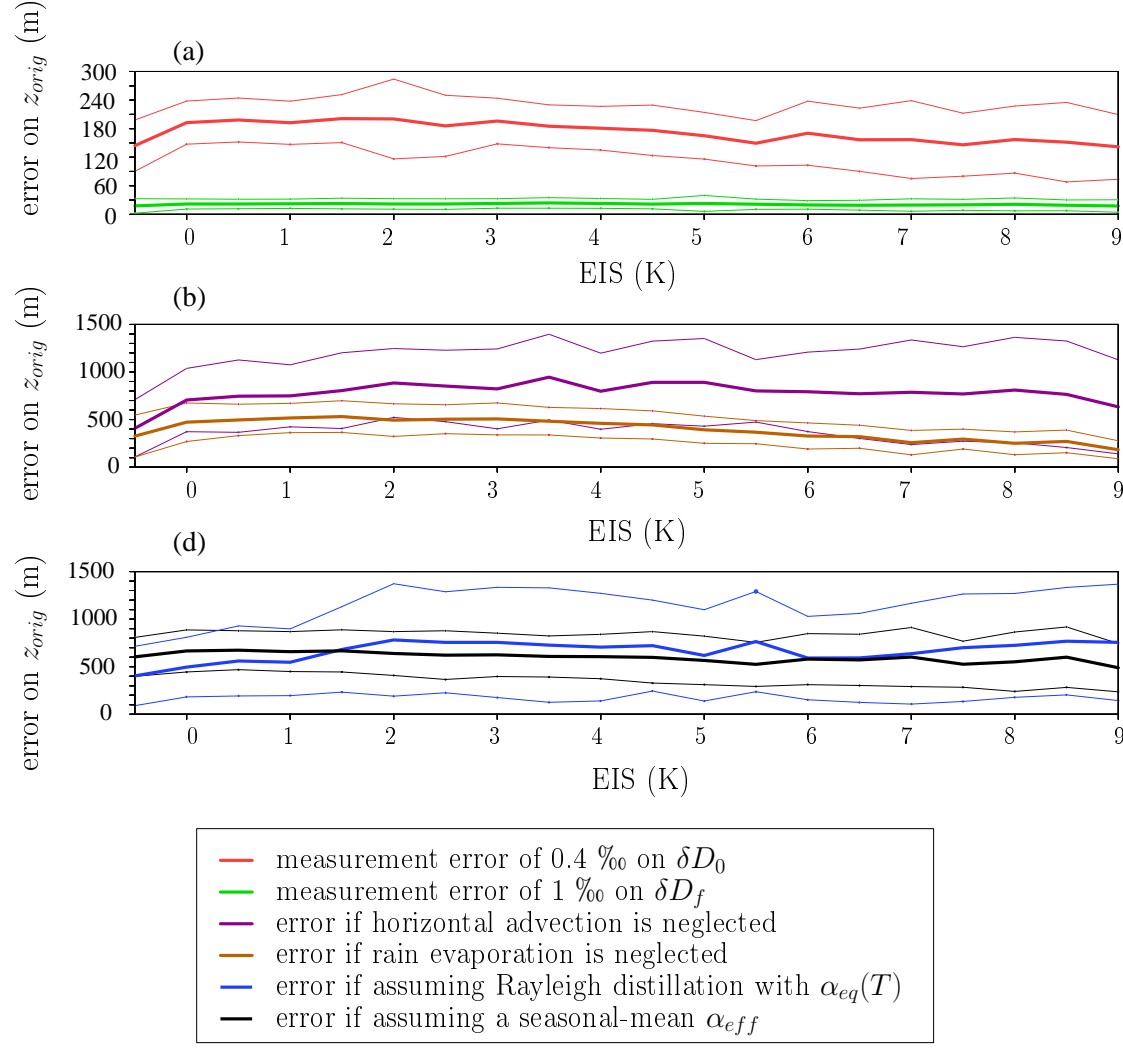

**Figure 14.** Errors when estimating $z_{orig}$ from $\delta D_0$ observations, as a function of EIS, as predicted by LMDZ. (a) Error we would make if $\delta D_0$ is measured with a 1‰ error (red), and error we would make if $\delta D_f$ is measured with a 1‰ error (green). (b) error we would make if we neglect horizontal advection effects (purple) and rain evaporation effects (brown). (c) Error if one uses $\alpha_{eq}$ as a function of local temperature to estimate $\alpha_{eff}$ (blue), error if one uses the seasonal-mean profile instead of the daily profile to estimate $\alpha_{eff}$ (black). The standard deviations among all daily errors estimated in each bin of EIS are also shown.

the uncertainty associated with the assumed shape for $\delta D$ profiles. Finally, rain evaporation is an issue in all regimes, even for strato-cumulus clouds. Innovative techniques would need to be developed to quantify this effect from observations.

This study is preliminary in many respects. First, it would be safe to check using water tagging experiments in LMDZ that $z_{orig}$ estimates really represent the altitude from which the air originates, and is not biased by our simplifying assumptions.

Second, the coarse vertical resolution of LMDZ, and the simplicity of mixing parameterizations (e.g. cloud top entrainment is not represented) are a limitation of this study. Ideally, the relationship between $\delta D_0$, $z_{orig}$ and the type of mixing processes should be investigated in isotope-enabled Large Eddy Simulations (LES) (Blossey et al., 2010; Moore et al., 2014). Artificial tracers and structure detection methods (Park et al., 2016; Brient et al., 2019), combined with conditional sampling methods (Couvreux et al., 2010), could help detect the different kinds of mixing structures, estimate their contributions to vertical trans-

port, and describe their isotopic signature. This would allow us to confirm, or infirm, many of the hypotheses and conclusions in this paper. Finally, if the sensitivity of $\delta D_0$ to the type of mixing processes is confirmed, paired isotopic simulations of single-column model (SCM) versions of general circulation models (GCM) and LES, forced by the same forcing, could be very useful to help evaluate and improve the representation of mixing and entrainment processes in GCMs, as is routinely the case for non-isotopic variables (Randall et al., 2003; Hourdin et al., 2013; Zhang et al., 2013).

*Code and data availability.* LMDZ can be downloaded from http://lmdz.lmd.jussieu.fr/. Program codes used for the analysis are available on https://prodn.idris.fr/thredds/catalog/ipsl_public/rlmd698/article_mixing_processes/d_pgmf/catalog.html.

Isotopic measurements from STRASSE can be downloaded from http://cds-espri.ipsl.fr/isowvdataatlantic/. All other datasets and processed files are available on https://prodn.idris.fr/thredds/catalog/ipsl_public/rlmd698/article_mixing_processes/catalog.html.

## Appendix A:  Closure if the tropospheric profile follows a mixing line

For simplicity, we neglect here horizontal advection and rain evaporation effects, but results would be similar otherwise. If we assume that $R_{orig}$ is uniquely related to $q_{orig}$ through a mixing line between the SCL air and a dry end-member ($q_f$, $R_f$):

$$q_{orig} = a \cdot q_0 + (1 - a) \cdot q_f \tag{A1}$$

and

$$R_{orig} = a \cdot q_0 \cdot R_0 + (1 - a) \cdot q_f \cdot R_f \tag{A2}$$

Reorganizing Eq. (A1), we get $a = \frac{r-p}{1-p}$ with $p = q_f/q_0$. Since $q_f \leq q_{orig} \leq q_0$, $p \leq r_{orig}$. Injecting Eq. (A2) into 6, we get:

$$R_0 = \frac{R_{oce}/\alpha_{eq} + p/(1-p) \cdot R_f \cdot \alpha_K \cdot (1 - h_0)}{h_0 + \alpha_K \cdot (1 - h_0)/(1 - p)} \tag{A3}$$

As a consistency check, in the limit case where the end-member is totally dry ($p = 0$), we find the MJ79 equation, i.e. Eq. (10).

It is intriguing to realize that $r_{orig}$ has disappeared from Eq. (A3). This can be understood physically: if the vertical profile follows a mixing line, it does not matter from which altitude the air comes: ultimately, what matters is how much dry air has been mixed directly or indirectly into the SCL (Fig. 4b). Therefore, if $R_{orig}$ follows a mixing line, we lose the sensitivity to $z_{orig}$.

## Appendix B: Diagnostics for rain evaporation in LMDZ

Rain evaporation can be accounted for in Eq. (8) if we can quantify $\eta$, the ratio of water vapor originating from rain evaporation to that originating from surface evaporation, and $\alpha_{evap}$, the ratio of isotopic ratio in the rain evaporation flux to $R_0$.

### B1 Equations

In LMDZ, two parameterization schemes can produce rain evaporation: the convective scheme and the large-scale condensation scheme. Their respective precipitation evaporation tendencies, $(dq/dt)_{evap,conv}$ and $(dq/dt)_{evap,lsc}$ , are given in $kg_{water} \cdot kg_{air}^{-1} \cdot s^{-1}$ and are used to calculate $F_{evap}$ in $kg_{water} \cdot m^{-2} \cdot s^{-1}$:

$$F_{evap} = \sum_{k=1}^{k_{LCL}} \left( (dq/dt)_{evap,conv} + (dq/dt)_{evap,lsc} \right) \cdot \frac{\Delta P_k}{g}$$

where $k_{LCL}$ is the last layer below the LCL, $\Delta P_k$ is the depth of layer $k$ in pressure coordinate and $g$ is gravity.

The isotopic equivalent of this flux, $F_{evap,iso}$ , is used to calculate $R_{evap} = F_{evap,iso}/F_{evap}$.

Only grid boxes and days where $F_{evap} > 0.05$ mm/d are considered to calculate $\alpha_{evap}$. This represents 94.0 % of all tropical oceanic grid boxes.

### B2 Results

Consistent with the larger amount of precipitation available for evaporation, $\eta$ is maximum in regions of deep convection, reaching 30 % around the Maritime continent (Fig. C1a). It is minimal over the dry descending regions, reaching 5 % off the coasts of Mauritania, Peru or Namibia. The rain evaporation is more depleted than the SCL in regions of strong deep convection, by as much as 70 ‰ around the Maritime continent (Fig. C1b). When the fraction of raindrops that evaporate is small, as is the case in such moist regions, isotopic fractionation favors evaporation of the lighter isotopologues. In these regions, rain evaporation has a depleting effect on the SCL, consistent with Worden et al. (2007). In contrast, in other regions, rain evaporation has a enriching effect on the SCL, up to 70 ‰ in dry regions. This is because in dry regions, rain evaporates almost totally, so that the evaporation flux has almost the same composition as the initial rain, which is more enriched than the water vapor.

## Appendix C: Diagnostics for horizontal advection in LMDZ

We can account for horizontal advection in Eq. (8) if we can quantify parameters $\phi = \frac{F_{adv} \cdot q_{adv}}{E}$, the ratio of water vapor coming from horizontal advection to that coming from surface evaporation, and $\beta = R_{adv}/R_0$, the ratio of isotopic ratios of horizontal advection to that of the SCL.

## C1 Equations

Let's assume that the box representing the SCL has a zonal extent $\Delta y$, a meridional extent $\Delta x$ and is composed of $k_{LCL}$ layers of vertical extent $\Delta z_k$. The quantity $F_{adv} \cdot q_{adv}$ represents the mass flux of water entering the grid box by horizontal advection per surface area, expressed in $kg_{water} \cdot s^{-1} \cdot m^{-2}$. Assuming an upstream advection scheme, it can be expressed as:

$$F_{adv} \cdot q_{adv} = \frac{\sum_{k=1}^{k_{LCL}} \left( \rho_k \cdot |u_k| \cdot q_{uk} \cdot \Delta y \cdot \Delta z_k + \rho_k \cdot |v_v| \cdot q_{vk} \cdot \Delta x \cdot \Delta z_k \right)}{\Delta x \cdot \Delta y} \tag{C1}$$

where $u_k$ and $v_k$ are the zonal and meridional wind components at layer $k$, $\rho_k$ is the volumic mass of air at layer $k$, and $q_{uk}$ and $q_{vk}$ are the humidities of the incoming air from zonal and meridional advection at layer $k$. When $u_k > 0$, $q_{uk}$ is the humidity in the grid box to the West. When $u_k < 0$, $q_{uk}$ is the humidity in the grid box to the East. When $v_k > 0$, $q_{vk}$ is the humidity in the grid box to the South. When $v_k < 0$, $q_{vk}$ is the humidity in the grid box to the North.

Applying the hydrostatic equation at each layer ($\Delta P_k = \rho_k \cdot g \cdot \Delta z_k$, where $g$ is gravity and $\Delta P_k$ is the vertical extent of the

layer $k$ in pressure coordinate), we get:

$$F_{adv} \cdot q_{adv} = \sum_{k=1}^{k_{LCL}} \frac{\Delta P_k}{g} \cdot \left( \frac{|u_k| \cdot q_{uk}}{\Delta x} + \frac{|v_k| \cdot q_{vk}}{\Delta y} \right)$$

The quantity $F_{adv}$ represents the incoming air mass flux by horizontal advection, and $q_{adv}$ represents the humidity of the incoming air. We can thus write them as:

$$F_{adv} = \sum_{k=1}^{k_{LCL}} \left( \frac{|u_k|}{\Delta x} + \frac{|v_k|}{\Delta y} \right) \cdot \frac{\Delta P_k}{g}$$

and

$$q_{adv} = \frac{\sum_{k=1}^{k_{LCL}} \left( \frac{|u_k|}{\Delta x} \cdot q_{uk} + \frac{|v_k|}{\Delta y} \cdot q_{vk} \right) \cdot \frac{\Delta P_k}{g}}{\sum_{k=1}^{k_{LCL}} \left( \frac{|u_k|}{\Delta x} + \frac{|v_k|}{\Delta y} \right) \cdot \frac{\Delta P_k}{g}}$$

The same budget as in Eq. C1 can be written for water isotopes:

$$F_{adv} \cdot q_{adv} \cdot R_{adv} = \frac{\sum_{k=1}^{k_{LCL}} \left( \rho_k \cdot |u_k| \cdot q_{uk} \cdot R_{uk} \cdot \Delta y \cdot \Delta z_k + \rho_k \cdot |v_v| \cdot q_{vk} \cdot R_{vk} \cdot \Delta x \cdot \Delta z_k \right)}{\Delta x \cdot \Delta y}$$

where $R_{adv}$ represents the isotopic ratio of the incoming water vapor:

$$R_{adv} = \frac{\sum_{k=1}^{k_{LCL}} \left( \frac{|u_k|}{\Delta x} \cdot q_{uk} \cdot R_{uk} + \frac{|v_k|}{\Delta y} \cdot q_{vk} \cdot R_{vk} \right) \cdot \frac{\Delta P_k}{g}}{\sum_{k=1}^{k_{LCL}} \left( \frac{|u_k|}{\Delta x} \cdot q_{uk} + \frac{|v_k|}{\Delta y} \cdot q_{vk} \right) \cdot \frac{\Delta P_k}{g}}$$

Note that the upstream advection scheme assumed here overestimates the effect of advection compared to the Van Leer (1977) advection scheme used in LMDZ. We thus estimate here an upper bound for the advection effect.

In practice, rather than calculating $\beta = R_{adv}/R_0$, we calculate $\beta = R_{adv}/R_{SCL}$ where $R_{SCL}$ is the isotopic ratio in average through the SCL:

$$R_{SCL} = \frac{\sum_{k=1}^{k_{LCL}} q_k \cdot R_k \frac{\Delta P_k}{g}}{\sum_{k=1}^{k_{LCL}} q_k \frac{\Delta P_k}{g}}$$

This prevents the advected water vapor to be systematically more depleted when the mixed-layer hypothesis is not exactly verified.

## C2  Results

Parameter $\phi$ is maximum where winds are maximum, such as near the extra-tropics or in the North Atlantic (Fig. C1a). Horizontal advection has an enriching effect in deep convective regions (probably because water vapor comes from nearby drier regions that have been less depleted by deep convection), and a depleting effect near the coasts (probably because of winds bringing vapor from the nearby land that is depleted by the continental effect) (Fig. C1b).

Note that in this formulation, parameters $\phi$ and $\beta$ are resolution-dependent. For example, in a finer resolution, $\phi$ would be larger and $\beta$ would be closer to 1, but $F_{adv} \cdot q_{adv} \cdot R_{adv}$ and thus the contribution of horizontal advection in Eq. (8) would remain the same.

## Appendix D:  LMDZ free tropospheric profiles

The goal of this appendix is to document the spatio-temporal variability in the shape (section D1) and steepness (section D2) of simulated free tropospheric $\delta D$ profiles. Note that a detailed interpretation of these profiles is beyond the scope of this paper. This paper aims at understanding $\delta D_0$, which is the first step towards understanding full tropospheric profiles. In turn, understanding full tropospheric profiles in future studies will help refine our model for $\delta D_0$.

## D1  Shape of tropospheric profiles

First, we test whether the $\delta D$ vertical profiles simulated by LMDZ follow a Rayleigh or mixing line as a function of $q$. For the Rayleigh curve, $\alpha_{eff}$ is estimated as explained in section 3.3. For the mixing line (Appendix A), the end member $(q_f, R_f)$ is also taken at 500 hPa.

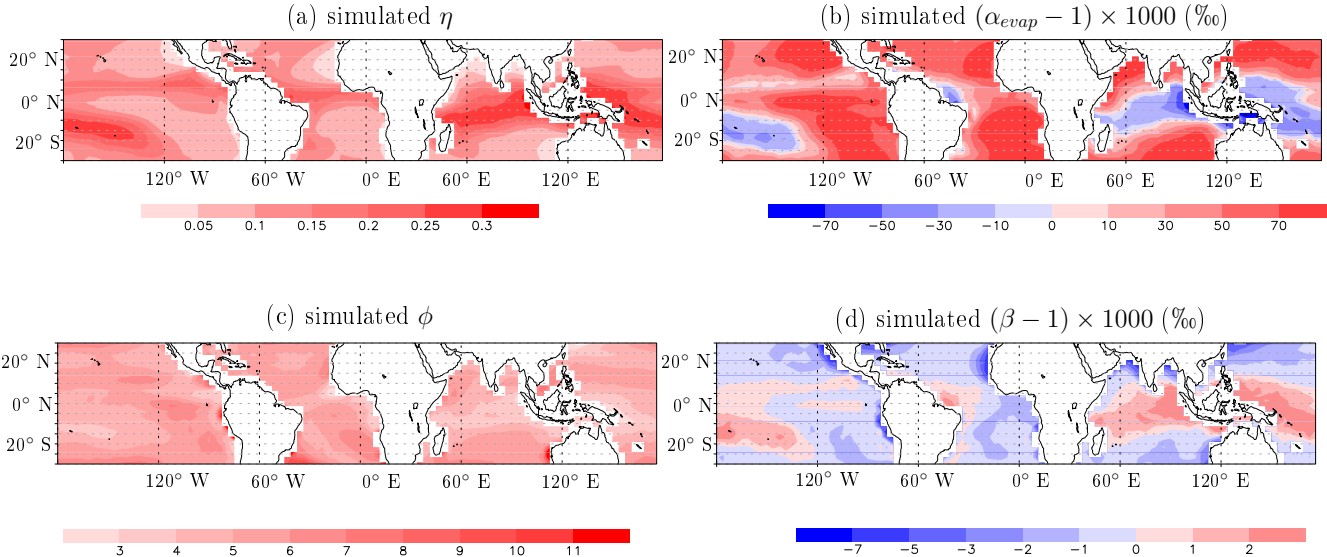

**Figure C1.** (a) Ratio $\eta$ of water in the SCL coming from rain evaporation to that coming from surface evaporation. (b) Effective fractionation coefficient $\alpha_{evap}$ between the SCL water vapor and the rain evaporation. (c) Ratio $\phi$ of water in the SCL coming from horizontal advection to that coming from surface evaporation. (d) Effective fractionation coefficient $\beta$ between the SCL water vapor and the water vapor coming from horizontal advection.

The tropical-mean vertical $\delta D$ profiles simulated by LMDZ is bounded by Rayleigh and mixing lines (Fig. D1a). To better document the spatial variability in the shape of $\delta D$ profiles, we plot parameter $f = \frac{\delta D_{LMDZ} - \delta D_{Rayleigh}}{\delta D_{mix} - \delta D_{Rayleigh}}$, describing how close is the simulated $\delta D$ ($\delta D_{LMDZ}$) to the Rayleigh ($\delta D_{Rayleigh}$) and mixing ($\delta D_{mix}$) lines. We have $f = 0$ in case of a Raleigh line, $f = 1$ in case of a mixing line, and $f > 1$ if $\delta D$ is more enriched than a mixing line. In the lower-troposphere,
$\delta D_{LMDZ}$ is close to a mixing line (and sometimes even more enriched) in deep convective regions (Indian Ocean, South Pacific Convergence Zone, Atlantic ITCZ), probably because deep convection efficiently mixes the lower troposphere. Elsewhere, $\delta D_{LMDZ}$ is intermediate between the two lines (Fig D1e). In the middle troposphere, $\delta D_{LMDZ}$ is relatively closer to Rayleigh everywhere (Fig D1b).

The daily variability of $f$ is large everywhere and at all levels (Fig D1c,e), with standard deviation of 0.23 and 0.44 on
tropical average at 1000 m and 4000 m respectively. A large daily variability in the shape of profiles is also observed in nature (Sodemann et al., 2017).

### D2    Steepness of tropospheric profiles

The steepness of the $\delta D$ gradient from the surface to the middle troposphere is described by the parameter $\alpha_{eff}$. It is maximum in regions of deep convection, for example around the Maritime Continent (Fig. D2a). This is consistent with the maximum
depletion simulated in deep convective regions in the mid-troposphere simulated by models (Bony et al., 2008), leading to

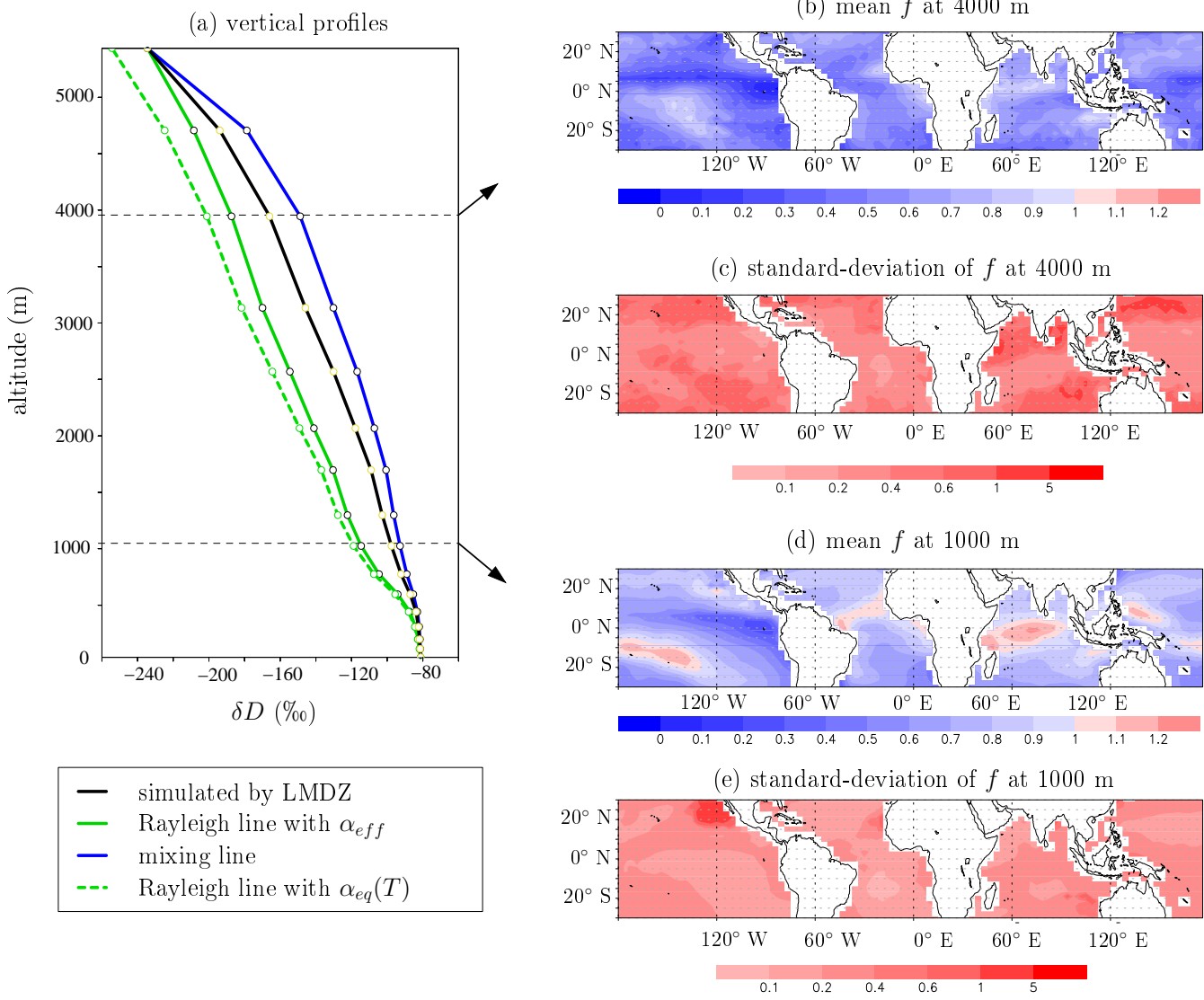

**Figure D1.** (a) Vertical profiles of water vapor $\delta D$ simulated by LMDZ (black), calculated if assuming a Rayleigh-type curve with $\alpha_{eff}$ estimated from $(q_f, \delta D_f)$ at 500 hPa (green), calculated if assuming a Rayleigh-type curve with $\alpha_{eq}(T)$ (dashed green) and calculated if assuming a mixing curve between the first layer and $(q_f, \delta D_f)$ at 500 hPa (blue), in average over all tropical oceanic locations and days. (b) Winter-mean map of parameter $f = \frac{\delta D_{LMDZ} - \delta D_{Rayleigh}}{\delta D_{mix} - \delta D_{Rayleigh}}$ at 4000 m, i.e. slightly below 500 hPa where $q_f$ and $\delta D_f$ are taken. Parameter $f$ describes how close is the simulated $\delta D$ ($\delta D_{LMDZ}$) to the Rayleigh ($\delta D_{Rayleigh}$) and mixing lines ($\delta D_{mix}$): $f = 0$ in case of a Raleigh line, $f = 1$ in case of a mixing line, and $f > 1$ if $\delta D$ is more enriched than a mixing line. (c) Standard deviation of parameter $f$ among all days in winter of all years, at 4000 m. (d) Same as (b but at 1000 m, i.e. slightly above the LCL. (e) Same as (c) but at 1000 m. To avoid numerical problems, only days and locations where $|\delta D_{mix} - \delta D_{Rayleigh}| > 5\,‰$ are used in the calculations.

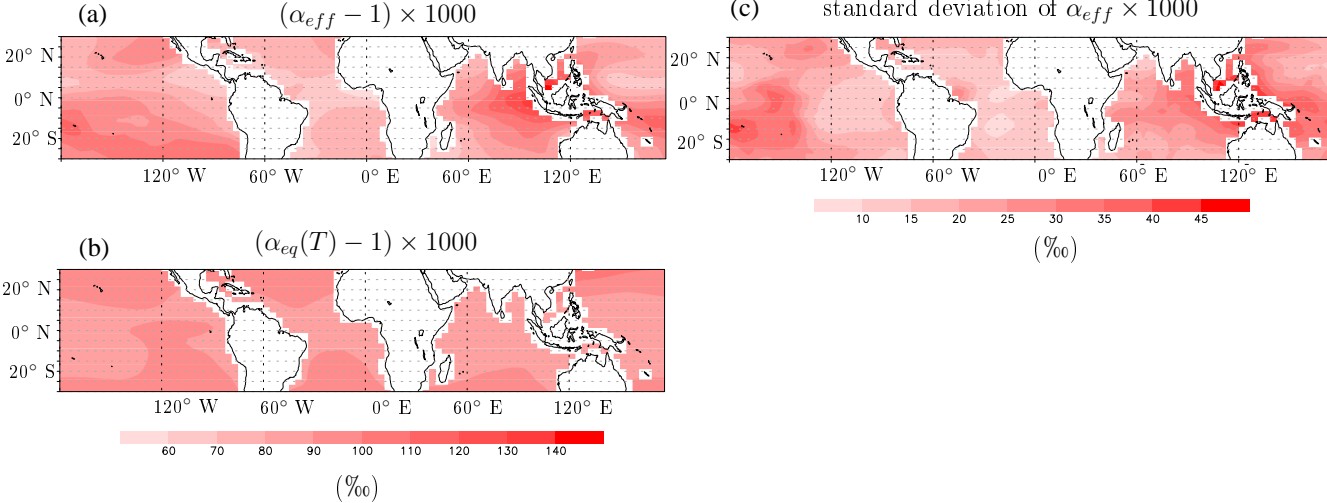

**Figure D2.** a) $\alpha_{eff} - 1$, where $\alpha_{eff}$ is the effective fractionation coefficient, expressed in ‰. b) $\alpha_{eq}(T) - 1$ expressed in ‰. All daily values are averaged over all days in winters of all years. c) Standard deviation of $\alpha_{eff}$ among all days in winter of all years, expressed in ‰.

steeper $\delta D$ profiles. The pattern of $\alpha_{eff}$ may also reflect horizontal advection effects, where strong isotopic gradients align with winds (e.g. from the Eastern to the Western Pacific, Dee et al., 2018).

Values of $\alpha_{eff}$ are of the same order of magnitude as real fractionation factors, but the spatial variations do not reflect those predicted if using a fractionation coefficient $\alpha_{eq}$ as a function of temperature $T$ (Fig. D2b).

5     The daily standard deviation of $\alpha_{eff}$ ($\sigma_{\alpha_{eff}}$) for a given season ranges from $5$‰ in the Central Atlantic to $40$‰ near the Maritime Continent (Fig. D2c). On average over all seasons and locations, daily $\alpha_{eff} - 1$ at a given location varies within $\pm 25$ % of its seasonal-mean mean value.

*Author contributions.* CR thought about the equations, ran the LMDZ simulations, made the analysis and wrote the manuscript. JG initiated the discussion on the subject and discussed regularly about the results. GR provided the STRASSE radiosoundings. FB provided insight and
10   references about cloud processes. JG, GR and FB all gave comments on the manuscript.

*Competing interests.* The authors declare that they have no conflict of interest.

*Acknowledgements.* This work was granted access to the HPC resources of IDRIS under the allocation 2092 made by GENCI. J. Galewsky was supported by the LABEX-IPSL visitor program, the Franco-American Fulbright Foundation, and NSF AGS Grant 1738075 to JG. We

thank Marion Benetti and Sandrine Bony for her previous studies on this subject and useful discussions. We thank two anonymous reviewers for their comments.

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
