# Peer review of "Controls on the water vapor isotopic composition near the surface of tropical oceans and role of boundary layer mixing processes"

_Atmospheric Chemistry and Physics, 2019_

## Referee Comment (RC1) · Anonymous Referee #1 · 17 May 2019

**Review of "Controls on the water vapor isotopic composition near the surface of tropical oceans and role of boundary layer mixing processes"**
by Camille Risi et al.

This paper presents an analytical steady state vertical mixing model to investigate the controls on the water vapor isotopic composition in the subcloud layer over the tropical oceans. It is a nicely simple model that considers the most important processes, which are surface evaporation and vertical mixing and predicts the subcloud layer water vapour isotope composition from a combination of mass balance equations for all isotope species. I enjoyed reading this paper very much, I particularly like the approach chosen for testing this analytical model, which combines model simulation data and ship-based observations. The ideas presented in this paper are exciting and very valuable for upcoming large field campaigns in which isotope observations are planned in different parts of the lower troposphere.

I thus recommend minor revisions with the following minor points:

1) In the abstract it should be clearly stated that the proposed analytical model is a steady state formulation, which neglects horizontal gradients and thus the impact of horizontal advection.
2) P. 1, L. 10: "When the air mixing into the SCL is lower in altitude it is moister": I think this is true most of the time and certainly in a climatological sense, but of course when including differential advection elevated moist layers can occur. I guess adding "it is generally moister" would make me very happy.
3) P. 3, L.4: Shouldn't cloud top cooling be mentioned here as well?
4) P. 5, L. 4: Neglecting the large-scale horizontal gradients in air properties, particularly in the trade wind regions seems to me like a strong assumption. Given the sensitivity of $\delta D$ to SST and the considerable SST gradient across the North Atlantic, I find that this caveat could be discussed a bit more explicitly here.
5) P. 5, L. 20: "qs is the saturation specific humidity at SST"
6) P. 6, L26: In the closure section and the discussion of the free tropospheric profile the role of horizontal advection is again neglected. This is maybe a good assumption in the tropic but it should still be mentioned explicitly.
7) P. 8, L26: "Depending on microphysical details that are too complex to be addressed here", Graf et al. 2019 could be referenced here
8) P. 9. Fig. 3: Which value was chosen for the SST? This could be mentioned in the caption as well as a reference to which equilibrium fractionation factor was used
9) P. 12, L. 2: "However, if the end member is defined below 500 hPa (e.g. 600 hPa), results are not always reasonable", why is this so?
10) P. 12, L. 7: In my opinion, this makes it difficult to interpret rorig. But probably there are conditions when rorig and thus zorig are more physically meaningful than others. Could the authors maybe add a list with explicit and quantitatively expressed conditions in which they would argue that the assumptions involved in Eq. 9 are satisfied?
11) P. 13, L. 4-5: Is there a literature reference that the authors could indicated for this calculation of zi from observations?

12) P. 13, L. 15: I did not immediately understand what was meant by composites belonging to a given interval of omega500, I was expecting a map. A reference to the results figure referred to here would have helped me.

13) P. 13, L. 22: By how much (range of variability) were the four factors varied?

14) P. 14, Section 4.1: This section seemed very technical for me. I also see it more as a methodological aspect than a result. I would recommend to either shift it to a technical appendix (since the paper is quite long) or to the methods section.

15) P. 15, Fig. 7: Mention that these are different random (?) grid points in the caption. I would have liked a more general evaluation also describing the temporal and spatial variability in the vertical profiles simulation by LMDZ.

16) P. 15, L. 3: could the authors mention the region where they think that alphaeff may also reflect horizontal advection effects?

17) P.16: I find it interesting that the mixing and Rayleigh lines have large biases in front of the eastern continental boundaries where the inversion is strongest and, where there is a strong decoupling between the FT and BL. In particular in these regions, I would expect horizontal advection to play a key role, (e.g. the SAL layer in front of the eastern North African Coast, see Lacour et al. 2017, ACP). Maybe the authors find a good way to shortly note this in the text.

18) P. 17, L. 2: Maybe one could add **oceanic** upwelling and **atmospheric** deep convection. Jumping from upwelling to deep convection in the same sentence, I was not sure whether deep convection in the ocean or the atmosphere was meant here.

19) P. 17, L. 4: "Decreases as omega500 is more strongly ascending or descending" -> "with increasing vertical winds (omega500) of both signs"

20) P. 17, L. 11: "in more ascending regions" -> a reference to Fig. 8d would have helped me here.

21) P. 17, L. 25: "The fact that the effect…" I had difficulties to understand this sentence.

22) P. 17, L. 33: h0 (62%) is the largest explained fraction of all the variables considered and should thus be put first. This could be a hint that large-scale horizontal advection plays an important role at the synoptic timescale in these regions.

23) P. 19, Fig. 10: The bin sizes (number of data points per bin should be added). If I understood correctly from the caption, the authors used the seasonal averaged fields from LMDZ. Why not making these composites using the 6-hourly outputs? For me there is a timescale discrepancy between the processes (mixing, evaporation) that the authors look at and the averaging timescale of the used fields.

24) P. 27, L. 28: a reference to a more technical paper such as Aemisegger et al. 2012 AMT, would be nice here.

Small technical comments:

1) P. 2, L. 22: "suffers **from** a low bias"

2) P. 3, L. 28: "capturing **the** second-order…"

3) P. 8, L. 26: no parenthesis after B)

4) P. 12, L. 7: "based" -> "b**i**ased"

5) P. 15, L. 1: Figure 8d

6) P. 15, L. 4: "Values **of** alphaeff…"

7) P. 15, L. 5: using a fractionation coefficient alpha eq **as** a function of temperature"

8) P. 17, L. 14: space missing between rorig and (

9) P. 18, L. 1: "Overall, **the** results…"

10) In general, the authors do not consistently use B15 for Benetti et al. (2015)

11) P. 21, L. 2: "with **the** strongest inversion"

12) P. 27, L. 27: measurement errors

13) P. 28, L. 2: "if **we** measure…"

14) P. 29, L. 14: very precise

15) P. 30, L. 11: **from** which altitude the air comes

**References:**

Aemisegger, F., Sturm, P., Graf, P., Sodemann, H., Pfahl, S., Knohl, A., and Wernli, H.: Measuring variations of $\delta^{18}O$ and $\delta^2H$ in atmospheric water vapour using two commercial laser-based spectrometers: an instrument characterisation study, Atmos. Meas. Tech., 5, 1491-1511, https://doi.org/10.5194/amt-5-1491-2012, 2012.

Graf, P., Wernli, H., Pfahl, S., and Sodemann, H.: A new interpretative framework for below-cloud effects on stable water isotopes in vapour and rain, Atmos. Chem. Phys., 19, 747-765, https://doi.org/10.5194/acp-19-747-2019, 2019.

Lacour, J.-L., Flamant, C., Risi, C., Clerbaux, C., and Coheur, P.-F.: Importance of the Saharan heat low in controlling the North Atlantic free tropospheric humidity budget deduced from IASI $\delta D$ observations, Atmos. Chem. Phys., 17, 9645-9663, https://doi.org/10.5194/acp-17-9645-2017, 2017.

---

## Referee Comment (RC2) · Anonymous Referee #2 · 25 May 2019

Summary

This paper presents a simple box model solving the water isotope budget in the sub-cloud layer to quantify the relative contributions of sea surface temperature, relative humidity, mid-tropospheric depletion, and the fraction of moisture from the free troposphere ($r_{orig}$) on the variability of δD in near-surface water vapor ($δD_0$). The contribution of $r_{orig}$ is further separated into contributions of specific humidity at the surface, and the height ($z_{orig}$), relative humidity and temperature from which the free tropospheric air originates. $Z_{orig}$ is found to be an important factor explaining the seasonal-spatial and daily variations of $δD_0$. This means that measurements of $δD_0$, if precise enough, can potentially be used to estimate $z_{orig}$ and distinguish between different mixing processes in the atmosphere.

The paper is interesting and well written, and it nicely demonstrates the use of measuring water vapor isotopes on short time scales. The box model's theoretical framework is described in detail and its drawbacks are clearly identified by the authors. I only have a few comments about the methods, the rest are mainly ideas for clarifying the paper. I recommend that the paper be published after minor revisions.

General comments

1) I like the method for quantifying the contributions of different factors by linear regression. I see how this works when the contributing factors have the same units as the variable of interest, which was the case in the previous studies that used this method and are cited in this paper (Risi et al. 2010, Oueslati et al., 2016). Here the different factors all have different units, and the slope therefore depends on the units, or how much the components vary. I assume this was accounted for somehow, as the slopes in the tables are all unitless, but it is not clear from the text, and makes me a bit skeptical about the results. More explanation on that would be useful.

2) As stated in the paper, the methods rely on the assumption that the δD profile follows a Rayleigh-like line, and that there is no effect of rain evaporation. Figures 7 and 8 show that the δD profile is often closer to a mixing line than a Rayleigh line, and the large contribution of $r_{orig}$ mainly comes from ascending regions, where clouds are most likely precipitating. It would be nice to see some quantification of how this impacts the results. A possible way to do this is to remove days/locations where the RMSE of the mixing line is smaller than the RMSE of the Rayleigh line and where there is precipitation, then repeat the analysis for these new fields and add the results in brackets in Tables 1, 2 and as dotted lines in Figures 10, 12.

3) The paper presents the new box model as an extension of the model by Benetti et al. (2015), which is technically true, but can be a bit misleading because its application is different. Rather than predicting $δD_0$ from $z_{orig}$, it predicts $z_{orig}$ from $δD_0$ and therefore requires $δD_0$ to be known. This means it cannot be applied to initialize Rayleigh models like the model by Benetti et al. (2015), which assumes constant $z_{orig}$. This could be written more clearly (e.g., from the abstract it seems like the model can be used to predict $δD_0$, which is only possible if $z_{orig}$ is known).

4) Changing some of the colors and colormaps could make the figures easier to understand. For example, I think the contributions of different factors and how they add up in Figures 9 and 11 would be more intuitive with a perceptually uniform colormap going from light to dark colors. Also, the red and pink lines in Figures 3, 4, 7, 10, 12, 15 look very similar to each other. It would be good to use a different color for one of them.

Specific comments

P1 L13: [D]/[H] instead of $[HDO]/[H_2O]$
P2 L22: high bias instead of low bias?
P2 L30: Please introduce the abbreviation for LCL
P3 L4: pointed out the important role
P3 L15: "We do not call it entrained": The word entrained/entrainment still appears a few times in the text (e.g. in Fig. 2, the title of section 4.4)
P3 L23: during a field campaign, global outputs of an isotope-enabled GCM.
P3 L24: "at the global scale": Really? There are no global maps. Are the numbers in Tables 1, 2 and the lines in Figures 10, 12 from global output, or from the region shown on the maps?
P3 L28: capturing the second-order parameter d-excess
P3 L32: "MJ79 already performs quite well for d-excess": Pfahl and Wernli (2009) would probably disagree.
P5 L23: $r \rightarrow r_{orig}$
P6 L20: measurements
P6 L20: "Therefore, variations of $\delta D_0$ that are mediated by $q_0$ or $h_0$ do not interest us": But $\delta D$ in the FT is prescribed as a function of q (confusing).
P8 L11: Refer to l'Hopital's rule?
P8 L21: follows as mixing line
P9: Fig.3: $\alpha_{eff} = \alpha_{eq}$ instead of $\alpha_{eff} = 1/\alpha_{eq}$
P11 L25: "Only profiles during the ascending phase of the balloons are considered": (Why?)
P11 L27 (title): write somewhere that these results are based on LMDZ output (not observations)
P12 Fig. 5: Describe abbreviations (LCL, EIS, SCL) in caption.
P12 L2: "if the end member is defined below 500hPa (e.g. 600hPa) results are not always reasonable": In what sense? Why?
P15 Fig. 7: What meteorological conditions do these examples represent? Would it be possible to show all (/more) simulated profiles in the background, e.g. in some transparent color, to get a better feeling for the variability? Also, I suggest adding markers to highlight where the levels are.
P15 L1: Figure 8d instead of 8c.
P15 L5: $\alpha_{eq}$ as a function of temperature
P16 Fig. 8: in boreal winters of all years
P17 L22: "in the cold upwelling regions": for example where?
P17 L23: probably reflects
P17 L24: "the effect of $r_{orig}$ can be seen on the composites as a function of EIS and not as a function of $\omega_{500}$": I don't see this, please elaborate.
P17 L30: followed by $h_0$ (23%), $r_{orig}$ (16%), ...

P18 Fig. 9: Are the correlations significant everywhere? Otherwise, add hatching where not significant?

P19 Fig.10: $\omega_{500}$ (hPa/d)

P20 Tab. 2: $q_0$ seems to be important in Fig. 12, but the slope is 0.0 here, $h_0$ seems to be unimportant in Fig. 12 but slope is 0.91 here. Why is that?

P20 L1: "it would translate into a lower $z_{orig}$.": Why?

P22 Fig.12: $\omega_{500}$ (hPa/d)

P25 L6: the cruises goes

P25 L8: "when considering only the 6 data points when $z_{orig}$ < 2000m": Rationale behind this?

P25 L14: ... at the seasonal-spatial and daily scale is the proportion of the water vapor in the SCL that is originates from above

P26 Fig. 15: r → $r_{orig}$

P27 L1: there → they

P27 L13: the temporal variability of $\alpha_{eff}$. Is it possible to estimate the uncertainty from the spatial variability of $\alpha_{eff}$ as well (in the vertical, i.e. how much the $\delta D$ profile differs from a Rayleigh line with constant $\alpha_{eff}$)?

P27 L21: estimating $z_{orig}$ from $\delta D_0$ measurements on a daily basis (?)

P28 L2: and if we measure

P28 L3: swap trade-wind cumulus and strato-cumulus clouds

P29 L14: very precised estimates

P29 L18: the altitude from which the air is originates, and is not to biased by

References

Benetti, M., Aloisi, G., Reverdin, G., Risi, C., and Sèze, G.: Importance of boundary layer mixing for the isotopic composition of surface vapor over the subtropical North Atlantic Ocean, J. Geophys. Res. Atmos., 120, 2190–2209, doi:10.1002/2014JD021947, 2015.

Oueslati, B., Bony, S., Risi, C., and Dufresne, J.-L.: Interpreting the inter-model spread in regional precipitation projections in the tropics: role of surface evaporation and cloud radiative effects, Clim. Dyn., 47, 2801–2815, doi:0.1007/s00382-016-2998-6, 2016.

Pfahl, S., and H. Wernli, Lagrangian simulations of stable isotopes in water vapor: An evaluation of nonequilibrium fractionation in the Craig-Gordon model, J. Geophys. Res., 114, D20108, doi:10.1029/2009JD012054, 2009.

Risi, C., Bony, S., Vimeux, F., Frankenberg, C., and Noone, D.: Understanding the Sahelian water budget through the isotopic composition of water vapor and precipitation, J. Geophys. Res, 115, D24110, doi:10.1029/2010JD014 690, 2010.

---

## Author Comment (AC1) · 4 Jul 2019

**Response to reviewers**

July 4, 2019

**Summary:**

We thank both reviewers for their comments. As detailed in the point-by-point response below, we have taken them into account. We have already implemented most of the necessary changes in a revised manuscript, which we are motivated to re-submit in the next few weeks. However, we still need more time to work on 4 aspects that are mentionned by the reviewers, because they need substantial new analyses:

1. quantify the effect of horizontal advection, add an appendix on the method for this quantification, and discuss this effect in the main text.

2. quantify the effect of rain evaporation, add this quantification in the appendix that is already devoted to this subject, and discuss this effect in the main text.

3. quantify the effect of assuming a Rayleigh curve for $\delta D$ with constant $\alpha_{eff}$, and discuss this effect in the main text.

4. better document and illustrate the spatio-temporal variability in free tropospheric profiles.

We have already started to work on these aspects, as detailed below, but we have not finished yet. We will use the revision time to finish the work and address more completely the reviewers' comments on these specific issues.

**1 Reviewer 1**

We thank reviewer 1 for his/her comments.

This paper presents an analytical steady state vertical mixing model to investigate the controls on the water vapor isotopic composition in the subcloud layer over the tropical oceans. It is a nicely simple model that considers the most important processes, which are surface evaporation and vertical mixing and predicts the subcloud layer water vapour isotope composition from a combination of mass balance equations for all isotope species. I enjoyed reading this paper very much, I particularly like the approach chosen for testing this analytical model, which combines model simulation data and ship-based observations. The ideas presented in this paper are exciting and very valuable for upcoming large field campaigns in which isotope observations are planned in different parts of the lower troposphere.

I thus recommend minor revisions with the following minor points:

1) In the abstract it should be clearly stated that the proposed analytical model is a steady state formulation, which neglects horizontal gradients and thus the impact of horizontal advection.

We agree: we modify as follows: "We propose a steady-state analytical model ... The model relies on the hypothesis that $\delta D$ profiles are steeper than mixing lines, and we neglect the effects of rain evaporation and horizontal advection on $\delta D_0$"

2) P. 1, L. 10: "When the air mixing into the SCL is lower in altitude it is moister": I think this is true most of the time and certainly in a climatological sense, but of course when including differential advection elevated moist layers can occur. I guess adding "it is generally moister" would make me very happy.

We modify as suggested.

3) P. 3, L. 4: Shouldn't cloud top cooling be mentioned here as well?

We modify as: "driven by cloud-top radiative cooling, mixing and evaporative cooling of droplets"

4) P. 5, L. 4: Neglecting the large-scale horizontal gradients in air properties, particularly in the trade wind regions seems to me like a strong assumption. Given the sensitivity of dD to SST and the considerable SST gradient across the North Atlantic, I find that this caveat could be discussed a bit more explicitly here.

We agree that the neglect of horizontal advection is an important caveat that we need to discuss more, and if possible, quantify. There are several comments along this line from you and from reviewer 2.

We are working on a way to rigorously estimate the effect of horizontal advection of

isotopic gradients on our results. To do so, we extend our Eq. 9: we get:

$$R_0 = \frac{R_{oce}}{\alpha_{eq}} \cdot \frac{1}{h_0 + \alpha_K \cdot (1 - h_0) \cdot \left( \frac{1 - r_{orig}^{\alpha_{eff}}}{1 - r_{orig}} + \phi \cdot (1 - \beta) \right)} \tag{1}$$

where $\phi = \frac{F_{adv} \cdot q_0}{E}$, $\beta = \frac{R_{adv}}{R_0}$, $F_{adv}$ is the air flux coming from advection in kg/m2/s, $R_{adv}$ is the isotopic ratio of the advected air assuming an upstream advection scheme. All these variables can be diagnosed from the model outputs.

We have not finished yet, we will continue to work on it during the revision time. If this method works well, we will add an additional appendix in the paper, analogous to the appendix on rain evaporation. In the main text, we will add the results from this quantification. Discussions will be added on how horizontal advection affects our results.

Coming back to the effect of horizontal gradients in the North Atlantic, with the above-mentionned equation, we hope to get a map that quantifies the effect of horizontal advection. If this maps supports your comment, we will add a sentence on the North Atlantic that reflects your comment.

5) P. 5, L. 20: "qs is the saturation specific humidity at SST"

corrected

6) P. 6, L26: In the closure section and the discussion of the free tropospheric profile the role of horizontal advection is again neglected. This is maybe a good assumption in the tropic but it should still be mentioned explicitly.

We neglect horizontal advection in the box model described in section 2.1. However,

our closure assumption does not need to neglect horizontal advection. Horizontal advection effects can be implicitly taken into account through the $\alpha_{eff}$: we now clarify this: "Effects of horizontal advection and rain evaporation are encapsulated into $\alpha_{eff}$. " As detailed below as a response to another comment, we now give specific examples of how horizontal advection may affect $\alpha_{eff}$.

It is possible that horizontal advection distorts the $\delta D$ profile from a Rayleigh curve. We will quantify the effect of assuming a Rayleigh curve compared to the full simulated profile by LMDZ, as detailed further down (equation 2 of this response). This will allow us to address this issue.

7) P. 8, L26: "Depending on microphysical details that are too complex to be addressed here", Graf et al. 2019 could be referenced here

We add this reference.

8) P. 9. Fig. 3: Which value was chosen for the SST? This could be mentioned in the caption as well as a reference to which equilibrium fractionation factor was used

We add this information in the caption: "For this illustrative purpose, we assume SST=30℃, $h_0 = 0.8$ and $\delta D_{oce} = 0$. ". We also add it to Fig. 4.

9) P. 12, L. 2: "However, if the end member is defined below 500 hPa (e.g. 600 hPa), results are not always reasonable", why is this so?

Now we explain this in the text: "However, the end member should be defined above

500 hPa to ensure that it is well above boundary layer processes. If the end member is defined below 500 hPa (e.g. 600 hPa), there are a few cases where $q$ increases with altitude ($q_f > q_0$) due to horizontal advection or convective detrainment from nearby moister regions. Meanwhile, $\delta D$ decreases monotically, leading to unrealistic values for $\alpha_{eff}$.".

10) P. 12, L. 7: In my opinion, this makes it difficult to interpret rorig. But probably there are conditions when rorig and thus zorig are more physically meaningful than others. Could the authors maybe add a list with explicit and quantitatively expressed conditions in which they would argue that the assumptions involved in Eq. 9 are satisfied?

We will work on "quantitatively expressed conditions" for the valididity of assumption in Eq. 9 during the revision time. Off course, the difficulty in this work is the "quantitatively".

- As explained above (equation 1 of this response), we will quantify the effect of horizontal advection on $\delta D_0$, $r_{orig}$ and $z_{orig}$.

- In addition, we will make use of the equation that is already in appendix B to quantify the effect of rain evaporation. We already have all the necessary LMDZ outputs to diagnose the $\eta$ and $\alpha_{re}$ parameters.

11) P. 13, L. 4-5: Is there a literature reference that the authors could indicated for this calculation of zi from observations?

We now give several litterature references and more explanation on this calculation

method: "The temperature inversion is an abrupt increase in temperature that capps the boundary layer. Therefore, a method to automatically estimate its altitude is to detect a maximum in the vertical gradient of potential temperature (Stull (1988); Oke (1988); Sorbjan (1989); Garratt (1994); Siebert et al. (2000)). This method is senstive to the resolution of vertical profiles (Siebert et al. (2000); Seidel et al. (2010)). Therefore, we adapted this method in order to yield $z_i$ values that best agree with what we would estimate from visual inspection of individual temperature profiles. In LMDZ, we calculate $z_i$ as the first level at which the vertical potential temperature gradient exceeds 3 times the moist-adiabatic lapse rate. In observations, we calculate $z_i$ as the first level at which the vertical potential temperature gradient exceeds 5 times the moist-adiabatic lapse rate, because radio-soundings are noisier than simulated profiles. "

12) P. 13, L. 15: I did not immediately understand what was meant by composites belonging to a given interval of omega500, I was expecting a map. A reference to the results figure referred to here would have helped me.

We now explain better how the composites are calculated, and we add a reference to Fig. 10 as an example: "The type of clouds and mixing processes depends strongly on the large-scale velocity at 500 hPa ($\omega_{500}$, map show in Fig. 6a), with shallow clouds in subsiding regions and deeper clouds in ascending regions (Fig. 1). Therefore, it is convenient to plot variables as composites as a function of $\omega_{500}$ (Bony et al. (2004)). To make such plots, we divide the $\omega_{500}$ range from -30 to 50 hPa/d into intervals of 5 hPa/d. In each given interval, we average all seasonal-mean values at all locations over tropical oceans that belong to a this interval (e.g. Fig. 10a will be an example).

The cloud cover strongly correlates with the inversion strength, which can ve quantified by the Estimated Inversion Strength (EIS, Wood and Bretherton (2006), map shown in Fig. 6b). We thus also plot variables as composites as a function of EIS. To make such

plots, we divide the EIS range from -1 K to 9 K into intervals of 0.5K. In each given interval, we average all seasonal-mean values at all locations over tropical oceans that belong to this interval l (e.g. Fig. 10b will be an example). "

13) P. 13, L. 22: By how much (range of variability) were the four factors varied?

The four factors were varied from a control value to their simulated value.

We now clarify this in the text with a modified paragraph: "To understand what controls the $\delta D_0$ spatio-temporal variations, $\delta D_0$ is decomposed into 4 contributions based on Eq. (9). First, we define $r_{orig,basic} = 0.6$, $\alpha_{eff,basic} = 1.09$, $SST_{basic}$=25 °C, $h_{0,basic} = 0.8$ as a basic state. We call $\delta D_{eq9}(r_{orig}, \alpha_{eff}, SST, h_0)$ the function giving $\delta D_0$ as a function of $r_{orig}$, $\alpha_{eff}$, SST and $h_0$ following the Eq. (9). The relative contributions of $r_{orig}$, $\alpha_{eff}$, $SST$ and $h_0$ to $\delta D_0$ variations are estimated as $\delta D_{eq9}(r_{orig}, \alpha_{eff,basic}, SST_{basic}, h_{0,basic})$, $\delta D_{eq9}(r_{orig,basic}, \alpha_{eff}, SST_{basic}, h_{0,basic})$, $\delta D_{eq9}(r_{orig,basic}, \alpha_{eff,basic}, SST, h_{0,basic})$ and $\delta D_{eq9}(r_{orig,basic}, \alpha_{eff,basic}, SST_{basic}, h_0)$ respectively. ".

14) P. 14, Section 4.1: This section seemed very technical for me. I also see it more as a methodological aspect than a result. I would recommend to either shift it to a technical appendix (since the paper is quite long) or to the methods section.

We will work during the revision time on moving this section to the method section or appendix. This may depend on what we do to address the comments on better documenting the spatio-temporal variability in free tropospheric $\delta D$ profiles.

15) P. 15, Fig. 7: Mention that these are different random (?) grid points in the caption. I would have liked a more general evaluation also describing the temporal and spatial variability in the vertical profiles simulation by LMDZ.

We now explain how these points were selected: "These profiles were selected automatically as the first day and tropical ocean location (scanning all latitudes from South to North and longitudes from 180W to 180E) for which the RMS difference between the LMDZ profile and the Rayleigh line ($RMS_{Rayleigh}$) and the RMS difference between the LMDZ profile and the Rayleigh line ($RMS_{mixing}$) satisfy the following conditions: (a) $|RMS_{Rayleigh} - 10| < 1 permil$ and $|RMS_{mixing} - 25| < 1 permil$, (b) $|RMS_{Rayleigh} - 25| < 1 permil$ and $|RMS_{Rayleigh} - 10| < 1$ and (c) $|RMS_{Rayleigh} - 25| < 1 permil$ and $|RMS_{Rayleigh} - 25| < 1 permil$.".

Regarding the "more general evaluation also describing the temporal and spatial variability in the vertical profiles simulation by LMDZ": this comment echoes one form the second reviewer. We will think about how to address it during the revision time.

16) P. 15, L. 3: could the authors mention the region where they think that alphaeff may also reflect horizontal advection effects?

Although we will work on quantifying the effect of horizntal advection on $\delta D_0$, i.e. in the SCL, the effect of horizontal advection on $\alpha_{eff}$ is a completely different subject that is beyond the scope of this paper. Therefore, we only reply based on the litterature: "The pattern of $\alpha_{eff}$ may also reflect horizontal advection effects, where strong isotopic gradients align with winds (e.g. from the Eastern to the Western Pacific, Dee et al. (2018)). "

17) P.16: I find it interesting that the mixing and Rayleigh lines have large biases in front of the eastern continental boundaries where the inversion is strongest and, where there is a strong decoupling between the FT and BL. In particular in these regions, I would expect horizontal advection to play a key role, (e.g. the SAL layer in front of the eastern North African Coast, see Lacour et al. 2017, ACP). Maybe the authors find a good way to shortly note this in the text.

We now add this comment: "For example, we note that mixing and Rayleigh lines have large biases in front of the eastern continental boundaries where the inversion is strongest, leading to a strong decoupling between the FT and the boundary layer. Horizontal advection is expected to play a key role in these regions (e.g. the Saharian layer in front of the eastern North African Coast, Lacour et al. (2017)). "

18)P. 17, L. 2: Maybe one could add oceanic upwelling and atmospheric deep convection. Jumping from upwelling to deep convection in the same sentence, I was not sure whether deep convection in the ocean or the atmosphere was meant here.

We modify as suggested.

19) P. 17, L. 4: "Decreases as omega500 is more strongly ascending or descending" -> "with increasing vertical winds (omega500) of both signs"

We modify as suggested.

20) P. 17, L. 11: "in more ascending regions" -> a reference to Fig. 8d would have helped me here.

We add this reference

21) P. 17, L. 25: "The fact that the effect..." I had difficulties to understand this sentence.

We remove this sentence that was not so useful.

22) P. 17, L. 33: $h_0$ (62%) is the largest explained fraction of all the variables considered and should thus be put first. This could be a hint that large-scale horizontal advection plays an important role at the synoptic timescale in these regions.

We now move $h_0$ first in the sentence.

We will write a comment on horizontal advection once we have quantified its effects.

23) P. 19, Fig. 10: The bin sizes (number of data points per bin should be added).

We now add this information in the figures, and we write in the caption: "The number of samples in each bin is indicated on a logarithmic scale on the right-hand-side as bars."

If I understood correctly from the caption, the authors used the seasonal averaged fields from LMDZ. Why not making these composites using the 6-hourly outputs? For me there is a timescale discrepancy between the processes (mixing, evaporation) that the authors look at and the averaging timescale of the used fields.

The composites are based on seasonal-mean EIS or $\omega_{500}$ because this allows a better link with the large-scale dynamical regime. Mixing and evaporation are processes that act at short time scales, but their relationship to the large-scale circulation is best constrained by energetics at time scales longer than synoptic. Let's consider $\omega_{500,}$ for example. It relates to convective activity and other diabatic processes through the conservation equation of moist static energy. Adiabatic cooling by large-scale ascent balances latent heating by convection (Yanai et al. (1973)), or adiabatic heating by large-scale subsidence balances radiative cooling (Emanuel et al. (1994)). The stationarity in the conservation of moist static equation is most valid at scales longer than synoptic, otherwise, the storage term becomes important (Masunaga and Sumi (2017)). This is why in Bony et al. (2004) and subsequent papers (e.g. Bony et al. (2013)) based on $\omega_{500}$, monthly-mean $\omega_{500}$ is used, which yields similar results to seasonal-mean.

A similar rationale applies to EIS. This is why many papers on EIS use seasonal-mean values, notably the paper defining this quantity (Wood and Bretherton (2006)).

We now explain this in section 3.5: "Note that such composites are done on seasonal-mean $\omega_{500}$ because cloud processes and their associated diabatic heating are tied to the large-scale circulation through energetic constrains (Yanai et al. (1973); Emanuel et al. (1994)) that are best valid at longer time scales (otherwise, the energy storage term may become significant, e.g. Masunaga and Sumi (2017)). This is why $\omega_{500}$ is generally averaged over a month or longer (e.g. Bony et al. (2004); Bony et al. (2013))"

and for EIS: "Using seasonal-mean values is consistent with Wood and Bretherton (2006) and with the better link at longer time scales between cloud processes and the large-scale dynamical regime.".

In addition, we do not look at the diurnal variations: the stationarity assumption in our simple model would be violated. We now explain this in the abstract: "the steady-state assumption restricts the application of this model to time scales longer than daily." and in section 2.1: "We assume that the SCL is at steady state. For example, its depth is constant. Since the SCL properties may exhibit a diurnal cycle (Duynkerke et al. (2004)), this hypothesis restricts the application of this model to time scales longer than daily. "

24) P. 27, L. 28: a reference to a more technical paper such as Aemisegger et al. 2012 AMT, would be nice here.

We add this reference

Small technical comments:

1) P. 2, L. 22 : "suffers **from** a low bias"

2) P. 3, L. 28: "capturing **the** second-order..."

3) P. 8, L. 26: no parenthesis after B)

4) P. 12, L.7: "based" -> "biased"

5) P. 15, L . 1: Figure 8d

6) P. 15, L. 4: "Values **of** alphaeff..."

7) P. 15, L. 5: using a fractionation coefficient alpha eq **as** a function of temperature"

8) P. 17, L. 14: space missing between rorig and (

9) P. 18, L. 1: "Overall, **the** results..."

We correct all these mistakes.

10) In general, the authors do not consistently use B15 for Benetti et al. (2015)

We now use B15 consistently.

11) P. 21, L. 2: "with **the** strongest inversion"

12) P. 27, L. 27: measurement errors

13) P. 28, L. 2: "if **we** measure..."

14) P. 29, L. 14: very precise

15) P. 30, L. 11: **from** which altitude the air comes

We correct all these mistakes.

**2 Reviewer 2**

We thank reviewer 2 for his/her comments.

This paper presents a simple box model solving the water isotope budget in the sub-cloud layer to quantify the relative contributions of sea surface temperature, relative humidity, mid-tropospheric depletion, and the fraction of moisture from the free troposphere (rorig) on the variability of $\delta D$ in near-surface water vapor ($\delta D_0$). The contribution of $r_{orig}$ is further separated into contributions of specific humidity at the surface, and the height ($z_{orig}$), relative humidity and temperature from which the free tropospheric air originates. Zorigis found to be an important factor explaining the seasonal-spatial and daily variations of $\delta D_0$. This means that measurements of Dd0, if precise enough, can potentially be used to estimate $z_{orig}$ and distinguish between different mixing processes in the atmosphere.

The paper is interesting and well written, and it nicely demonstrates the use of measuring water vapor isotopes on short time scales. The box model's theoretical framework is describedin detail and its drawbacks are clearly identified by the authors. I only have a few comments about the methods, the rest are mainly ideas for clarifying the paper. I recommend that the paper be published after minor revisions.

General comments

1) I like the method for quantifying the contributions of different factors by linear regression. I see how this works when the contributing factors have the same units as the variable of interest, which was the case in the previous studies that used this method and are cited in this paper (Risi et al. 2010, Oueslati et al., 2016). Here the different factors all have different units, and the slope therefore depends on the units, or how much the components vary. I assume this was accounted for somehow, as the slopes in the tables are all unitless, but it is not clear from the text, and makes me a bit skeptical about the results. More explanation on that would be useful.

The contributing factors have the same units as the variable of interest, i.e. permil. We now clarify this in the text: "To understand what controls the $\delta D_0$ spatio-temporal variations, $\delta D_0$ is decomposed into 4 contributions based on Eq. (9). First, we define $r_{orig,basic} = 0.6$, $\alpha_{eff,basic} = 1.09$ , $SST_{basic}$=25℃, $h_{0,basic} = 0.8$ as a basic state. We call $\delta D_{eq9}(r_{orig}, \alpha_{eff}, SST, h_0)$ the function giving $\delta D_0$ as a function of $r_{orig}$, $\alpha_{eff}$, SST and $h_0$ following the Eq. (9). The relative contributions of $r_{orig}$, $\alpha_{eff}$, $SST$ and $h_0$ to $\delta D_0$ variations are estimated as $\delta D_{eq9}(r_{orig}, \alpha_{eff,basic}, SST_{basic}, h_{0,basic})$, $\delta D_{eq9}(r_{orig,basic}, \alpha_{eff}, SST_{basic}, h_{0,basic})$, $\delta D_{eq9}(r_{orig,basic}, \alpha_{eff,basic}, SST, h_{0,basic})$ and $\delta D_{eq9}(r_{orig,basic}, \alpha_{eff,basic}, SST_{basic}, h_0)$ respectively. All these components have the same units as $\delta D_0$ (permil). "

2) As stated in the paper, the methods rely on the assumption that the $\delta D$ profile follows a Rayleigh-like line, and that there is no effect of rain evaporation. Figures 7 and 8 show that the $\delta D$ profile is often closer to a mixing line than a Rayleigh line, and the large contribution of $r_{orig}$ mainly comes from ascending regions, where clouds are most likely precipitating. It would be nice to see some quantification of how this impacts the results. A possible way to do this is to remove days/locations where the RMSE of the mixing line is smaller than the RMSE of the Rayleigh line and where there is precipitation, then repeat the analysis for these new fields and add the results in brackets in Tables 1, 2 and as dotted lines in Figures 10, 12.

We are working on quantifying the effect of rain evaporation on our results using the equation in the appendix B and diagnostics of $\eta$ and $\alpha_{re}$ from LMDZ. We do not have the results yet. Depending on our results, we will decide what is the best way to show this quantification in the paper.

3) The paper presents the new box model as an extension of the model by Benetti et al. (2015), which is technically true, but can be a bit misleading because its application is different. Rather than predicting $\delta D_0$ from zorig, it predicts $z_{orig}$ from $\delta D_0$ and therefore requires $\delta D_0$ to be known. This means it cannot be applied to initialize Rayleigh models like the model by Benetti et al. (2015), which assumes constant zorig. This could be written more clearly (e.g., from the abstract it seems like the model can be used to predict $\delta D_0$, which is only possible if $z_{orig}$ is known).

The model can be used both ways, either to predict $\delta D_0$ as a function of $z_{orig}$, or to estimate $z_{orig}$ from $\delta D_0$. If someone wants to initialize Rayleigh models with it, one can

make assumptions on $z_{orig}$. We write: "We propose a steady-state analytical model to predict $\delta D_0$ as a function of ... and the altitude from which the free tropospheric air originates ($z_{orig}$). "

4) Changing some of the colors and colormaps could make the figures easier to understand. For example, I think the contributions of different factors and how they add up in Figures 9 and 11 would be more intuitive with a perceptually uniform colormap going from light to dark colors. Also, the red and pink lines in Figures 3, 4, 7, 10, 12, 15 look very similar to each other. It would be good to use a different color for one of them.

- I read about color scales and I changed the color scales from rainbow to single hue in all maps.

- We now change the pink into purple in all the Figures.

Specific comments

P1 L13: [D]/[H] instead of [HDO]/[H2O]

We modify.

P2 L22: high bias instead of low bias?

Corrected.

P2 L30: Please introduce the abbreviation for LCL

Done.

P3 L4: pointed **out** the important role

Corrected.

P3 L15: "We do not call it entrained": The word entrained/entrainment still appears a few times in the text (e.g. in Fig. 2, the title of section 4.4)

We modify all the occurences of "entrainment", except when it really refers to entrainment.

P3 L23: during **a** field campaign, global outputs **of** an isotope-enabled GCM.

Corrected.

P3 L24: "at the global scale": Really? There are no global maps. Are the numbers in Tables 1, 2 and the lines in Figures 10, 12 from global output, or from the region shown on the maps?

We now modify by "in the Tropics". We precise in the captions for these Tables and Figures: "All seasons and locations over tropical oceans ($30°N - 30°S$, ocean fraction>80%) are considered."

P3 L28: capturing **the** second-order parameter d-excess

Corrected.

P3 L32: "MJ79 already performs quite well for d-excess": Pfahl and Wernli (2009) would probably disagree.

Now we modify as: "since MJ79 the effect of convective mixing is larger on d-excess than on $\delta D$ (Risi et al. (2010); Benetti et al. (2014))." which does not contradict Pfahl and Wernli (2009).

P5 L23: r $\rightarrow r_{orig}$

Corrected.

P6 L20: measurement**s**

Corrected.

P6 L20: "Therefore, variations of $\delta D_0$ that are mediated by $q_0$ or $h_0$ do not interest us": But $\delta D$ in the FT is prescribed as a function of q (confusing).

We remove this confusing sentence and we write: "We attempt to express neither $h_0$ as a function of $q_0$ as in B15, and nor the $q$ profile as a function of $q_0$".

P8 L11: Refer to l'Hopital's rule?

Now we add: "(L'Hopital's rule was used to calculate this limit)."

P8 L21: follows a**s** mixing line

Corrected.

P9: Fig.3: $\alpha_{eff} = \alpha_{eq}$ instead of $\alpha_{eff} = 1/\alpha_{eq}$

Corrected.

P11 L25: "Only profiles during the ascending phase of the balloons are considered":
(Why?)

We now justify this choice: "Only profiles during the ascending phase of the balloon
are considered, because the descent phase is often located far away from the initial
launch point (McGrath et al. (2006); Seidel et al. (2011)). "

P11 L27 (title): write somewhere that these results are based on LMDZ output (not
observations)

Now we write: "Here we explain how $z_{orig}$ is estimated based on LMDZ outputs.". Later
in the sub-section, we write: "When estimating $z_{orig}$ from observations, we follow the
same methodology except that...".

P12 Fig. 5: Describe abbreviations (LCL, EIS, SCL) in caption.

Done

P12 L2: "if the end member is defined below 500hPa (e.g. 600hPa) results are not
always reasonable": In what sense? Why?

Now we write: "However, the end member should be defined above 500 hPa to ensure
that it is well above boundary layer processes. If the end member is defined below
500 hPa (e.g. 600 hPa), there are cases where $q$ increases with altitude ($q_f > q_0$)
due to horizontal advection or convective detrainment from nearby moister regions;

[Figure]

meanwhile, $\delta D$ decreases monotically, leading to unrealistic values for $\alpha_{eff}$."

P15 Fig. 7: What meteorological conditions do these examples represent? Would it be possible to show all (/more) simulated profiles in the background, e.g. in some transparent color, to get a better feeling for the variability? Also, I suggest adding markers to highlight where the levels are.

- Now we explain in the caption how these examples were selected (see response to rev 1).

- We will give more information on the type of meteorological conditions during these examples.

- This comments joins that from rev 1 who asks for a better documentation of the spatio-temporal variability among profiles. We will work on this question during the revision time to find the most adequate (and also concise) way to document and illustrate this variability.

- Now we add markers to highlight model levels.

P15 L1: Figure 8d instead of 8c.

Corrected

P15 L5: $\alpha_{eq}$ **as** a function of temperature

Corrected
P16 Fig. 8: in **boreal** winters of all years

Corrected

P17 L22: "in the cold upwelling regions": for example where?

Now we add: "cold upwelling regions, for example off Peru or Namibia"

P17 L23: probably reflect**s**

Corrected (here and also elsewhere)

P17 L24: "the effect of $r_{orig}$ can be seen on the composites as a function of EIS and not as a function of $\omega_{500}$ ": I don't see this, please elaborate.

We modify as: "We not that higher $r_{orig}$ in regions of stronger EIS contributes to the decrease of $\delta D_0$ with EIS (slightly decreasing green curve in Fig. 10b), but it does not contribute to the decrease of $\delta D_0$ with $\omega_{500}$ (flat green curve in Fig. 10a)."

P17 L30: followed by $h_0$ **(23%)**, $r_{orig}$ (16%), ...

Corrected

P18 Fig. 9: Are the correlations significant everywhere? Otherwise, add hatching where not significant?

In Fig. 9, we do not show the correlations, but rather the contributions on a $\delta D_0$ scale. When a map shows nearly constant values, it means that the contribution to $\delta D_0$ spatial

variations is small. When a map shows patterns that are similar to the simulated $\delta D_0$, it means that the contribution to $\delta D_0$ spatial variations is large. We add this explanation to the caption: "When a map shows patterns that are similar to the simulated $\delta D_0$, it means that the contribution to $\delta D_0$ spatial variations is large."

The spatial-seasonal correlations are shown in Table 1. We now write between brackets when correlations are not statistically significant at 99%. We write in the caption: "The threshold for the correlation coefficient to be statistically significant at 99 % is 0.15 or lower in all cases. We write correlation coefficient and slope values between brackets when they are not significant at 99%."

P19 Fig.10: $\omega_{500}$ (hPa/**d**)

Corrected

P20 Tab. 2: $q_0$ seems to be important in Fig. 12, but the slope is 0.0 here, $h_0$ seems to be unimportant in Fig. 12 but slope is 0.91 here. Why is that?

We now explain this: "Note that this effect can be seen only in most stable regions, but when considering all subsiding regions, the contribution is near zero (Table 2). "

P20 L1: "it would translate into a lower zorig.": Why?

We now explain this better at several places: section 3.3: "For example, in case of deep convection with depleting rain evaporation, a larger $r_{orig}$ is necessary to match the depleted $\delta D_0$, and a lower $z_{orig}$ is necessary to match this large $r_{orig}$." section 4.3: "if the large $r_{orig}$ was purely an artifact of the neglect of rain evaporation, it would translate totally into a lower $z_{orig}$, since a lower $z_{orig}$ is necessary to match a larger $r_{orig}$"

P22 Fig.12: $\omega_{500}$(hPa/d)

Corrected

P25 L6: the cruises goes

Corrected

P25 L8: "when considering only the 6 data points when $z_{orig}$ < 2000m": Rationale behind this?

We now clarify what we mean: "Remarkably, there are 6 days when $z_{orig}$ coincides with $z_i$ with a root means square error of 31 and correlation coefficient of 0.996 (Fig. 15c). This indicates that the air exactly comes from the inversion layer. When recalling that $z_{orig}$ and $z_i$ are estimated from completely independent observations, the coincidence is remarkable and lends support to the fact that on these days, our $z_{orig}$ estimate is physical. However, there remains 9 days when $z_{orig}$ is much higher than $z_i$. This may reflect more penetrative downdrafts as we approach deeper convective regimes. But it may also be an artifact of our neglect of horizontal advection. For example, on these days which are characterized by lower $h_0$, neglecting the advection of enriched water vapor from nearby regions with higher $h_0$ could be mis-interpreted as lower $r_{orig}$ and thus higher $z_{orig}$. ".

P25 L14: ... at the seasonal-spatial and daily scale is the proportion of the water vapor in the SCL that is originates from above

Corrected

P26 Fig. 15: r → $r_{orig}$

Corrected

P27 L1: there → they

Corrected

P27 L13: the **temporal** variability of $\alpha_{eff}$. Is it possible to estimate the uncertainty from the spatial variability of $\alpha_{eff}$ as well (in the vertical, i.e. how much the $\delta$ profile differs from a Rayleigh line with constant $\alpha_{eff}$)?

During the revision time, we will estimate this source of error. If the $\delta D$ profile doesn't follow a Rayleigh line with constant $\alpha_{eff}$, an analytical solution is not guaranteed, but a numerical solution can be found as long as $\delta D$ doesn't follow a mixing line. In the general case:

$$R_0 = \frac{\left(1 - \frac{q(z_{orig})}{q_0}\right) \cdot R_{oce}/\alpha_{eq} + \alpha_K \cdot (1 - h_0) \cdot \frac{q(z_{orig})}{q_0} \cdot R(z_{orig})}{\left(1 - \frac{q(z_{orig})}{q_0}\right) \cdot h_0 + \alpha_K \cdot (1 - h_0)} \tag{2}$$

We can thus numerically estimate the value for $z_{orig}$ that yields the simulated $R_0$, given the simulated vertical profiles of $q$ and $R$. We will compare this result with that obtained when assuming a Rayleigh line with constant $\alpha_{eff}$.

P27 L21: estimating $z_{orig}$ from $\delta D_0$measurements **on a daily basis** (?)

Now we write: "estimating $z_{orig}$ from daily $\delta D_0$ measurements cannot be useful unless we measure $\delta D$ profiles on a daily basis as well."

P28 L2: and if **we** measure

Corrected

P28 L3: swap trade-wind cumulus and strato-cumulus clouds

Corrected

P29 L14: very precise**d** estimates

Corrected

P29 L18: the altitude from which the air **is** originates, and is not **to** biase**d** by

Corrected

**References**

Benetti, M., Reverdin, G., Pierre, C., Merlivat, L., Risi, C., and Vimeux., F. (2014). Deuterium excess in marine water vapor: Dependency on relative humidity and surface wind speed during evaporation. *J. Geophys. Res*, 119:584–593, DOI: 10.1002/2013JD020535.

Bony, S., Bellon, G., Klocke, D., Sherwood, S., Fermepin, S., and Denvil, S. (2013). Robust direct effect of carbon dioxide on tropical circulation and regional precipitation. *Nature Geoscience*, 6(6):447–451.

Bony, S., Dufresne, J.-L., Le Treut, H., Morcrette, J.-J., and Senior, C. (2004). On dynamic and thermodynamic components of cloud changes. *Climate Dynamics*, 22:71–86.

Dee, S. G., Nusbaumer, J., Bailey, A., Russell, J. M., Lee, J.-E., Konecky, B., Buenning, N. H., and Noone, D. C. (2018). Tracking the strength of the walker circulation with stable isotopes in water vapor. *Journal of Geophysical Research: Atmospheres*, 123(14):7254–7270.

Duynkerke, P. G., de Roode, S. R., van Zanten, M. C., Calvo, J., Cuxart, J., Cheinet, S., Chlond, A., Grenier, H., Jonker, P. J., Köhler, M., et al. (2004). Observations and numerical simulations of the diurnal cycle of the eurocs stratocumulus case. *Quarterly Journal of the Royal Meteorological Society: A journal of the atmospheric sciences, applied meteorology and physical oceanography*, 130(604):3269–3296.

Emanuel, K., Neelin, D., and Bretherton, C. (1994). On large-scale circulations in convecting atmospheres. *Quaterly Journal of the Royal Meteorological Society*, 120:1111–1143.

Garratt, J. R. (1994). The atmospheric boundary layer. *Earth-Science Reviews*, 37(1-2):89–134.

Lacour, J.-L., Flamant, C., Risi, C., Clerbaux, C., and Coheur, P.-F. (2017). Importance of the saharan heat low in controlling the north atlantic free tropospheric humidity budget deduced from iasi $\delta$d observations. *Atmospheric Chemistry and Physics*, 17:9645–9663.

Masunaga, H. and Sumi, Y. (2017). A toy model of tropical convection with a moisture storage closure. *Journal of Advances in Modeling Earth Systems*, 9(1):647–667.

McGrath, R., Semmler, T., Sweeney, C., and Wang, S. (2006). Impact of balloon drift errors in radiosonde data on climate statistics. *Journal of climate*, 19(14):3430–3442.

Oke, T. R. (1988). *Boundary layer climates*. Halsted press, New York.

Risi, C., Landais, A., Bony, S., Masson-Delmotte, V., Jouzel, J., and Vimeux, F. (2010). Understanding the 17O-excess glacial-interglacial variations in Vostok precipitation. *J. Geophys. Res*, 115, D10112:doi:10.1029/2008JD011535.

Seidel, D. J., Ao, C. O., and Li, K. (2010). Estimating climatological planetary boundary layer hheight from radiosonde observations: comparison of methods and uncertainty analysis. *J. Geophy. Res.*, 115, D16113.

Seidel, D. J., Sun, B., Pettey, M., and Reale, A. (2011). Global radiosonde balloon drift statistics. *Journal of Geophysical Research: Atmospheres*, 116(D7).

Siebert, P., Beyrich, F., Gryning, S. E., Joffre, S., and Rasmussen, A. (2000). Review and intercomparison of operational methods for the determination of the mixing height. *Atmos. Environ.*, 34:1001–1027.

**[ACPD](ACPD)**

Interactive
comment

Sorbjan, Z. (1989). *Structure of the atmospheric boundary layer*. Prentice Hall, Englewood Cliffs, N. J.

Stull, R. B. (1988). *An intruduction to boundary layer meteorology*. Dordrect Kluwer.

Wood, R. and Bretherton, C. S. (2006). On the relationship between stratiform low cloud cover and lower-tropospheric stability. *Journal of climate*, 19(24):6425–6432.

Yanai, M., Esbensen, S., and Chu, J.-H. (1973). Determination of bulk properties of tropical cloud clusters from large-scale heat and moisture budgets. *Journal of the Atmospheric Sciences*, 30(4):611–627.

---

## Author Response (AR2)

**Response to reviewers**

August 28, 2019

**Summary:**

We thank both reviewers for their comments. As detailed in the point-by-point response below, we have done two major modifications to the manuscript:

1. Now we explicitly take into account rain evaporation and horizontal advection in our model. These effects are described as simply as possible in the main text, and technical details are given in appendix. This allows us to rigorously quantify these effects on $\delta D_0$ variations and to address all the related comments. This also simplifies the interpretation of $r_{orig}$ variations. We also address explicitly the effect of rain evaporation and horizontal advection on $z_{orig}$ estimates, and have modified our abstract and conclusion accordingly.

2. Now we better document and discuss the spatio-temporal variability of free tropospheric profiles. However, since this is not the core of the paper, and consistent with reviewer 1's suggestion, we have moved this discussion to the appendix.

**1   Reviewer 1**

We thank reviewer 1 for his/her comments.

This paper presents an analytical steady state vertical mixing model to investigate the controls on the water vapor isotopic composition in the subcloud layer over the tropical oceans. It is a nicely simple model that considers the most important processes, which are surface evaporation and vertical mixing and predicts the subcloud layer water vapour isotope composition from a combination of mass balance equations for all isotope species. I enjoyed reading this paper very much, I particularly like the approach chosen for testing this analytical model, which combines model simulation data and ship-based observations. The ideas presented in this paper are exciting and very valuable for upcoming large field campaigns in which isotope observations are planned in different parts of the lower troposphere.

I thus recommend minor revisions with the following minor points:

1) In the abstract it should be clearly stated that the proposed analytical model is a steady state formulation, which neglects horizontal gradients and thus the impact of horizontal advection.

- Regarding the steady state formulation. Now we write in the abstract: "The model relies on the assumption that $\delta D$ profiles are steeper than mixing lines, and that the SCL is at steady state, restricting its applications to time scales longer than daily."

- Regarding horizontal advection. Now we account for it explicitly. We write in the abstract: "We extend previous simple box models of the SCL by prescribing the shape of $\delta D$ vertical profiles as a function of humidity profiles and by accounting for rain evaporation and horizontal advection effects."

2) P. 1, L. 10: "When the air mixing into the SCL is lower in altitude it is moister": I think this is true most of the time and certainly in a climatological sense, but of course when including differential advection elevated moist layers can occur. I guess adding "it is generally moister" would make me very happy.
We modify as suggested.

3) P. 3, L. 4: Shouldn't cloud top cooling be mentioned here as well?

We modify as: "driven by cloud-top radiative cooling, mixing and evaporative cooling of droplets"

4) P. 5, L. 4: Neglecting the large-scale horizontal gradients in air properties, particularly in the trade wind regions seems to me like a strong assumption. Given the sensitivity of dD to SST and the considerable SST gradient across the North Atlantic, I find that this caveat could be discussed a bit more explicitly here.

We agree that the neglect of horizontal advection was an important caveat. Therefore, now we rigorously estimate the effect of horizontal advection of isotopic gradients on our results. This is detailed in section 2 in the main text. Eq. (8) provides the new equation for $R_0$ as a function of , the ratio of water vapor coming from horizontal advection to that coming from surface evaporation, and $\beta$, the ratio of isotopic ratios of horizontal advection to that of the SCL.

Coming back to the effect of horizontal gradients in the North Atlantic: our figure 7h shows that the effect of horizontal advection is not stronger in the North Atlantic than elsewhere. But it is larger near the Saharan coast. A later comment from you can be used to explain this feature: section 4.1: "Horizontal advection is slightly enriching in deep convective regions and depleting in coastal regions (e.g. off the coasts of California, Peru, Mauritania, Namibia, India and Australia). For example, the Saharan layer in front of the North-Western African Coast leads to a strong effect of horizontal advection there ([Lacour et al., 2017]). "

5) P. 5, L. 20: "qs is the saturation specific humidity at SST"
corrected

6) P. 6, L26: In the closure section and the discussion of the free tropospheric profile the role of horizontal advection is again neglected. This is maybe a good assumption in the tropic but it should still be mentioned explicitly.
Now we account for the effect of horizontal advection on $\delta D_0$ in our box model. However, our closure assumption does not need to neglect horizontal advection. Horizontal advection effects are implicitely taken into account through the $\alpha_{eff}$: we now clarify this: section 2.2: "Effects of horizontal advection and rain evaporation are encapsulated into $\alpha_{eff}$. "

7) P. 8, L26: "Depending on microphysical details that are too complex to be addressed here", Graf et al. 2019 could be referenced here
We add this reference.

8) P. 9. Fig. 3: Which value was chosen for the SST? This could be mentioned in the caption as well as a reference to which equilibrium fractionation factor was used

We add this information in the caption: "For this illustrative purpose, we assume SST=30℃, $h_0 = 0.8$ $\delta D_{oce} = 0‰$ and $\phi = \eta = 0$. ". We also add it to Fig. 4.

9) P. 12, L. 2: "However, if the end member is defined below 500 hPa (e.g. 600 hPa), results are not always reasonable", why is this so?

Now we explain this in the text: section 3.3: "However, the end member should be defined above 500 hPa to ensure that it is well above boundary layer processes. If the end member is defined below 500 hPa (e.g. 600 hPa), there are a few cases where $q$ increases with altitude ($q_f > q_0$) due to horizontal advection or convective detrainment from nearby moister regions. Meanwhile, $\delta D$ decreases monotonically, leading to unrealistic values for $\alpha_{eff}$.".

10) P. 12, L. 7: In my opinion, this makes it difficult to interpret rorig. But probably there are conditions when rorig and thus zorig are more physically meaningful than others. Could the authors maybe add a list with explicit and quantitatively expressed conditions in which they would argue that the assumptions involved in Eq. 9 are satisfied?

Now we explicitly account for rain evaporation and horizontal advection. Equations are given in the main text, so anyone can calculate these effects in their own cases. The contributions of these effects are plotted as maps in Fig. 7 g and h, and as composites as a function of $\omega_{500}$ and EIS in Fig. 8. Table 2 compares the effect of $r_{orig}$ if these effects are neglected or not. The effect of rain evaporation and horizontal advection on $z_{orig}$ estimates are plotted in Fig. 14b.

Note that a list of conditions would not be easy, since the effect of rain evaporation depends on parameters $\eta$ and $\alpha_{evap}$ which are very difficult to estimate in nature, and horizontal advection depends on parameters $\phi$ and $\beta$ which are resolution-dependent and depend on several variables (wind, humidity and $\delta D$ profiles in the SCL).

11) P. 13, L. 4-5: Is there a literature reference that the authors could indicated for this calculation of zi from observations?

We now give several literature references and more explanation on this calculation method: section 3.4: "The temperature inversion is an abrupt increase in temperature that caps the boundary layer. Therefore, a method to automatically estimate its altitude is to detect a maximum in the vertical gradient of potential temperature ([Stull, 1988, Oke, 1988, Sorbjan, 1989, Garratt, 1994, Siebert et al., 2000]). This method is sensitive to the resolution of vertical profiles ([Siebert et al., 2000, Seidel et al., 2010]). Therefore, we adapted this method in order to yield $z_i$ values that best agree with what we would estimate from visual inspection of individual temperature profiles. In LMDZ, we calculate $z_i$ as the first level at which the vertical potential temperature gradient exceeds 3 times the moist-adiabatic lapse rate. In observations, we calculate $z_i$ as the first level at which the vertical potential temperature gradient exceeds 5 times the moist-adiabatic lapse rate, because radio-soundings are noisier than simulated profiles. "

12) P. 13, L. 15: I did not immediately understand what was meant by composites belonging to a given interval of omega500, I was expecting a map. A reference to the results figure referred to here would have helped me.

We now explain better how the composites are calculated, and we add a reference to Fig. 10 as an example. Section 3.5: "The type of clouds and mixing processes depends strongly on the large-scale velocity at 500 hPa ($\omega_{500}$, map shown in Fig. 6a), with shallow clouds in subsiding regions and deeper clouds in ascending regions (Fig. 1). Therefore, it is convenient to plot variables as composites as a function of $\omega_{500}$ ([Bony et al., 2004]). To make such plots, we divide the $\omega_{500}$ range from -30 to 50 hPa/d into intervals of 5 hPa/d. In each given interval, we average all seasonal-mean values at all locations over tropical oceans for which seasonal-mean $\omega_{500}$ belongs to this interval (e.g. Fig. 10a will be an example). .

The cloud cover strongly correlates with the inversion strength, which can be quantified by the Estimated Inversion Strength (EIS, [Wood and Bretherton, 2006], map shown in Fig. 6b) as a measure of inversion strength. We thus also plot variables as composites as a function of EIS. To make such plots, we divide the EIS range from -1 K to 9 K into intervals of 0.5K. In each given interval, we average all seasonal-mean values at all locations over tropical oceans for which seasonal-mean EIS belongs to this interval (e.g. Fig. 10b will be an example). "

13) P. 13, L. 22: By how much (range of variability) were the four factors varied?

The four factors were varied from a control value to their simulated value.

Now we explain better the calculation with the contribution of $r_{orig}$ as an example, and we give all equations for other contributions in Table 1. Section 3.6: "To understand what controls the $\delta D_0$ spatio-temporal variations, $\delta D_0$ is decomposed into 4 contributions based on Eq. (8). First, we define $r_{orig,bas} = 0.3$, $\alpha_{eff,bas} = 1.09$, $SST_{bas}=25°C$, $h_{0,bas} = 0.7$, $\phi_{bas} = 0$, $\eta_{bas} = 0$, $\beta_{bas} = 1$ and $\alpha_{evap,bas} = 1$ as a basic state. We call $\delta D_{0,func}(r_{orig}, \alpha_{eff}, SST, h_0, \phi, \beta, \eta, \alpha_{evap})$ the function giving $\delta D_0$ as a function of $r_{orig}, \alpha_{eff}, SST, h_0, \phi, \beta, \eta$ and $\alpha_{evap}$ following Eq. (8), and $\delta D_{0,bas} = \delta D_{0,func}(r_{orig,bas}, \alpha_{eff,bas}, SST_{bas}, h_{0,bas}, \phi_{bas}, \beta_{bas}, \eta_{bas}, \alpha_{evap,bas})$. The relative contribution of $r_{orig}$ to $\delta D_0$ is estimated as $\delta D_{0,func}(r_{orig}, \alpha_{eff,bas}, SST_{bas}, h_{0,bas}, \phi_{bas}, \beta_{bas}, \eta_{bas}, \alpha_{evap,bas}) - \delta D_{0,bas}$. Similarly, the contributions of $\alpha_{eff}$, SST, $h_0$, $\phi$ and $\eta$ are to $\delta D_0$ are estimated as detailed in Table 1. All the contributions have the same units as $\delta D_0$ (‰).".

14) P. 14, Section 4.1: This section seemed very technical for me. I also see it more as a methodological aspect than a result. I would recommend to either shift it to a technical appendix (since the paper is quite long) or to the methods section.

We have now shifted this section to the appendix D.2. We have deeply modified this section to address comments from you and the other reviewer.

15) P. 15, Fig. 7: Mention that these are different random (?) grid points in the caption. I would have liked a more general evaluation also describing the temporal and spatial variability in the vertical profiles simulation by LMDZ.

We have removed this figure, which was misleading, and replaced it by a more general documentation of the temporal and spatial variability in vertical profiles simulated by LMDZ, to address this comment and those from the other reviewer. This discussion is in appendix D: "LMDZ free tropospheric profiles". We add a figure showing maps of a parameter $f = \frac{\delta D_{LMDZ} - \delta D_{Rayleigh}}{\delta D_{mix} - \delta D_{Rayleigh}}$ at 1000 m and 4000 m, representing how close is the simulated $\delta D$ compared to the Rayleigh and mixing lines (Fig. 16b,d). This documents the spatial variability in the shape of $\delta D$ profiles. To document the temporal variability, we add maps of the standard deviation of $f$ (Fig. 16c,e).

16) P. 15, L. 3: could the authors mention the region where they think that alphaeff may also reflect horizontal advection effects?

We now quantify the effect of horizontal advection on $\delta D_0$, i.e. in the SCL. However, the effect of horizontal advection on $\alpha_{eff}$ is a completely different subject that is beyond the scope of this paper. Therefore, we only reply based on the litterature: in appendix D: "The pattern of $\alpha_{eff}$ may also reflect horizontal advection effects, where strong isotopic gradients align with winds (e.g. from the Eastern to the Western Pacific, [Dee et al., 2018]). "
To clarify the scope of this paper, we start appendix D by stating: "The goal of this appendix is to document the spatio-temporal variability in the shape and steepness of simulated free tropospheric $\delta D$ profiles. Note that a detailed interpretation of these profiles is beyond the scope of this paper. This paper aims at understanding $\delta D_0$, which is the first step towards understanding full tropospheric profiles. In turn, understanding full tropospheric profiles in future studies will help refine our model for $\delta D_0$."

17) P.16: I find it interesting that the mixing and Rayleigh lines have large biases in front of the eastern continental boundaries where the inversion is strongest and, where there is a strong decoupling between the FT and BL. In particular in these regions, I would expect horizontal advection to play a key role, (e.g. the SAL layer in front of the eastern North African Coast, see Lacour et al. 2017, ACP). Maybe the authors find a good way to shortly note this in the text.

The map showing RMS errors was misleading. For example, if the simulated $\delta D$ behaves smoothly and is half-way between Rayleigh lines and mixing lines, it will result in local maxima in RMS values, but when we plot the new parameter $f = \frac{\delta D_{LMDZ} - \delta D_{Rayleigh}}{\delta D_{mix} - \delta D_{Rayleigh}}$, representing how close is the simulated $\delta D$ compared to the Rayleigh and mixing lines, nothing special emerges. This is the case in front of the Saharan coast. Therefore, we have not added this discussion here.
However, this comment is relevant for interpreting horizontal advection effects on $\delta D_0$: "Horizontal advection is slightly enriching in deep convective regions and depleting in coastal regions (e.g. off the coasts of California, Peru, Mauritania, Namibia, India and Australia). For example, the Saharan layer in front of the North-Western African Coast lead to a strong effect of horizontal advection ([Lacour et al., 2017])."

18)P. 17, L. 2: Maybe one could add oceanic upwelling and atmospheric deep convection. Jumping from upwelling to deep convection in the same sentence, I was not sure whether deep convection in the ocean or the atmosphere was meant here.

We modify as suggested.

19) P. 17, L. 4: "Decreases as omega500 is more strongly ascending or descending" -> "with increasing vertical winds (omega500) of both signs"
We modify as suggested.

20) P. 17, L. 11: "in more ascending regions" -> a reference to Fig. 8d would have helped me here.
We add this reference

211    21) P. 17, L. 25: "The fact that the effect..." I had difficulties to understand this sentence.

212

213    We remove this sentence that was not so useful.

214

215    22) P. 17, L. 33: $h_0$ (62%) is the largest explained fraction of all the variables considered and should thus be put
216 first. This could be a hint that large-scale horizontal advection plays an important role at the synoptic timescale
217 in these regions.

218

219    The numbers have changed because (1) we now account for rain evaporation and horizontal advection, and
220 (2) we now add an additional condition to make our calculation: section 3.1: "to avoid numerical problems when
221 estimating effect of horizontal advection and rain evaporation, only grid boxes and days where $E > 0.5$ mm/d are
222 considered. This represents 99.7% of all tropical oceanic grid boxes."
223    With the new values, the effect of $h_0$ is reduced and the effect of $r_{orig}$ is enhanced (table 3). Section 4.1: "In
224 subsiding regions, the effect of SST is muted due to its slow variability, and $r_{orig}$ (82 %) becomes the main factor.
225 "

226

227    23) P. 19, Fig. 10: The bin sizes (number of data points per bin should be added).
228    We now add this information in Fig 8, and we write in the caption: "The number of samples in each bin is
229 indicated on a logarithmic scale on the right-hand-side (dotted black line)."

230

231    If I understood correctly from the caption, the authors used the seasonal averaged fields from LMDZ. Why not
232 making these composites using the 6-hourly outputs? For me there is a timescale discrepancy between the processes
233 (mixing, evaporation) that the authors look at and the averaging timescale of the used fields.

234

235    The composites are based on seasonal-mean EIS or $\omega_{500}$ because this allows a better link with the large-scale
236 dynamical regime. Mixing and evaporation are processes that act at short time scales, but their relationship to the
237 large-scale circulation is best constrained by energetics at time scales longer than synoptic. Let's consider $\omega_{500}$, for
238 example. It relates to convective activity and other diabatic processes through the conservation equation of moist
239 static energy. Adiabatic cooling by large-scale ascent balances latent heating by convection ([Yanai et al., 1973]),
240 or adiabatic heating by large-scale subsidence balances radiative cooling ([Emanuel et al., 1994]). The stationarity
241 in the conservation of moist static equation is most valid at time scales longer than synoptic, otherwise, the storage
242 term becomes important ([Masunaga and Sumi, 2017]). This is why in [Bony et al., 2004] and subsequent papers
243 (e.g. [Bony et al., 2013]) based on $\omega_{500}$, monthly-mean $\omega_{500}$ is used, which yields similar results to seasonal-mean.
244    A similar rationale applies to EIS. This is why many papers on EIS use seasonal-mean values, notably the paper
245 defining this quantity ([Wood and Bretherton, 2006]).

246

247    We now explain this in section 3.5: "Note that such composites are done on seasonal-mean $\omega_{500}$ because cloud
248 processes and their associated diabatic heating are tied to the large-scale circulation through energetic constrains
249 ([Yanai et al., 1973, Emanuel et al., 1994]) that are best valid at longer time scales (otherwise, the energy storage
250 term may become significant, e.g. [Masunaga and Sumi, 2017]). This is why $\omega_{500}$ is generally averaged over a month
251 or longer (e.g. [Bony et al., 1997, Williams et al., 2003, Bony et al., 2004, Wyant et al., 2006, Bony et al., 2013]).
252 In addition, we primarily focus on understanding the seasonal and spatial distribution of $\delta D_0$."
253    and for EIS: "Using seasonal-mean values is consistent with [Wood and Bretherton, 2006] and with the better
254 link at longer time scales between cloud processes and the large-scale dynamical regime.".

255

256    In addition, we do not look at the diurnal variations: the stationarity assumption in our simple model would be
257 violated. We now explain this in the abstract: "the steady-state assumption restricts the application of this model
258 to time scales longer than daily." and in section 2.1: "We assume that the SCL is at steady state. For example, its
259 depth is constant. Since the SCL properties may exhibit a diurnal cycle ([Duynkerke et al., 2004]), this hypothesis
260 restricts the application of this model to time scales longer than daily. "

261

262    24) P. 27, L. 28: a reference to a more technical paper such as Aemisegger et al. 2012 AMT, would be nice here.
263    We add this reference

**Small technical comments:**

265 1) P. 2, L. 22 : "suffers **from** a low bias"

2) P. 3, L. 28: "capturing **the** second-order..."

3) P. 8, L. 26: no parenthesis after B)

4) P. 12, L.7: "based" -> "biased"

5) P. 15, L . 1: Figure 8d

6) P. 15, L. 4: "Values **of** alphaeff..."

7) P. 15, L. 5: using a fractionation coefficient alpha eq **as** a function of temperature"

8) P. 17, L. 14: space missing between rorig and (

9) P. 18, L. 1: "Overall, **the** results..."

We correct all these mistakes.

10) In general, the authors do not consistently use B15 for Benetti et al. (2015)

We now use B15 consistently.

11) P. 21, L. 2: "with **the** strongest inversion"

12) P. 27, L. 27: measurement errors

13) P. 28, L. 2: "if **we** measure..."

14) P. 29, L. 14: very precise

15) P. 30, L. 11: **from** which altitude the air comes

We correct all these mistakes.

**2    Reviewer 2**

We thank reviewer 2 for his/her comments.

This paper presents a simple box model solving the water isotope budget in the sub-cloud layer to quantify the relative contributions of sea surface temperature, relative humidity, mid-tropospheric depletion, and the fraction of moisture from the free troposphere (rorig) on the variability of $\delta D$ in near-surface water vapor ($\delta D_0$). The contribution of $r_{orig}$ is further separated into contributions of specific humidity at the surface, and the height ($z_{orig}$), relative humidity and temperature from which the free tropospheric air originates. Zorigis found to be an important factor explaining the seasonal-spatial and daily variations of $\delta D_0$. This means that measurements of Dδ0, if precise enough, can potentially be used to estimate $z_{orig}$ and distinguish between different mixing processes in the atmosphere.

The paper is interesting and well written, and it nicely demonstrates the use of measuring water vapor isotopes on short time scales. The box model's theoretical framework is describedin detail and its drawbacks are clearly identified by the authors. I only have a few comments about the methods, the rest are mainly ideas for clarifying the paper. I recommend that the paper be published after minor revisions.

**General comments**

1) I like the method for quantifying the contributions of different factors by linear regression. I see how this works when the contributing factors have the same units as the variable of interest, which was the case in the previous studies that used this method and are cited in this paper (Risi et al. 2010, Oueslati et al., 2016). Here the different factors all have different units, and the slope therefore depends on the units, or how much the components vary. I assume this was accounted for somehow, as the slopes in the tables are all unitless, but it is not clear from the text, and makes me a bit skeptical about the results. More explanation on that would be useful.

The contributing factors have the same units as the variable of interest, i.e. permil. We now clarify this in the text. We also explain better the calculation with the contribution of $r_{orig}$ as an example, and we give all equations for other contributions in Table 1. Section 3.6: "To understand what controls the $\delta D_0$ spatio-temporal variations, $\delta D_0$ is decomposed into 4 contributions based on Eq. (8). First, we define $r_{orig,bas} = 0.3$, $\alpha_{eff,bas} = 1.09$, $SST_{bas}=25°C$, $h_{0,bas} = 0.7$, $\phi_{bas} = 0$, $\eta_{bas} = 0$, $\beta_{bas} = 1$ and $\alpha_{evap,bas} = 1$ as a basic state. We call $\delta D_{0,func}(r_{orig}, \alpha_{eff}, SST, h_0, \phi, \beta, \eta, \alpha_{evap})$ the function giving $\delta D_0$ as a function of $r_{orig}$, $\alpha_{eff}$, SST, $h_0$, $\phi$, $\beta$, $\eta$ and $\alpha_{evap}$ following Eq. (8), and $\delta D_{0,bas} = \delta D_{0,func}(r_{orig,bas}, \alpha_{eff,bas}, SST_{bas}, h_{0,bas}, \phi_{bas}, \beta_{bas}, \eta_{bas}, \alpha_{evap,bas})$. The relative contribution of $r_{orig}$ to $\delta D_0$ is estimated as $\delta D_{0,func}(r_{orig}, \alpha_{eff,bas}, SST_{bas}, h_{0,bas}, \phi_{bas}, \beta_{bas}, \eta_{bas}, \alpha_{evap,bas}) - \delta D_{0,bas}$. Similarly, the contributions of $\alpha_{eff}$, SST, $h_0$, $\phi$ and $\eta$ are to $\delta D_0$ are estimated as detailed in Table 1. All the contributions have the same units as $\delta D_0$ (‰).".

2) As stated in the paper, the methods rely on the assumption that the $\delta D$ profile follows a Rayleigh-like line, and that there is no effect of rain evaporation. Figures 7 and 8 show that the $\delta D$ profile is often closer to a mixing line than a Rayleigh line, and the large contribution of $r_{orig}$ mainly comes from ascending regions, where clouds are most likely precipitating. It would be nice to see some quantification of how this impacts the results. A possible way to do this is to remove days/locations where the RMSE of the mixing line is smaller than the RMSE of the Rayleigh line and where there is precipitation, then repeat the analysis for these new fields and add the results in brackets in Tables 1, 2 and as dotted lines in Figures 10, 12.

- Effect of rain evaporation. Now we explicitly account for the effect of rain evaporation. Equations are given in the main text. The contributions of this effect are plotted as maps in Fig. 7g, and as composites as a function of $\omega_{500}$ and EIS in Fig. 8. Table 2 compares the effect of $r_{orig}$ if rain evaporation and horizontal advection are neglected or not. The effect of rain evaporation on $z_{orig}$ estimates are plotted in Fig. 14b.

- Effect of the deviation of the $\delta D$ profile from a Rayleigh distillation line. This is more difficult to quantify. Now we discuss this in detail in the discussion section: subsection 6.5 untitled "Rayleigh assumption for the shape of $\delta D$ profiles": "Finally, a fifth source of uncertainty comes the assumption that the $\delta D$ profile follows a Rayleigh distillation line (section 2.2). However, both in LMDZ (appendix D) and nature ([Sodemann et al., 2017]), $\delta D$ profiles are usually intermediate between Rayleigh and mixing lines. The precision of our $z_{orig}$ estimate is maximum in the Rayleigh distillation case. When trying to directly find a numerical solution for $z_{orig}$ from Eq. (6), a solution can be found only in 0.1 % of cases. This is because... However, it is possible that $\delta D$ profiles simulated by LMDZ are closer to mixing lines than real profiles, since GCMs are known to overestimate vertical mixing through the troposphere ([Risi et al., 2012]) and to mix the lower free troposphere too frequently by deep convection in trade-wind regions ([Nuijens et al., 2015b, Nuijens et al., 2015a]). Therefore, the shape of $\delta D$ profiles simulated by LMDZ is not a sufficient reason to reject the Rayleigh assumption. The uncertainty associated with this assumption is very difficult to quantify in LMDZ. More measurements of full $\delta D$ profiles are very welcome to help quantify it."

3) The paper presents the new box model as an extension of the model by Benetti et al. (2015), which is technically true, but can be a bit misleading because its application is different. Rather than predicting $\delta D_0$ from zorig, it predicts $z_{orig}$ from $\delta D_0$ and therefore requires $\delta D_0$ to be known. This means it cannot be applied to initialize Rayleigh models like the model by Benetti et al. (2015), which assumes constant zorig. This could be written more clearly (e.g., from the abstract it seems like the model can be used to predict $\delta D_0$, which is only possible if $z_{orig}$ is known).

The model can be used both ways, either to predict $\delta D_0$ as a function of $z_{orig}$, or to estimate $z_{orig}$ from $\delta D_0$. We now clarify this:

- in the abstract: "The model express $\delta D_0$ as a function of $z_{orig}$, humidity and temperature profiles, surface conditions a parameter describing the steepness of the $\delta D$ vertical gradient and a few parameters describing rain evaporation and horizontal advection effects. ".

- in section 2.2: "If the goal is to predict $R_0$ from $z_{orig}$, we can apply Eq. (6) if we know the $q$ and $\delta D$ vertical profiles. Conversely, if the goal is to predict $z_{orig}$ from $R_0$, we can numerically solve Eq. (6) if we know the $q$ and $\delta D$ vertical profiles.".

4) Changing some of the colors and colormaps could make the figures easier to understand. For example, I think the contributions of different factors and how they add up in Figures 9 and 11 would be more intuitive with a perceptually uniform colormap going from light to dark colors. Also, the red and pink lines in Figures 3, 4, 7, 10, 12, 15 look very similar to each other. It would be good to use a different color for one of them.

- Now all maps are colored in shaded of blue for negative values and shaded of red for positive values.

- We now change the line color from pink to purple to distinguish it from red (Figures 3, 4, 5), or from red to dark red to distinguish it from pink (Figures 8, 10, 13).

**Specific comments**

P1 L13: [D]/[H] instead of [HDO]/[H2O]
We modify.

P2 L22: high bias instead of low bias?
Corrected.

P2 L30: Please introduce the abbreviation for LCL
Done.

P3 L4: pointed **out** the important role
Corrected.

P3 L15: "We do not call it entrained": The word entrained/entrainment still appears a few times in the text (e.g. in Fig. 2, the title of section 4.4)
We modify all the occurences of "entrainment", except when it really refers to entrainment.

P3 L23: during **a** field campaign, global outputs **of** an isotope-enabled GCM.
Corrected.

P3 L24: "at the global scale": Really? There are no global maps. Are the numbers in Tables 1, 2 and the lines in Figures 10, 12 from global output, or from the region shown on the maps?
We now modify by "in the Tropics". We precise in the captions for these Tables and Figures: "All seasons and locations over tropical oceans ($30°N - 30°S$, ocean fraction>80%) are considered."

P3 L28: capturing **the** second-order parameter d-excess
Corrected.

P3 L32: "MJ79 already performs quite well for d-excess": Pfahl and Wernli (2009) would probably disagree.
Now we modify as (section 1.3): "In addition, the need for an extension of MJ79 is more needed for $\delta D$ than for d-excess, since the effect of convective mixing is larger on $\delta D$ than on d-excess ([Risi et al., 2010, Benetti et al., 2014]).". This does not contradict Pfahl and Wernli (2009).

P5 L23: r $\rightarrow r_{orig}$
Corrected.

P6 L20: measurement**s**
Corrected.

P6 L20: "Therefore, variations of $\delta D_0$ that are mediated by $q_0$ or $h_0$ do not interest us": But $\delta D$ in the FT is prescribed as a function of q (confusing).
We remove this confusing sentence and we write (section 2.1): "We attempt to express neither $h_0$ as a function of $q_0$ as in B15, and nor the $q$ profile as a function of $q_0$".

P8 L11: Refer to l'Hopital's rule?
Now we add: "(L'Hopital's rule was used to calculate this limit)." (section 2.2°

P8 L21: follows **as** mixing line
Corrected.

P9: Fig.3: $\alpha_{eff} = \alpha_{eq}$ instead of $\alpha_{eff} = 1/\alpha_{eq}$
Corrected.

P11 L25: "Only profiles during the ascending phase of the balloons are considered": (Why?)

We now justify this choice: "Only profiles during the ascending phase of the balloon are considered, because the descent phase is often located far away from the initial launch point ([McGrath et al., 2006, Seidel et al., 2011]). " (section 3.2)

P11 L27 (title): write somewhere that these results are based on LMDZ output (not observations)

Now we write (section 3.3): "Here we explain how $z_{orig}$ is estimated based on LMDZ outputs.". Later in the section, we write: "When estimating $z_{orig}$ from observations, we follow the same methodology except that...".

P12 Fig. 5: Describe abbreviations (LCL, EIS, SCL) in caption.

Done

P12 L2: "if the end member is defined below 500hPa (e.g. 600hPa) results are not always reasonable": In what sense? Why?

Now we write: "However, the end member should be defined above 500 hPa to ensure that it is well above boundary layer processes. If the end member is defined below 500 hPa (e.g. 600 hPa), there are cases where $q$ increases with altitude ($q_f > q_0$) due to horizontal advection or convective detrainment from nearby moister regions; meanwhile, $\delta D$ decreases monotically, leading to unrealistic values for $\alpha_{eff}$."

P15 Fig. 7: What meteorological conditions do these examples represent? Would it be possible to show all (/more) simulated profiles in the background, e.g. in some transparent color, to get a better feeling for the variability? Also, I suggest adding markers to highlight where the levels are.

We have removed this figure, because it was misleading. We replace it by Fig. 16 in appendix D to better document the temporal and spatial variability in free tropospheric profiles. Maps show parameter $f = \frac{\delta D_{LMDZ} - \delta D_{Rayleigh}}{\delta D_{mix} - \delta D_{Rayleigh}}$ describing whether simulated $\delta D$ is closer to Rayleigh line or a mixing line (Fig 16b,d). We also show maps for the standard deviation of $f$ to illustrate the temporal variability (Fig 16 c,e).

In addition, markers are added in Fig 16a to highlight the model levels.

P15 L1: Figure 8d instead of 8c.

Corrected

P15 L5: $\alpha_{eq}$ **as** a function of temperature

Corrected

P16 Fig. 8: in **boreal** winters of all years

Corrected

P17 L22: "in the cold upwelling regions": for example where?

Now we add: "cold upwelling regions, for example off Peru or Namibia" (section 4.1)

P17 L23: probably reflect**s**

This sentence was removed because $r_{orig}$ cannot reflect rain evaporation any more.

P17 L24: "the effect of $r_{orig}$ can be seen on the composites as a function of EIS and not as a function of $\omega_{500}$": I don't see this, please elaborate.

This sentence was removed.

P17 L30: followed by $h_0$ **(23%)**, $r_{orig}$ (16%), ...

Corrected

P18 Fig. 9: Are the correlations significant everywhere? Otherwise, add hatching where not significant?

In this figure (now Fig. 7), we do not show the correlations, but rather the contributions to $\delta D_0$ on a ‰ scale. We now explain better in the text how these contributions are calculated (section 3.6), and we add table 1 to give the exact equations used to calculate each contribution.

⁴⁷⁹    The spatial-seasonal correlations are shown in Table 2. We now write between brackets when correlations are
⁴⁸⁰ not statistically significant at 99%. We write in the caption: "The threshold for the correlation coefficient to be
⁴⁸¹ statistically significant at 99 % is 0.15 or lower in all cases. We write correlation coefficient and slope values between
⁴⁸² brackets when they are not significant at 99%."

⁴⁸⁴    P19 Fig.10: $\omega_{500}$ (hPa/**d**)
⁴⁸⁵    Corrected

⁴⁸⁷    P20 Tab. 2: $q_0$ seems to be important in Fig. 12, but the slope is 0.0 here, $h_0$ seems to be unimportant in Fig.
⁴⁸⁸ 12 but slope is 0.91 here. Why is that?
⁴⁸⁹    We now explain this: section 4.2: "Note that this effect can be seen only in most stable regions, but when
⁴⁹⁰ considering all subsiding regions, the contribution is small (Table 2). "

⁴⁹²    P20 L1: "it would translate into a lower zorig.": Why?

⁴⁹⁴    For example, in case of deep convection with depleting rain evaporation, a larger $r_{orig}$ is necessary to match the
⁴⁹⁵ depleted $\delta D_0$, and a lower $z_{orig}$ is necessary to match this large $r_{orig}$.
⁴⁹⁶    Now this sentence is removed, since we explicitly account for rain evaporation and horizontal advection. We
⁴⁹⁷ do not need this kind of rationale any more.

⁴⁹⁹    P22 Fig.12: $\omega_{500}$(hPa/d)
⁵⁰⁰    Corrected

⁵⁰²    P25 L6: the cruises goes
⁵⁰³    Corrected

⁵⁰⁵    P25 L8: "when considering only the 6 data points when $z_{orig} < 2000m$": Rationale behind this?

⁵⁰⁷    We now clarify what we mean: section 5: "Remarkably, there are 6 days when $z_{orig}$ coincides with $z_i$ with a root
⁵⁰⁸ means square error of 31‰ and correlation coefficient of 0.996 (Fig. 15c). This indicates that the air exactly comes
⁵⁰⁹ from the inversion layer. When recalling that $z_{orig}$ and $z_i$ are estimated from completely independent observations,
⁵¹⁰ the coincidence is remarkable and lends support to the fact that on these days, our $z_{orig}$ estimate is physical.
⁵¹¹ However, there remains 9 days when $z_{orig}$ is much higher than $z_i$. This may reflect more penetrative downdrafts as
⁵¹² we approach deeper convective regimes. But it may also be an artifact of our neglect of horizontal advection. For
⁵¹³ example, on these days which are characterized by lower $h_0$, neglecting the advection of enriched water vapor from
⁵¹⁴ nearby regions with higher $h_0$ could be mis-interpreted as lower $r_{orig}$ and thus higher $z_{orig}$. ".

⁵¹⁶    P25 L14: ... at the seasonal-spatial and daily scale is the proportion of the water vapor in the SCL that is
⁵¹⁷ originates from above
⁵¹⁸    Corrected

⁵²⁰    P26 Fig. 15: r → $r_{orig}$
⁵²¹    Corrected

⁵²³    P27 L1: there → they
⁵²⁴    Corrected

⁵²⁶    P27 L13: the **temporal** variability of $\alpha_{eff}$. Is it possible to estimate the uncertainty from the spatial variability
⁵²⁷ of $\alpha_{eff}$ as well (in the vertical, i.e. how much the $\delta$ profile differs from a Rayleigh line with constant $\alpha_{eff}$)?

⁵²⁹    We now better document the spatio-temporal variability in the shape of free tropospheric $\delta D$ profiles in the
⁵³⁰ appendix D.1. To address this specific comment, we now plot parameter $f = \frac{\delta D_{LMDZ} - \delta D_{Rayleigh}}{\delta D_{mix} - \delta D_{Rayleigh}}$ describing whether
⁵³¹ simulated $\delta D$ is closer to Rayleigh line or a mixing line. We show vertical profiles and maps of $f$ at both 1000 m
⁵³² and 4000 m (Fig 16 b,d).
⁵³³    We tried to compare the $z_{orig}$ estimate with and without the assumption that the $\delta D$ profile follows a Rayleigh
⁵³⁴ line. However, it didn't work as well as expected. We now explain this when examining all errors on $z_{orig}$ (section

6.5): "When trying to find a numerical solution for $z_{orig}$ directly from Eq. (6), a solution can be found only in 0.1 % of cases.". We explain why, and we also explain why it may work better in nature.

P27 L21: estimating $z_{orig}$ from $\delta D_0$ measurements **on a daily basis** (?)

Now we write: "estimating $z_{orig}$ from daily $\delta D_0$ measurements cannot be useful unless we measure $\delta D$ profiles on a daily basis as well."

P28 L2: and if **we** measure
Corrected

P28 L3: swap trade-wind cumulus and strato-cumulus clouds
Corrected

P29 L14: very precise**d** estimates
Corrected

[revised manuscript text omitted]

**Response to Co-editor report**

September 5, 2019

- P1, L8-10: These two sentence are quite difficult to follow and I would suggest to rephrase these. Do you mean "In the model delD0 is expressed as......"? What do you mean with "dampening hope"? I guess the sentence is not complete and thus confusing.

We now write: "In the model, $\delta D_0$ is expressed as a function of $z_{orig}$, humidity and temperature profiles, surface conditions, a parameter describing the steepness of the $\delta D$ vertical gradient and a few parameters describing rain evaporation and horizontal advection effects. We show that $\delta D_0$ does not depend on the intensity of entrainment, in contrast to several previous studies that had hoped that $\delta D_0$ measurements could help estimate this quantity."

- P1, L15: This paragraph should be combined with the previous paragraph.

Done

- P1: Since you do not have any copyright statement this line is obsolete and can be deleted.

OK

- P3, 6-7: Space between number and unit is missing. This should be corrected throughout the manuscript.

Done

- P3, L20: Here you should use the abbreviation "B15" you introduced before.

Done

- P11, Fig 4 caption: "If delD profiles follow a Rayleigh distillation....." Not entirely clear. I understand that this is the figure for the case where the delD profiles follow a Rayleigh distillation. I would clearly state that and write here e.g.: " (a) For the case where the delD profiles follow the Rayleigh distillation". Same holds than for the figure caption of (b).

Now we clarify by writing: "for the case where the tropospheric $\delta D$ profile follow Rayleigh distillation (a) or a mixing line (b)."

- P13, Fig 5 caption: "\citetet" instead of "\citep" should be used here.

Done

- P13, L5: strato-cumulus -> stratocumulus (?)

OK, replaced everywhere

- P15, L2: table 1 -> Table 1

OK, replaced everywhere

- P15, L10: To avoid the double parentheses you should use in latex: \citep[e.g.][]{ref1,ref2} for the references.

Done

- P15, L7: Same here.

Done

- P17, L33: from higher in altitude -> from higher altitudes

Done

- P18, Fig 7 caption: For better readability I suggets the following changes a) -> (a) and in the text then instead of just a -> (a), Same holds for panel (b).

Corrected here and everywhere

- P19, Fig 8 legend: add a space between "season" and "location" so that it reads "(season, location)".

Done

- P19, Fig. 8: Why is the y-axes labelling at the right axes not done for all ticks?

Because the number of samples doesn't go beyond $10^4$. But we have now added all the tick labels.

- P20, Table 2: ascending -> Ascending

Corrected here and everywhere

- P24, Fig 11: Same here as for Foig 7.

OK

- Further, the writing of the z_orig is a bit confusing. I would not write the r_orig $= 0.6$ as subscript. I would suggest to write in parentheses z_orig (r_orig $= 0.6$).

Done everywhere

- P26, L14: increasingly higher in altitude -> increasingly higher altitudes

Done

- P29, L7: to -> too

Done

- P29, L25: space between parentheses and delD_f obsolete.

Done

- P29, L26: use write the text first and then give the citations wit \citep (e.g. on Islands such as ....\citep{ref1,ref2,ref3})

We now write: "(e.g. on Islands such as Hawaii or La Réunion: Galewsky et al., 2007; Bailey et al., 2013; Guilpart et al., 2017) ."
We have improved the citation format everywhere.

- P29, L28: Finally, as fifth -> Finally, as a fifth

Done

- P30, L4: appendix -> Appendix

Done here and everywhere

- P30, L19: Benetti et al. (2015) -> B15

Done

- P30, L19: make explicit the -> make the explicit

Done

- P30, L28: hundreds -> hundred

Done

- P33, L3: visualize -> visualized. Further, I would rather write has been than can be since you have done it and shown in Fig. 4.

We write: "...into the SCL (Fig. 4b)."

- Fig C1 and D1: There are lost parentheses without content in the figure titles.

This was to mean there was no unit. Now we remove these parentheses, here and everywhere.